# Matrix Denoising with Doubly Heteroscedastic Noise: Fundamental Limits and Optimal Spectral Methods

**Yihan Zhang**
Institute of Science and Technology Austria
zephyr.z798@gmail.com

**Marco Mondelli**
Institute of Science and Technology Austria
marco.mondelli@ist.ac.at

## Abstract

We study the matrix denoising problem of estimating the singular vectors of a rank-1 signal corrupted by noise with both column and row correlations. Existing works are either unable to pinpoint the exact asymptotic estimation error or, when they do so, the resulting approaches (e.g., based on whitening or singular value shrinkage) remain vastly suboptimal. On top of this, most of the literature has focused on the special case of estimating the left singular vector of the signal when the noise only possesses row correlation (one-sided heteroscedasticity). In contrast, our work establishes the information-theoretic and algorithmic limits of matrix denoising with doubly heteroscedastic noise. We characterize the exact asymptotic minimum mean square error, and design a novel spectral estimator with rigorous optimality guarantees: under a technical condition, it attains positive correlation with the signals whenever information-theoretically possible and, for one-sided heteroscedasticity, it also achieves the Bayes-optimal error. Numerical experiments demonstrate the significant advantage of our theoretically principled method with the state of the art. The proofs draw connections with statistical physics and approximate message passing, departing drastically from standard random matrix theory techniques.

## 1 Introduction

Matrix denoising is a central primitive in statistics and machine learning, and the problem is to recover a signal $X \in \mathbb{R}^{n \times d}$ from an observation $A = X + W$ corrupted by additive noise $W$. This finds applications across multiple domains of sciences, e.g., imaging [24, 63], biology [14, 46] and astronomy [70, 5]. When $X$ has low rank and $W$ i.i.d. entries, $A$ is the standard model for principal component analysis, typically referred to as the Johnstone spiked covariance model [42]. When $n, d$ are both large and proportional, which corresponds to the most sample-efficient regime, its Bayes-optimal limits are well understood [52], and it has been established how to achieve them efficiently [56]. Minimax/non-asymptotic guarantees are also available in special cases, such as sparse PCA [18], Gaussian mixtures [72] and certain joint scalings of $(n, d)$ [57].

However, in most applications, noise is highly structured and correlated, thereby calling for more realistic assumptions on $W$ than having i.i.d. entries. A recent line of work addresses this concern by studying matrix denoising with heteroscedastic noise [1, 69, 33, 44, 26], resting on two basic ideas: whitening and singular value shrinkage. Whitening refers to multiplying the data matrix by the square root of the inverse covariance, in order to reduce the model to one with i.i.d. noise; and singular value shrinkage retains the singular vectors of the data while deflating the singular values to correct for the noise. Though the exact asymptotic performance of these algorithms has been derived [69, 33, 44, 26], their optimality is yet to be determined from a Bayesian standpoint. In fact, we will prove that whitening and shrinkage are *not* the correct way to approach Bayes optimality.

38th Conference on Neural Information Processing Systems (NeurIPS 2024).

**Main contributions.** We focus on the prototypical model $A = X + W$, where $X = \frac{\lambda}{n}u^*v^{*\top}$ is a rank-1 signal, $\lambda$ is the signal-to-noise ratio (SNR), and $W = \Xi^{1/2}\widetilde{W}\Sigma^{1/2}$ is doubly heterogeneous noise. Here $u^*, v^*$ follow i.i.d. priors; $\widetilde{W}$ contains i.i.d. Gaussian entries; the covariance matrices $\Xi, \Sigma$ capture column and row correlations; and we consider the typical high-dimensional regime in which $n, d$ are both large and proportional. Our main results are summarized below.

1. We design an efficient spectral estimator to recover $u^*, v^*$, and we provide a precise asymptotic analysis of its performance, see Theorem 5.1. This estimator is given by the top singular vectors of a matrix obtained by carefully pre-processing $A$, see (5.3).

2. When the priors of $u^*, v^*$ are standard Gaussian, we show in Corollary 5.2 that the spectral estimator above is optimal in the following sense: *(i)* under a technical condition, it achieves the *optimal weak recovery threshold*, namely its mean square error is non-trivial as soon as this is information-theoretically possible; *(ii)* it achieves the *Bayes-optimal error* for $u^*$ (resp. $v^*$) when $\Xi$ (resp. $\Sigma$) is the identity. These optimality guarantees follow from rigorously obtaining the asymptotic minimum mean square error (MMSE) for the estimation of the whitened signals $\Xi^{-1/2}u^*$ and $\Sigma^{-1/2}v^*$, see Theorem 4.2.

Our spectral estimator only involves matrix multiplication and computing principal singular vectors. Practically, this can be efficiently done using standard SVD algorithms or power iteration [48]. For both one-sided and double heteroscedasticity, numerical experiments in Figures 2 and 3 show significant advantage of our spectral estimator for moderate SNRs over HeteroPCA [76] and shrinkage-based methods, i.e., Whiten-Shrink-reColor [44, 45], OptShrink [59], and ScreeNOT [27].

**Proof techniques.** We take a completely different route from classical approaches in statistics and random matrix theory (e.g., whitening and shrinkage), and instead exploit tools from statistical physics and the theory of approximate message passing. In particular, the MMSE for the whitened signals $\Xi^{-1/2}u^*, \Sigma^{-1/2}v^*$ is obtained via an interpolation argument [10, 52, 53]. This result allows us to derive the weak recovery threshold for estimating the true signals $u^*, v^*$. Moreover, for one-sided heteroscedasticity, this MMSE coincides with that for estimating the true signal on the homoscedastic side. Evaluating the Bayes-optimal estimators requires solving high-dimensional integrals that are computationally intractable. To circumvent this issue, we propose an efficient spectral method that still enjoys optimality guarantees. Its design and analysis draw connections with a family of iterative algorithms called approximate message passing (AMP) [11, 32]. All our results are mathematically rigorous, with the only technical condition being "(5.1) implies $\sigma_2^* < 1$" in Theorem 5.1 that we only managed to verify numerically, but not analytically; see Remark 5.1.

## 2 Related work

Research on matrix denoising in the homoscedastic case ($\Xi = I_n, \Sigma = I_d$) has a rich history, and in random matrix theory properties of the spectrum and eigenspaces of $A$ have been studied exhaustively. Most prominently, the BBP phase transition phenomenon [4] (and its finite-sample counterpart [60]) unveils a threshold of the SNR $\lambda$ above which a pair of outlier singular value and singular vector emerge. Under i.i.d. priors, the asymptotic Bayes-optimal estimation error has been derived [52, 53], rigorously justifying predictions from statistical physics [47]. The proof uses the interpolation method due to Guerra [35], originally developed in the context of mean-field spin glasses. Besides low-rank matrix estimation, this method (including its adaptive variant [10] and the Aizenman–Sims–Starr scheme [2]) has also been applied to a range of problems, including spiked tensor estimation [49], generalized linear models [9], stochastic block models [74] and group synchronization [73].

Moving to the heteroscedastic case, an active line of work concerns optimal singular value shrinkage methods [44, 33, 45, 69, 59, 26]. These methods can be regarded as a special family of rotationally invariant estimators, which apply a univariate function $\eta \colon \mathbb{R}_{\geq 0} \to \mathbb{R}$ to each empirical singular value. An example widely employed by practitioners is the thresholding function $\eta_\theta(y) = y\mathbb{1}\{y > \theta\}$ [27]. In the presence of noise heteroscedasticity, most of these results are based on whitening [43]. Another model of noise heterogeneity common in the literature takes $W = \widetilde{W} \circ \Delta^{\circ 1/2}$, where $\widetilde{W}$ has i.i.d. Gaussian entries, $\Delta$ is a deterministic block matrix with fixed (i.e., constant with respect to $n, d$) number of blocks, and $\circ$ denotes the element-wise product. This means that the entries of the noise are independent but non-identically distributed, and they follow the variance profile $\Delta$. The corresponding low-rank perturbation $A$, known as a spiked inhomogeneous matrix, has attracted attention from both the information-theoretic [12, 66, 36] and the algorithmic sides [38, 50, 61].

Spiked inhomogeneous matrices have some connections with the model considered in this paper: if $\Delta$ has rank 1, such $A$ can be realized by taking $\Xi, \Sigma$ to be diagonal with suitable block structures. Finally, non-asymptotic results for the heteroscedastic and the inhomogeneous models have been derived in varying generality in [76, 82, 23, 1, 17]. We highlight that our paper is the *first* to establish information-theoretic and algorithmic limits for doubly heteroscedastic noise.

Our characterization of the spectral estimator relies on an AMP algorithm that converges to it by performing power iteration. AMP refers to a family of iterative procedures, whose performance in the high-dimensional limit is precisely characterized by a low-dimensional deterministic recursion called state evolution [11, 15]. Originally introduced for compressed sensing [28], AMP algorithms have been developed for various settings, including low-rank estimation [56, 31, 6] and inference in generalized linear models [64, 65, 71]. Beyond statistical estimation, AMP proves its versatility as both an efficient algorithm and a proof technique for studying e.g. posterior sampling [58], spectral universality [30], first order methods with random data [22], mismatched estimation [8], spectral estimators for generalized linear models [79, 80] and their combination with linear estimators [54].

## 3  Problem setup

Consider the following rank-1 rectangular matrix estimation problem with doubly heteroscedastic noise where we observe

$$A = \frac{\lambda}{n} u^* v^{*\top} + W \in \mathbb{R}^{n \times d}, \tag{3.1}$$

and aim to estimate $u^*, v^*$. The following assumptions are imposed throughout the paper. The dimensions $n, d \to \infty$ obey the proportional scaling $n/d \to \delta \in (0, \infty)$, where $\delta$ is the aspect ratio. The SNR $\lambda \in [0, \infty)$ is a known constant (relative to $n, d$). The signals $(u^*, v^*) \sim P^{\otimes n} \otimes Q^{\otimes d}$ have i.i.d. priors, where $P, Q$ are distributions on $\mathbb{R}$ with mean 0 and variance 1. The unknown noise matrix has the form $W = \Xi^{1/2} \widetilde{W} \Sigma^{1/2} \in \mathbb{R}^{n \times d}$, with $\widetilde{W}_{i,j} \overset{\text{i.i.d.}}{\sim} \mathcal{N}(0, 1/n)$ independent of $(u^*, v^*)$. The covariances $\Xi \in \mathbb{R}^{n \times n}, \Sigma \in \mathbb{R}^{d \times d}$ are known, deterministic,[1] strictly positive definite and satisfy

$$\lim_{n \to \infty} \frac{1}{n} \operatorname{Tr}(\Xi) = \lim_{d \to \infty} \frac{1}{d} \operatorname{Tr}(\Sigma) = 1. \tag{3.2}$$

Their empirical spectral distributions (ESD) converge (as $n, d \to \infty$ s.t. $n/d \to \infty$) weakly to the laws of the random variables $\overline{\Xi}$ and $\overline{\Sigma}$. Furthermore, $\|\Xi\|_2, \|\Sigma\|_2$ are uniformly bounded over $d$. The supports of $\overline{\Xi}, \overline{\Sigma}$ are compact subsets of $(0, \infty)$. For all $\varepsilon > 0$, there exists $d_0 \in \mathbb{N}$ s.t. for all $d \geq d_0$,

$$\operatorname{supp}(\operatorname{ESD}(\Xi)) \subset \operatorname{supp}(\overline{\Xi}) + [-\varepsilon, \varepsilon], \quad \operatorname{supp}(\operatorname{ESD}(\Sigma)) \subset \operatorname{supp}(\overline{\Sigma}) + [-\varepsilon, \varepsilon]. \tag{3.3}$$

The trace assumption (3.2) on the covariances is for normalization purposes since the values of the traces, if not 1, can be absorbed into $\lambda$. The support assumption (3.3) excludes outliers in the spectra of covariances which may contribute to undesirable spikes in $A$ [69].

## 4  Information-theoretic limits

For mathematical convenience, in this section, we switch to an equivalent rescaled model

$$Y \coloneqq \sqrt{n} A = \sqrt{\frac{\gamma}{n}} u^* v^{*\top} + \Xi^{1/2} Z \Sigma^{1/2} \in \mathbb{R}^{n \times d}, \tag{4.1}$$

where $\gamma \coloneqq \lambda^2$ and $Z = \sqrt{n} \widetilde{W}$ contains i.i.d. elements $Z_{i,j} \overset{\text{i.i.d.}}{\sim} \mathcal{N}(0, 1)$. Abusing terminology, we refer to $\gamma$ as the SNR of $Y$. Define also $\alpha \coloneqq 1/\delta \in (0, \infty)$ so that $d/n \to \alpha$. The scaling of the parameters in (4.1) turns out to be more convenient for presenting the results in this section. Results for $Y$ can be easily translated to $A$ by a change of variables.

Let $\widetilde{u}^* \coloneqq \Xi^{-1/2} u^*$ and $\widetilde{v}^* \coloneqq \Sigma^{-1/2} v^*$ denote the whitened signals. The main result of this section is Theorem 4.2, which characterizes the performance of the matrix minimum mean square error

---

[1] All our results hold verbatim if $\Xi, \Sigma$ are random matrices independent of each other and of $u^*, v^*, \widetilde{W}$.

(MMSE) associated to the estimation of $\widetilde{u}^*(\widetilde{v}^*)^\top, \widetilde{u}^*(\widetilde{u}^*)^\top$ and $\widetilde{v}^*(\widetilde{v}^*)^\top$, via the corresponding Bayes-optimal estimators:

$$\mathrm{MMSE}_n(\gamma) := \frac{1}{nd}\mathbb{E}\Big[\big\|\widetilde{u}^*(\widetilde{v}^*)^\top - \mathbb{E}\big[\widetilde{u}^*(\widetilde{v}^*)^\top \mid Y\big]\big\|_\mathrm{F}^2\Big], \tag{4.2}$$

$$\mathrm{MMSE}_n^u(\gamma) := \frac{1}{n^2}\mathbb{E}\Big[\big\|\widetilde{u}^*(\widetilde{u}^*)^\top - \mathbb{E}\big[\widetilde{u}(\widetilde{u}^*)^\top \mid Y\big]\big\|_\mathrm{F}^2\Big], \tag{4.3}$$

$$\mathrm{MMSE}_n^v(\gamma) := \frac{1}{d^2}\mathbb{E}\Big[\big\|\widetilde{v}^*(\widetilde{v}^*)^\top - \mathbb{E}\big[\widetilde{v}^*(\widetilde{v}^*)^\top \mid Y\big]\big\|_\mathrm{F}^2\Big]. \tag{4.4}$$

Our characterization involves a pair of parameters $(q_u^*, q_v^*) \in \mathbb{R}_{\geq 0}^2$ defined as the largest solution to

$$q_u = \mathbb{E}\left[\frac{\alpha\gamma q_v \overline{\Xi}^{-2}}{1 + \alpha\gamma q_v \overline{\Xi}^{-1}}\right], \qquad q_v = \mathbb{E}\left[\frac{\gamma q_u \overline{\Sigma}^{-2}}{1 + \gamma q_u \overline{\Sigma}^{-1}}\right]. \tag{4.5}$$

Here and throughout the paper, all expectations involving $\overline{\Xi}, \overline{\Sigma}$ are computed as integrals against the limiting spectral distributions of $\Xi, \Sigma$.

The proposition below, proved in Appendix A, justifies the existence of the solution to (4.5) and identifies when a non-trivial solution emerges.

**Proposition 4.1.** *The fixed point equation* (4.5) *always has a trivial solution* $(0,0)$. *There exists a non-trivial solution* $(q_u^*, q_v^*) \in \mathbb{R}_{>0}^2$ *if and only if*

$$\alpha\gamma^2 \mathbb{E}\big[\overline{\Sigma}^{-2}\big]\mathbb{E}\big[\overline{\Xi}^{-2}\big] > 1, \tag{4.6}$$

*in which case the non-trivial solution is unique.*

We are now ready to state our main result on the MMSE.

**Theorem 4.2.** *Assume* $P = Q = \mathcal{N}(0,1)$. *For almost every* $\gamma > 0$,

$$\lim_{n\to\infty}\mathrm{MMSE}_n(\gamma) = \mathbb{E}\big[\overline{\Xi}^{-1}\big]\mathbb{E}\big[\overline{\Sigma}^{-1}\big] - q_u^* q_v^*, \tag{4.7}$$

$$\lim_{n\to\infty}\mathrm{MMSE}_n^u(\gamma) = \mathbb{E}\big[\overline{\Xi}^{-1}\big]^2 - q_u^{*2}, \qquad \lim_{n\to\infty}\mathrm{MMSE}_n^v(\gamma) = \mathbb{E}\big[\overline{\Sigma}^{-1}\big]^2 - q_v^{*2}. \tag{4.8}$$

We note that

$$\lim_{n\to\infty}\frac{1}{nd}\mathbb{E}\Big[\big\|\widetilde{u}^*(\widetilde{v}^*)^\top\big\|_\mathrm{F}^2\Big] = \lim_{n\to\infty}\frac{1}{nd}\mathbb{E}\Big[\|\widetilde{u}^*\|_2^2\Big]\mathbb{E}\Big[\|\widetilde{v}^*\|_2^2\Big] = \mathbb{E}\big[\overline{\Xi}^{-1}\big]\mathbb{E}\big[\overline{\Sigma}^{-1}\big], \tag{4.9}$$

where the last step follows from Proposition G.2. This quantity represents the trivial error in the estimation of $\widetilde{u}^*(\widetilde{v}^*)^\top$, which is achieved by the all-0 estimator. Analogous considerations hold for $\widetilde{u}^*(\widetilde{u}^*)^\top$ and $\widetilde{v}^*(\widetilde{v}^*)^\top$, for which the trivial estimation error is $\mathbb{E}\big[\overline{\Xi}^{-1}\big]^2$ and $\mathbb{E}\big[\overline{\Sigma}^{-1}\big]^2$, respectively. Thus, Proposition 4.1 and Theorem 4.2 identify (4.6) as the condition for non-trivial estimation, and the smallest $\gamma$ that satisfies (4.6) gives the *weak recovery threshold*.

We show below that the weak recovery threshold is the same for the estimation of the true signals $u^*v^{*\top}, u^*u^{*\top}$ and $v^*v^{*\top}$. In this case, since the signal priors are Gaussian, using the same passages as in (4.9) one has that the trivial estimation error for $u^*v^{*\top}, u^*u^{*\top}$ and $v^*v^{*\top}$ is always equal to 1.

**Corollary 4.3.** *Assume* $P = Q = \mathcal{N}(0,1)$. *The MMSE associated to the estimation of* $u^*v^{*\top}$ *is non-trivial, i.e,*

$$\lim_{n\to\infty}\frac{1}{nd}\mathbb{E}\Big[\big\|u^*v^{*\top} - \mathbb{E}\big[u^*v^{*\top} \mid Y\big]\big\|_\mathrm{F}^2\Big] < 1 \tag{4.10}$$

*if and only if* (4.6) *holds. The same result holds for the MMSE of* $u^*u^{*\top}$ *and* $v^*v^{*\top}$.

**Proof strategy.** To derive the characterizations in Theorem 4.2, we write the posterior distribution of $u^*, v^*$ given $Y$ in a Gibbs form, i.e., its density is the exponential of a Hamiltonian normalized by a partition function. The interpolation argument relates the log-partition function (also referred to as the 'free energy') of the posterior to that of the posteriors of two Gaussian location models. Since i.i.d. Gaussianity is key to this approach, the challenge is to handle noise covariances. Our idea is

to incorporate the covariances into the priors. In terms of the Hamiltonian, the model is equivalent to the estimation of the whitened signals $\Xi^{-1/2}u^*, \Sigma^{-1/2}v^*$, whose priors have covariances, in the presence of i.i.d. Gaussian noise. We then manage to carry out the interpolation argument for the equivalent model and evaluate the free energy of the corresponding Gaussian location models.

Specifically, let us starting by writing down the expression of the posterior distribution after setting up some notation. For $u \in \mathbb{R}^n, v \in \mathbb{R}^d$, let $\widetilde{u} := \Xi^{-1/2}u, \widetilde{v} := \Sigma^{-1/2}v$. Define the densities

$$\mathrm{d}\widetilde{P}(\widetilde{u}) := \sqrt{\det(\Xi)}\,\mathrm{d}P^{\otimes n}(\Xi^{1/2}\widetilde{u}), \qquad \mathrm{d}\widetilde{Q}(\widetilde{v}) := \sqrt{\det(\Sigma)}\,\mathrm{d}Q^{\otimes d}(\Sigma^{1/2}\widetilde{v}),$$

where the determinant factors ensure that the integrals equal 1. With $P = Q = \mathcal{N}(0,1)$, we have $\widetilde{P} = \mathcal{N}(0_n, \Xi^{-1}), \widetilde{Q} = \mathcal{N}(0_d, \Sigma^{-1})$, and from Bayes' rule the posterior of $(u^*, v^*)$ given $Y$ is

$$\mathrm{d}P(u,v \,|\, Y) = \frac{1}{\mathcal{Z}_n(\gamma)}\exp\Big(\mathcal{H}_n(\Xi^{-1/2}u, \Sigma^{-1/2}v)\Big)\,\mathrm{d}P^{\otimes n}(u)\,\mathrm{d}Q^{\otimes d}(v), \qquad (4.11)$$

where the Hamiltonian and the partition function are given respectively by

$$\mathcal{H}_n(\widetilde{u}, \widetilde{v}) := \sqrt{\frac{\gamma}{n}}\widetilde{u}^\top Z\widetilde{v} + \frac{\gamma}{n}\widetilde{u}^\top\widetilde{u}^*\widetilde{v}^\top\widetilde{v}^* - \frac{\gamma}{2n}\|\widetilde{u}\|_2^2\|\widetilde{v}\|_2^2, \qquad (4.12)$$

$$\mathcal{Z}_n(\gamma) := \iint\exp\Big(\mathcal{H}_n(\Xi^{-1/2}u, \Sigma^{-1/2}v)\Big)\,\mathrm{d}P^{\otimes n}(u)\,\mathrm{d}Q^{\otimes d}(v) = \iint\exp(\mathcal{H}_n(\widetilde{u}, \widetilde{v}))\,\mathrm{d}\widetilde{P}(\widetilde{u})\,\mathrm{d}\widetilde{Q}(\widetilde{v}).$$
$$(4.13)$$

Define the free energy as

$$\mathcal{F}_n(\gamma) := \frac{1}{n}\mathbb{E}[\log\mathcal{Z}_n(\gamma)]. \qquad (4.14)$$

The major technical step is to characterize $\mathcal{F}_n(\gamma)$ in the large $n$ limit in terms of a bivariate functional $\mathcal{F}$ introduced below. This is the core component to derive the MMSE characterization.

For a positive random variable $\overline{\Sigma}$ subject to the conditions in Section 3, let

$$\psi_{\overline{\Sigma}}(\gamma) := \frac{1}{2}\Big(\gamma\mathbb{E}\Big[\overline{\Sigma}^{-1}\Big] - \mathbb{E}\Big[\log\Big(1 + \gamma\overline{\Sigma}^{-1}\Big)\Big]\Big). \qquad (4.15)$$

As shown in Appendix B, $\psi_{\overline{\Sigma}}(\gamma)$ is the limiting free energy of a Gaussian channel, in which one wishes to estimate $x^* \in \mathbb{R}^n$ from the observation $Y = \sqrt{\gamma}x^* + \Sigma^{1/2}Z$ corrupted by anisotropic Gaussian noise with covariance $\Sigma$. Using (4.15), let us define the replica symmetric potential $\mathcal{F}$:

$$\mathcal{F}(q_u, q_v) := \psi_{\overline{\Xi}}(\alpha\gamma q_v) + \alpha\psi_{\overline{\Sigma}}(\gamma q_u) - \frac{\alpha\gamma}{2}q_u q_v,$$

and the set of critical points of $\mathcal{F}$:

$$\begin{aligned}\mathcal{C}(\gamma, \alpha) &:= \big\{(q_u, q_v) \in \mathbb{R}_{\geq 0}^2 : \partial_1\mathcal{F}(q_u, q_v) = 0, \partial_2\mathcal{F}(q_u, q_v) = 0\big\} \\ &= \big\{(q_u, q_v) \in \mathbb{R}_{\geq 0}^2 : q_u = 2\psi_{\overline{\Xi}}'(\alpha\gamma q_v), q_v = 2\psi_{\overline{\Sigma}}'(\gamma q_u)\big\} \\ &= \big\{(q_u, q_v) \in \mathbb{R}_{\geq 0}^2 : (q_u, q_v)\text{ solves }(4.5)\big\},\end{aligned} \qquad (4.16)$$

where the last equality is a direct calculation of $\psi_{\overline{\Xi}}', \psi_{\overline{\Sigma}}'$. The following result, proved in Appendix C, shows that the limit of $\mathcal{F}_n(\gamma)$ is given by a dimension-free variational problem involving $\mathcal{F}(q_u, q_v)$.

**Theorem 4.4** (Free energy). *Assume $P = Q = \mathcal{N}(0,1)$. Then, we have*

$$\lim_{n\to\infty}\mathcal{F}_n(\gamma) = \sup_{q_v \geq 0}\inf_{q_u \geq 0}\mathcal{F}(q_u, q_v) = \sup_{(q_u, q_v) \in \mathcal{C}(\gamma, \alpha)}\mathcal{F}(q_u, q_v),$$

*and $\sup_{q_v}\inf_{q_u}$ and $\sup_{(q_u, q_v)}$ are achieved by the same $(q_u^*, q_v^*)$ in Proposition 4.1.*

*Remark* 4.1 (Equivalent models). Informally, the above result says that the matrix model (4.1) is equivalent at the level of Hamiltonian to the following two statistically uncorrelated vector models:

$$Y^u := \sqrt{\alpha\gamma q_v^*}u^* + \Xi^{1/2}Z^u \in \mathbb{R}^n, \quad Y^v := \sqrt{\gamma q_u^*}v^* + \Sigma^{1/2}Z^v \in \mathbb{R}^d, \qquad (4.17)$$

with $q_u^*, q_v^*$ the largest solution to (4.5) and $(u^*, v^*, Z^u, Z^v) \sim P^{\otimes n} \otimes Q^{\otimes d} \otimes \mathcal{N}(0_n, I_n) \otimes \mathcal{N}(0_d, I_d)$.

*Remark* 4.2 (Gaussian priors). Theorem 4.4 crucially relies on having Gaussian priors $P, Q$. This assumption is mainly used to derive single-letter (i.e., dimension-free) expressions of the free energy of the vector models in (4.17) which, under Gaussian priors, are nothing but Gaussian integrals. The free energy, and hence the MMSE, are expected to be sensitive to the priors. Indeed, this is already the case in the homoscedastic setting $\Xi = I_n, \Sigma = I_d$ [52]. An extension towards general i.i.d. priors is a challenging open problem and, in fact, without posing additional assumptions on $\Xi, \Sigma$, it is unclear whether a single-letter expression for free energy and MMSE is possible.

At this point, the MMSE can be derived from the above characterization of free energy. Indeed, let

$$\mathcal{D}(\alpha) := \{\gamma > 0 : \mathcal{F} \text{ has a unique maximizer } (q_u^*, q_v^*) \text{ over } \mathcal{C}(\gamma, \alpha)\}.$$

The envelope theorem [51, Corollary 4] ensures that $\mathcal{D}(\alpha)$ is equal to $\mathbb{R}_{>0}$ up to a countable set. Using algebraic relations between free energy and MMSE, we prove (4.7) and (4.8) for all $\gamma \in \mathcal{D}(\alpha)$ (and, thus, for almost every $\gamma > 0$). Then, using the Nishimori identity (Proposition G.4) and the fact that the ESDs of $\Xi, \Sigma$ are upper and lower bounded by constants independent of $n$ and $d$, Corollary 4.3 also follows. The formal arguments are contained in Appendix D.

## 5   Spectral estimator

This section introduces a spectral estimator that meets the weak recovery threshold and, for one-sided heteroscedasticity, attains the Bayes-optimal error. Suppose that the following condition holds

$$\frac{\lambda^4}{\delta}\mathbb{E}\left[\overline{\Sigma}^{-2}\right]\mathbb{E}\left[\overline{\Xi}^{-2}\right] > 1, \tag{5.1}$$

which is equivalent to (4.6). Under this condition, the fixed point equations (4.5) have a unique pair of positive solutions $(q_u^*, q_v^*)$. For convenience, we also define the rescalings $\mu^* := \lambda q_v^*/\delta, \nu^* := \lambda q_u^*$, and the auxiliary quantities

$$b^* := \frac{1}{\delta}\mathbb{E}\left[\frac{\lambda}{\lambda\nu^* + \overline{\Sigma}}\right], \qquad c^* := \mathbb{E}\left[\frac{\lambda}{\lambda\mu^* + \overline{\Xi}}\right]. \tag{5.2}$$

Now, we pre-process the data matrix $A$ as

$$A^* := \lambda(\lambda(\mu^* + b^*)I_n + \Xi)^{-1/2}\Xi^{-1/2}A\Sigma^{-1/2}(\lambda(\nu^* + c^*)I_d + \Sigma)^{-1/2}, \tag{5.3}$$

from which we obtain the spectral estimators

$$\widehat{u} := \eta_u\sqrt{n}\,\frac{\Xi^{1/2}(\lambda(\mu^* + b^*)I_n + \Xi)^{-1/2}(\lambda\mu^*I_n + \Xi)u_1(A^*)}{\left\|\Xi^{1/2}(\lambda(\mu^* + b^*)I_n + \Xi)^{-1/2}(\lambda\mu^*I_n + \Xi)u_1(A^*)\right\|_2}, \tag{5.4a}$$

$$\widehat{v} := \eta_v\sqrt{d}\,\frac{\Sigma^{1/2}(\lambda(\nu^* + c^*)I_d + \Sigma)^{-1/2}(\lambda\nu^*I_d + \Sigma)v_1(A^*)}{\left\|\Sigma^{1/2}(\lambda(\nu^* + c^*)I_d + \Sigma)^{-1/2}(\lambda\nu^*I_d + \Sigma)v_1(A^*)\right\|_2}, \tag{5.4b}$$

where $u_1(\cdot)/v_1(\cdot)$ denote the top left/right singular vectors and

$$\eta_u := \sqrt{\frac{\lambda\mu^*}{\lambda\mu^* + 1}}, \qquad \eta_v := \sqrt{\frac{\lambda\nu^*}{\lambda\nu^* + 1}}. \tag{5.5}$$

Note that $\eta_u, \eta_v > 0$, provided that (5.1) holds. The pre-processing of $A$ in (5.3) and the form of the spectral estimators in (5.4) come from the derivation of a suitable AMP algorithm, and they are discussed at the end of the section. We finally defer to Appendix E.3 the definition of the scalar quantity $\sigma_2^*$ obtained via a fixed point equation depending only on $\overline{\Xi}, \overline{\Sigma}, \lambda, \delta$, see (E.26) for details.

Our main result, Theorem 5.1, shows that, under the criticality condition (5.1), the matrix $A^*$ exhibits a spectral gap between the top two singular values, and it characterizes the performance of the spectral estimators in (5.4), proving that they achieve weak recovery of $u^*$ and $v^*$, respectively.

**Theorem 5.1.** *Suppose that* (5.1) *holds and that, for any $c > 0$,*

$$\lim_{\beta \downarrow s}\mathbb{E}\left[\frac{\overline{\Sigma}^*}{\beta - c\overline{\Sigma}^*}\right] = \lim_{\beta \downarrow s}\mathbb{E}\left[\left(\frac{\overline{\Sigma}^*}{\beta - c\overline{\Sigma}^*}\right)^2\right] = \infty, \qquad \lim_{\alpha \downarrow \sup \text{supp}(\overline{\Xi}^*)}\mathbb{E}\left[\frac{\overline{\Xi}^*}{\alpha - \overline{\Xi}^*}\right] = \infty, \tag{5.6}$$

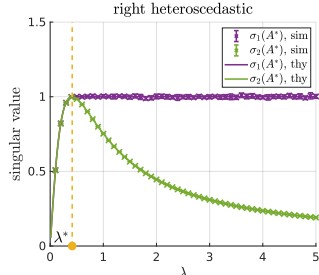

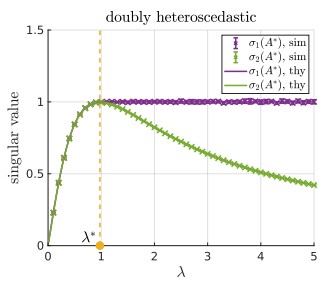

(a) $\Xi = I_n$ and $\Sigma$ a Toeplitz matrix with $\rho = 0.9$.

(b) $\Xi$ a circulant matrix with $c = 0.1, \ell = 5$ and $\Sigma$ a Toeplitz matrix with $\rho = 0.5$.

Figure 1: Top two singular values of $A^*$ in (5.3), where $d = 4000, \delta = 4$ and each simulation is averaged over 10 i.i.d. trials. The singular values computed experimentally ('sim' in the legends and $\times$ in the plots) closely match our theoretical prediction in (5.7) ('thy' in the legends and solid curves with the same color in the plots). The threshold $\lambda^*$ is such that equality holds in (5.1). We note that the green curve corresponding to $\sigma_2^*$ is smaller than 1 for $\lambda > \lambda^*$, i.e., when (5.1) holds.

*where* $\overline{\Xi}^* := \frac{\lambda}{\lambda(\mu^* + b^*) + \overline{\Xi}}$, $\overline{\Sigma}^* := \frac{\lambda}{\lambda(\nu^* + c^*) + \overline{\Sigma}}$ *and* $s := c \cdot \sup \mathrm{supp}(\overline{\Sigma}^*)$. *Let* $A^*, \widehat{u}, \widehat{v}, \sigma_2^*$ *be defined in* (5.3), (5.4) *and* (E.26), *and* $\sigma_i(A^*)$ *denote the* $i$-*th largest singular value of* $A^*$. *Then, if* $\sigma_2^* < 1$, *the following limits hold in probability:*

$$\lim_{n \to \infty} \sigma_1(A^*) = 1 > \sigma_2^* = \lim_{n \to \infty} \sigma_2(A^*), \tag{5.7}$$

$$\lim_{n \to \infty} \frac{|\langle \widehat{u}, u^* \rangle|}{\|\widehat{u}\|_2 \|u^*\|_2} = \eta_u, \quad \lim_{d \to \infty} \frac{|\langle \widehat{v}, v^* \rangle|}{\|\widehat{v}\|_2 \|v^*\|_2} = \eta_v \tag{5.8}$$

$$\lim_{n \to \infty} \frac{1}{n^2} \left\| u^* u^{*\top} - \widehat{u}\widehat{u}^\top \right\|_F^2 = 1 - \eta_u^4, \quad \lim_{d \to \infty} \frac{1}{d^2} \left\| v^* v^{*\top} - \widehat{v}\widehat{v}^\top \right\|_F^2 = 1 - \eta_v^4, \tag{5.9}$$

$$\lim_{n \to \infty} \frac{1}{nd} \left\| u^* v^{*\top} - \widehat{u}\widehat{v}^\top \right\|_F^2 = 1 - \eta_u^2 \eta_v^2. \tag{5.10}$$

*Remark* 5.1 (Assumptions). To guarantee a spectral gap for $A^*$ and the weak recoverability of $u^*, v^*$ via the proposed spectral method, we also require the algebraic condition $\sigma_2^* < 1$. We conjecture that this condition is implied by (5.1), and we have verified that this is the case in all our numerical experiments (see Figure 1 for two concrete examples). The additional assumption (5.6) is a mild regularity condition on the covariances. It ensures that the densities of $\overline{\Xi}^*, \overline{\Sigma}^*$ decay sufficiently slowly at the edges of the support, so that $\sigma_2^*$ is well-posed [79].

*Remark* 5.2 (Signal priors). Theorem 5.1 does not require the prior distributions $P, Q$ to be Gaussian, and it is valid for any i.i.d. prior with mean 0 and variance 1.

On the one hand, Corollary 4.3 shows that, if (5.1) is violated, the problem is information-theoretically impossible, i.e., no estimator achieves non-trivial error. On the other hand, Theorem 5.1 exhibits a pair of estimators that achieves non-trivial error as soon as (5.1) holds – under the additional assumption $\sigma_2^* < 1$ which we conjecture to be implied by (5.1). Thus, the spectral method in (5.4) is optimal in terms of weak recovery threshold. Though such estimators do not attain the optimal error, when both priors are Gaussian and $\Xi = I_n$, $\widehat{u}\widehat{u}^\top$ is the Bayes-optimal estimate for $u^* u^{*\top}$.

**Corollary 5.2.** *Assume* $P = Q = \mathcal{N}(0, 1)$, *and consider the setting of* Theorem 5.1 *with the additional assumption* $\Xi = I_n$. *Then,* $\eta_u = \sqrt{q_u^*}$, *i.e.,* $\widehat{u}\widehat{u}^\top$ *achieves the MMSE for* $u^* u^{*\top}$.

The claim readily follows by noting that, when $\Xi = I_n$, the first equation in (4.5) becomes

$$q_u^* = \frac{\alpha \gamma q_v^*}{1 + \alpha \gamma q_v^*} = \frac{(\lambda^2/\delta)(\delta \mu^*/\lambda)}{1 + (\lambda^2/\delta)(\delta \mu^*/\lambda)} = \frac{\lambda \mu^*}{1 + \lambda \mu^*} = \eta_u^2,$$

where the last equality is by the definition (5.5) of $\eta_u$. Let us highlight that, even if $\Xi = I_n$, $\widehat{u}$ still makes non-trivial use of the other covariance $\Sigma^{1/2}$. At the information-theoretic level, this is reflected by the fact that $\Sigma^{1/2}$ enters $q_u^*$ through the fixed point equations (4.5). Therefore, even though the matrix model in (4.1) is equivalent to a pair of uncorrelated vector models in (4.17) in the sense of the free energy, the tasks of estimating $u^*$ and $v^*$ cannot be decoupled.

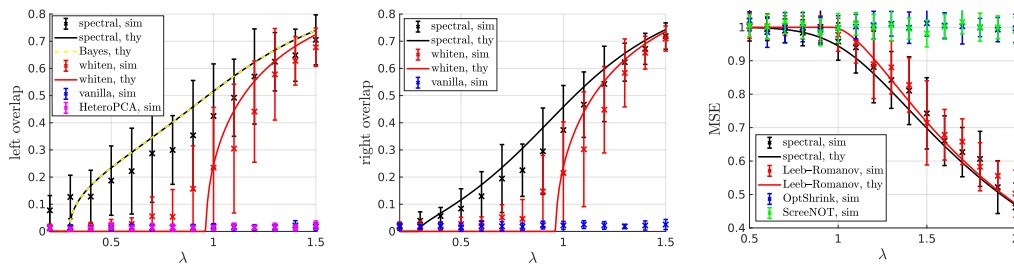

(a) Normalized correlation with $u^*$    (b) Normalized correlation with $v^*$    (c) Matrix MSE for $u^*v^{*\top}$

Figure 2: Performance comparison when $\Xi = I_n$ and $\Sigma$ is a circulant matrix. The numerical results closely follow the predictions of Theorem 5.1, and our spectral estimators in (5.4) outperform all other methods (Leeb–Romanov, OptShrink, ScreeNOT, and HeteroPCA), especially at low SNR.

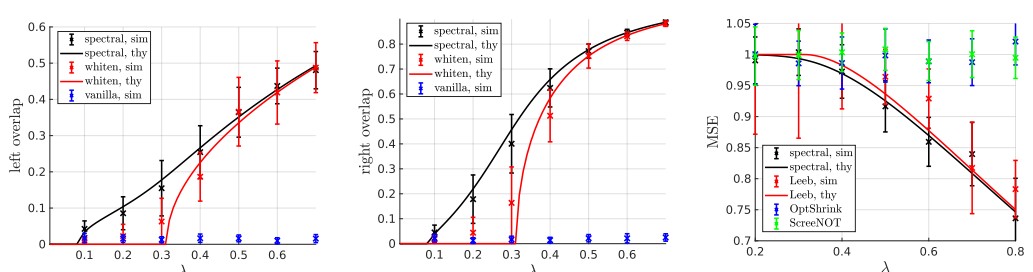

(a) Normalized correlation with $u^*$    (b) Normalized correlation with $v^*$    (c) Matrix MSE for $u^*v^{*\top}$

Figure 3: Performance comparison when $\Xi$ is a Toeplitz matrix and $\Sigma$ is circulant. The numerical results closely follow the predictions of Theorem 5.1, and our spectral estimators in (5.4) outperform all other methods (Leeb, OptShrink, and ScreeNOT), especially at low SNR.

**Numerical experiments.** Figures 2 and 3 demonstrate the advantage of our method over existing approaches, and they display an accurate agreement between simulations ('sim' in the legends and $\times$ in the plots) and the theoretical predictions of Theorem 5.1 ('thy' in the legends and solid curves with the same color in the plots), both plotted as a function of $\lambda$. In both figures, $n = 4000, d = 2000$ (so $\delta = 2$), and $P = Q = \mathcal{N}(0, 1)$. Each data point is computed from 20 i.i.d. trials and error bars are reported at 1 standard deviation. We let $\Xi$ be either the identity or a Toeplitz matrix [77, 41, 19], i.e., $\Xi_{i,j} = \rho^{|i-j|}$ with $\rho = 0.9$. We let $\Sigma$ be a circulant matrix [40, 39]: the first row has 1 in the first position, $c = 0.0078$ in the second through $(\ell + 1)$-st position and in the last $\ell$ positions ($\ell = 300$), with the remaining entries being 0; for $2 \leq i \leq d$, the $i$-th row is a cyclic shift of the $(i-1)$-st row to the right by 1 position. Both matrices satisfy (5.6) and the conditions of Section 3.

Our spectral estimator outperforms all other approaches: Leeb–Romanov [44], OptShrink [59], ScreeNOT [27], and HeteroPCA [76] in the one-sided heteroscedastic case (Figure 2); Leeb [45], OptShrink, and ScreeNOT in the doubly heteroscedastic case (Figure 3). When computing the normalized correlation with the signals (left/right overlap), the performance of Leeb–Romanov and Leeb is the same as the estimators $\Xi^{1/2}u_1(\Xi^{-1/2}A\Sigma^{-1/2}), \Sigma^{1/2}v_1(\Xi^{-1/2}A\Sigma^{-1/2})$ referred to as 'whiten' in Figures 2a and 2b; the performance of OptShrink and ScreeNOT is the same as the estimators $u_1(A), v_1(A)$ referred to as 'vanilla' in Figures 3a and 3b. The advantage of our approach (in black) is especially significant at low SNR; as SNR increases, Leeb-Romanov and Leeb (in red) achieve similar performance; a much larger SNR ($> 2$ and $> 3$ in Figures 2 and 3) is required by HeteroPCA, OptShrink and ScreeNOT (in magenta, blue and green) to perform comparably.

Finally, Figure 4 shows the presence of spectral outliers in $A^*$ and their absence in $A$ at a fixed $\lambda$.

**Proof strategy.** The design and analysis of the spectral estimator in (5.4) comprise two steps, detailed in Appendix E. The *first step* is to present an AMP algorithm dubbed Bayes-AMP for matrix denoising with doubly heteroscedastic noise. Specifically, its iterates are updated as

$$u^t = \Xi^{-1}A\Sigma^{-1}\widetilde{v}^t - b_t\Xi^{-1}\widetilde{u}^{t-1}, \quad \widetilde{u}^t = g_t^*(u^t), \quad c_t = \frac{1}{n}\text{Tr}((\nabla g_t^*(u^t))\Xi^{-1}), \quad (5.11)$$

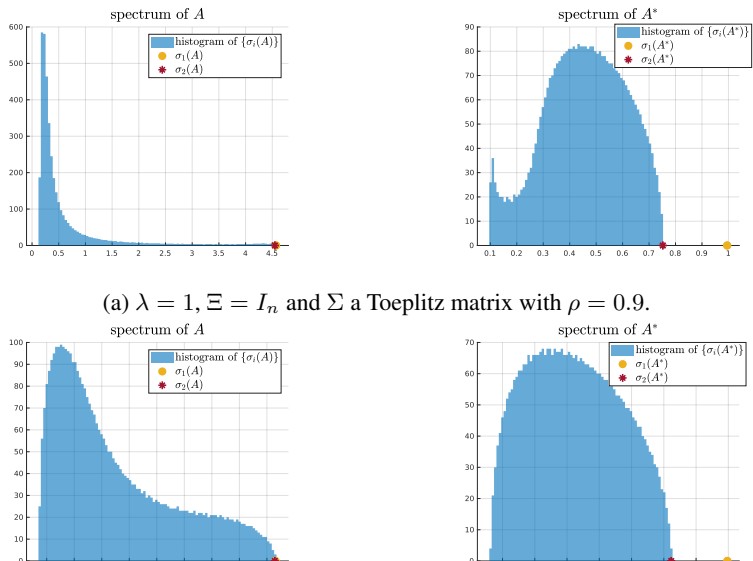

(a) $\lambda = 1, \Xi = I_n$ and $\Sigma$ a Toeplitz matrix with $\rho = 0.9$.

(b) $\lambda = 2, \Xi$ a circulant matrix with $c = 0.1, \ell = 5$ and $\Sigma$ a Toeplitz matrix with $\rho = 0.5$.

Figure 4: Spectra of $A$ and $A^*$ averaged over 10 i.i.d. trials, where $d = 4000, \delta = 4$. An outlier singular value emerges in the spectrum of $A^*$ due to the pre-processing on $A$.

$$v^{t+1} = \Sigma^{-1} A^\top \Xi^{-1} \widetilde{u}^t - c_t \Sigma^{-1} \widetilde{v}^t, \quad \widetilde{v}^{t+1} = f_{t+1}^*(v^{t+1}), \quad b_{t+1} = \frac{1}{n} \operatorname{Tr}((\nabla f_{t+1}^*(v^{t+1}))\Sigma^{-1}),$$

where $\nabla$ denotes the Jacobian matrix, and the functions $g_t^*, f_{t+1}^*$ are specified below in (5.12). As common in AMP algorithms, the iterates (5.11) are accompanied with a state evolution which accurately tracks their behavior via a simple deterministic recursion: the joint empirical distribution of $(u^*, v^*, u^t, v^{t+1})$ converges to the random variables $(U^*, V^*, U_t, V_{t+1})$, see Proposition E.1 for a formal statement and the recursive description of the laws of such random variables. Then, the name 'Bayes-AMP' is motivated by the fact that $g_t^*, f_{t+1}^*$ are the posterior-mean denoisers given by

$$g_t^*(u) := \mathbb{E}[U^* \mid U_t = u], \qquad f_{t+1}^*(v) := \mathbb{E}[V^* \mid V_{t+1} = v]. \tag{5.12}$$

Remarkably, Bayes-AMP operates on $\Xi^{-1} A \Sigma^{-1}$, as opposed to the widely adopted ansatz of considering the whitened matrix $\Xi^{-1/2} A \Sigma^{-1/2}$. The advantage of operating on $\Xi^{-1} A \Sigma^{-1}$ is that the fixed point of the corresponding state evolution matches the extremizers of the free energy in (4.5). This would *not* be the case if Bayes-AMP used the whitening $\Xi^{-1/2} A \Sigma^{-1/2}$. Indeed, one can repeat the analysis of an AMP that operates on $\Xi^{-1/2} A \Sigma^{-1/2}$. The fixed point equations of the resulting state evolution do not match the information-theoretically optimal one in (4.5). In particular, the weak recovery threshold coming out of this approach is strictly larger than the optimal one in (4.6), as long as at least one of $\Xi, \Sigma$ is not a multiple of the identity. Since these derivations led to suboptimal results, the details were left out from the paper.

The design of Bayes-AMP and the proof of its state evolution follow a two-step reduction detailed in Appendix F. Using a change of variables, we show in Appendix F.2 that Bayes-AMP can be realized by an auxiliary AMP with non-separable denoising functions (meaning that $g_t, f_{t+1}$ cannot be written as univariate functions applied component-wise) operating on $\Xi^{-1/2} A \Sigma^{-1/2} = \frac{\lambda}{n} \widetilde{u}^*(\widetilde{v}^*)^\top + \widetilde{W}$. Then, in Appendix F.1 we simulate the auxiliary AMP using a standard AMP operating on the i.i.d. Gaussian matrix $\widetilde{W}$, whose state evolution has been established in [13, 34].

However, Bayes-AMP by itself is not a practical algorithm since it needs a warm start, i.e., an initialization that achieves non-trivial error. Thus, the *second step* is to design a spectral estimator that solves the fixed point equation of Bayes-AMP, which turns out to be an eigen-equation for $A^*$.

To offer the readers an intuition on how the spectral estimators arise from Bayes-AMP, we now heuristically derive the form (5.3) of $A^*$ and the expression (5.4) of the spectral estimator. To do so, we note that the large-$n$ limits of $c_t, b_{t+1}$ coincide with the auxiliary quantities $c^*, b^*$ defined in (5.2).

Furthermore, when the priors of $u^*, v^*$ are Gaussian, (5.12) reduces to

$$g_t^*(u) = \lambda(\lambda\mu^*\Xi^{-1} + I_n)^{-1}u, \qquad f_{t+1}^*(v) = \lambda(\lambda\nu^*\Sigma^{-1} + I_d)^{-1}v,$$

where we recall that $\mu^* = \lambda q_v^*/\delta$ and $\nu^* = \lambda q_u^*$ are rescalings of the non-trivial solution $(q_u^*, q_v^*)$ of (4.5). Denoting by $u, v$ the fixed points of the iteration (5.11), after some manipulations we have

$$\mathfrak{g}(\Xi)u = A^*\mathfrak{f}(\Sigma)v, \qquad \mathfrak{f}(\Sigma)v = A^{*\top}\mathfrak{g}(\Xi)u,$$

where $A^*$ is given in (5.3) and

$$\mathfrak{g}(\Xi) := \sqrt{\lambda}(\lambda(\mu^* + b^*)I_n + \Xi)^{1/2}(\lambda\mu^*I_n + \Xi)^{-1}\Xi^{1/2},$$

$$\mathfrak{f}(\Sigma) := \sqrt{\lambda}(\lambda(\nu^* + c^*)I_d + \Sigma)^{1/2}(\lambda\nu^*I_d + \Sigma)^{-1}\Sigma^{1/2}.$$

This suggests that $A^*$ has top singular value equal to 1 and $(\mathfrak{g}(\Xi)u, \mathfrak{f}(\Sigma)v)$ are aligned with the corresponding singular vectors $(u_1(A^*), v_1(A^*))$. Moreover, state evolution implies that the distribution of the fixed point $(u, v)$ is close to that of

$$(\mu^*\Xi^{-1}u^* + \sqrt{\mu^*/\lambda}w_u, \nu^*\Sigma^{-1}v^* + \sqrt{\nu^*/\lambda}w_v),$$

with $(w_u, w_v) \sim \mathcal{N}(0_n, \Xi^{-1}) \otimes \mathcal{N}(0_d, \Sigma^{-1})$ independent of $u^*, v^*$. Thus, to obtain estimates of $(u^*, v^*)$, we take $(\Xi\mathfrak{g}(\Xi)^{-1}u_1(A^*), \Sigma\mathfrak{f}(\Sigma)^{-1}v_1(A^*))$ and suitably rescale their norm, which leads to the expressions in (5.4). More details on the above heuristics are discussed in Appendix E.2.

The most outstanding step remains to make the heuristics rigorous. This involves proving that $\Xi u^t, \Sigma v^{t+1}$ are aligned with the proposed spectral estimator, which allows for a performance characterization via state evolution. The formal argument is carried out in Appendix E.4.

# 6   Concluding remarks

In this work, we establish information-theoretic limits and propose an efficient spectral method with optimality guarantees, for matrix estimation with doubly heteroscedastic noise. On the one hand, under Gaussian priors, we give a rigorous characterization of the MMSE; on the other hand, we present a spectral estimator that *(i)* achieves the information-theoretic weak recovery threshold, and *(ii)* is Bayes-optimal for the estimation of one of the signals, when the noise is heteroscedastic only on the other side. While our analysis focuses on rank-1 estimation, we expect that all results admit proper extensions to rank-$r$ signals, where $r$ is a constant independent on $n, d$.

The design and analysis of the spectral estimator draws connections with approximate message passing and, along the way, we introduce a Bayes-AMP algorithm which could be of independent interest. In this paper, we employ Bayes-AMP solely as a proof technique. However, one could use the spectral method designed here as an initialization of Bayes-AMP itself, after suitably correcting its iterates. This strategy has been successfully carried out for i.i.d. Gaussian noise in [56] and for rotationally invariant noise in [55, 81]. Bayes-AMP is well equipped to exploit signal priors more informative than the Gaussian one, and AMP algorithms are known to achieve the information-theoretically optimal estimation error for low-rank matrix inference [56, 6]. Nevertheless, we point out two obstacles towards doing so in the presence of doubly heteroscedastic noise. First, for general priors, establishing the information-theoretic limits remains a challenging open problem, and it is unclear whether a low-dimensional characterization of the free energy (and, hence, of the MMSE) is possible. Second, even for Gaussian priors, Bayes-AMP reduces to the proposed spectral estimator, which is not Bayes-optimal for the general case of doubly heteroscedastic noise.

Finally, the proposed spectral estimator makes non-trivial use of the covariances $\Xi, \Sigma$, which are assumed to be known. When such matrices possess additional structure – e.g., they are sparse [21], their inverses are sparse [19] or they are circulant or Toeplitz [75] – their consistent estimation is possible, see also the survey [20]. However, in general, $\Xi, \Sigma$ cannot be consistently estimated from the data when $n$ and $d$ grow proportionally. Thus, a challenging open problem is to construct estimators that retain comparable performance without knowing the noise covariances. The paper [33] addresses the challenge of unknown covariances by considering a modified model where one additionally observes an independent copy of noise. The statistician can then estimate the covariance from the noise-only observation and use it as a surrogate of the true covariance for estimating the signals from the spiked model. It is possible to derive similar results in the doubly heteroskedastic setting considered in our paper. If the covariances are completely unknown, then our model (with Gaussian priors) is equivalent to a spiked matrix model with a certain bi-rotationally invariant noise. This problem is expected to exhibit rather different behaviors than when covariances are known, see [7, 29] for recent progress on understanding the statistical and computational limits for such models.

## Acknowledgments and Disclosure of Funding

YZ thanks Shashank Vatedka for discussions at the early stage of this project. MM thanks Jean Barbier for sharing his insights into the interpolation argument. This research is partially supported by the 2019 Lopez-Loreta Prize and by the Interdisciplinary Projects Committee (IPC) at the Institute of Science and Technology Austria (ISTA). This work was done in part while the authors were visiting the Simons Institute for the Theory of Computing.

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

**Notation.** All vectors are column vectors. The singular values of a matrix $A \in \mathbb{R}^{n \times d}$ (where $n \geq d$ without loss of generality) are denoted by $\sigma_1(A) \geq \cdots \geq \sigma_d(A) \geq 0$ and the corresponding left/right singular vectors are denoted by $u_1(A), \cdots, u_d(A) \in \mathbb{S}^{n-1}$ and $v_1(A), \cdots, v_d(A) \in \mathbb{S}^{d-1}$. The (real) eigenvalues of a symmetric matrix $B \in \mathbb{R}^{d \times d}$ are denoted by $\lambda_1(B) \geq \cdots \geq \lambda_d(B)$ and the corresponding eigenvectors are denoted by $v_1(B), \cdots, v_d(B) \in \mathbb{S}^{d-1}$ (which will not be confused with the right singular vectors, whenever they are different, since we will never talk about both simultaneously for a square asymmetric matrix). We generally put overlines on capital letters to indicate a scalar random variable, e.g., $\overline{X} \in \mathbb{R}$, whose support is denoted by $\mathrm{supp}(\overline{X})$. The limit/liminf/limsup in probability are denoted by p-lim, p-liminf, p-limsup. The product distribution whose $i$-th ($i \in [k]$) marginal is given by $P_i$ is denoted by $P_1 \otimes \cdots \otimes P_k$, with the shorthand $P^{\otimes k}$ when all $P_i$'s are equal to $P$. The gradient of $f \colon \mathbb{R}^n \to \mathbb{R}$, or with abuse of notation, the Jacobian matrix of $F \colon \mathbb{R}^n \to \mathbb{R}^d$ are denoted by $\nabla f \in \mathbb{R}^n, \nabla F \in \mathbb{R}^{d \times n}$. The partial derivative of $f(x_1, \cdots, x_n)$ with respect to $x_i$ is denoted by either $\frac{\partial}{\partial x_i} f(x_1, \cdots, x_n)$ or $\partial_i f(x_1, \cdots, x_n)$. All log and exp are to the base $e$. We use the standard notation of $\sup(S), \inf(S)$ for a subset $S \subset \mathbb{R}$. We generally use $C > 0$ to denote a sufficiently large constant independent of $n, d$. Its dependence on other parameters will be specified, though its value may change across passages. We use the standard big O notation.

## A  Proof of Proposition 4.1

We eliminate $q_u$ and write a fixed point equation only involving $q_v$:

$$q_v = \mathbb{E}\left[ \frac{\gamma \mathbb{E}\left[ \frac{\alpha \gamma q_v \overline{\Xi}^{-2}}{\alpha \gamma q_v \overline{\Xi}^{-1} + 1} \right] \overline{\Sigma}^{-2}}{\gamma \mathbb{E}\left[ \frac{\alpha \gamma q_v \overline{\Xi}^{-2}}{\alpha \gamma q_v \overline{\Xi}^{-1} + 1} \right] \overline{\Sigma}^{-1} + 1} \right].$$

Denote the RHS by $f(q_v)$. Recall that we are only interested in non-negative solutions $(q_u, q_v)$. So let us restrict attention on $f$ to the domain $\mathbb{R}_{\geq 0}$. We have $f(0) = 0$ and

$$f'(q_u) = \mathbb{E}\left[ \frac{\gamma \overline{\Sigma}^{-2}}{\left( \gamma \mathbb{E}\left[ \frac{\alpha \gamma q_v \overline{\Xi}^{-2}}{\alpha \gamma q_v \overline{\Xi}^{-1} + 1} \right] \overline{\Sigma}^{-1} + 1 \right)^2} \right] \mathbb{E}\left[ \frac{\alpha \gamma \overline{\Xi}^{-2}}{\left( \alpha \gamma q_v \overline{\Xi}^{-1} + 1 \right)^2} \right] > 0,$$

$$f'(0) = \alpha \gamma^2 \mathbb{E}\left[ \overline{\Sigma}^{-2} \right] \mathbb{E}\left[ \overline{\Xi}^{-2} \right],$$

$$f''(q_v) = -2 \left( \mathbb{E}\left[ \frac{\alpha \gamma \overline{\Xi}^{-2}}{\left( \alpha \gamma q_v \overline{\Xi}^{-1} + 1 \right)^2} \right]^2 \mathbb{E}\left[ \frac{\gamma^2 \overline{\Sigma}^{-3}}{\left( \gamma \mathbb{E}\left[ \frac{\alpha \gamma q_v \overline{\Xi}^{-2}}{\alpha \gamma q_v \overline{\Xi}^{-1} + 1} \right] \overline{\Sigma}^{-1} + 1 \right)^3} \right] \right.$$

$$\left. + \mathbb{E}\left[ \frac{\gamma \overline{\Sigma}^{-2}}{\left( \gamma \mathbb{E}\left[ \frac{\alpha \gamma q_v \overline{\Xi}^{-2}}{\alpha \gamma q_v \overline{\Xi}^{-1} + 1} \right] \overline{\Sigma}^{-1} + 1 \right)^2} \right] \mathbb{E}\left[ \frac{\alpha^2 \gamma^2 \overline{\Xi}^{-3}}{\left( \alpha \gamma q_v \overline{\Xi}^{-1} + 1 \right)^3} \right] \right) < 0,$$

$$\lim_{q_v \to \infty} f(q_v) = \mathbb{E}\left[ \frac{\gamma \mathbb{E}\left[ \overline{\Xi}^{-1} \right] \overline{\Sigma}^{-2}}{\gamma \mathbb{E}\left[ \overline{\Xi}^{-1} \right] \overline{\Sigma}^{-1} + 1} \right] \in (0, \infty).$$

It then becomes evident that a non-trivial fixed point $q_v > 0$ exists if and only if $f'(0) > 1$ and in this case, the non-trivial fixed point is unique.

Finally, by the first equation in (4.5), there is a non-trivial fixed point $q_u$ if and only if there is a non-trivial fixed point $q_v$, which completes the proof.

## B  Auxiliary Gaussian channel

We formally introduce here the auxiliary model mentioned in Section 4. Consider a Gaussian channel with blocklength $n$, input $x^*$, output $Y$, anisotropic Gaussian noise $\Sigma^{1/2} Z$ and SNR $\gamma$:

$$Y = \sqrt{\gamma} x^* + \Sigma^{1/2} Z \in \mathbb{R}^n, \tag{B.1}$$

where

$$(x^*, Z) \sim P^{\otimes n} \otimes \mathcal{N}(0_n, I_n).$$

By similar derivations as in Section 4, the posterior distribution of $x^*$ given $Y$ can be written as

$$dP(x \mid Y) = \frac{1}{Z_n(\gamma)} \exp(H_n(x)) \, dP^{\otimes n}(x),$$

where the Hamiltonian and the partition function are

$$H_n(x) := \gamma x^{*\top} \Sigma^{-1} x + \sqrt{\gamma} Z^\top \Sigma^{-1/2} x - \frac{\gamma}{2} x^\top \Sigma^{-1} x,$$

$$Z_n(\gamma) := \int_{\mathbb{R}^n} \exp(H_n(x)) \, dP^{\otimes n}(x).$$

Define the free energy as

$$F_n(\gamma) := \frac{1}{n} \mathbb{E}[\log Z_n(\gamma)].$$

With $P = \mathcal{N}(0, 1)$, $Z_n(\gamma)$ becomes a Gaussian integral that can be computed as below using Proposition G.1:

$$Z_n(\gamma) = \frac{1}{\sqrt{\det(\gamma \Sigma^{-1} + I_n)}} \exp\left( \frac{1}{2} \left( \gamma \Sigma^{-1} x^* + \sqrt{\gamma} \Sigma^{-1/2} Z \right)^\top \left( \gamma \Sigma^{-1} + I_n \right)^{-1} \left( \gamma \Sigma^{-1} x^* + \sqrt{\gamma} \Sigma^{-1/2} Z \right) \right).$$

Therefore, by Proposition G.2,

$$\operatorname*{p\text{-}lim}_{n \to \infty} F_n(\gamma) = -\frac{1}{2} \mathbb{E}\left[ \log\left( \gamma \overline{\Sigma}^{-1} + 1 \right) \right] + \frac{1}{2} \gamma^2 \mathbb{E}\left[ \overline{\Sigma}^{-2} \left( \gamma \overline{\Sigma}^{-1} + 1 \right)^{-1} \right] + \frac{1}{2} \gamma \mathbb{E}\left[ \overline{\Sigma}^{-1} \left( \gamma \overline{\Sigma}^{-1} + 1 \right)^{-1} \right]$$

$$= \frac{1}{2} \left( \gamma \mathbb{E}\left[ \overline{\Sigma}^{-1} \right] - \mathbb{E}\left[ \log\left( 1 + \gamma \overline{\Sigma}^{-1} \right) \right] \right). \tag{B.2}$$

The above functional is nothing but $\psi_{\overline{\Sigma}}(\gamma)$ introduced in (4.15) which will play an important role in characterizing the free energy of the original model (4.1).

## C  Proof of Theorem 4.4

Before diving into the proof, we make further notation adjustments for the ease of applying the interpolation argument. Specifically, we will henceforth assume $\gamma = 1$ by incorporating the actual value of $\gamma$ into the prior distributions $P, Q$,

$$\int_{\mathbb{R}} x^2 \, dP(x) = \gamma, \quad \int_{\mathbb{R}} x^2 \, dQ(x) = 1.$$

This is obviously equivalent to the previous setting. So we can drop the dependence on $\gamma$ and write $\mathrm{MMSE}_n, \mathcal{Z}_n, \mathcal{F}_n$ for $\mathrm{MMSE}_n(\gamma), \mathcal{Z}_n(\gamma), \mathcal{F}_n(\gamma)$ defined in (4.2), (4.13) and (4.14).

We will also assume that $\Xi, \Sigma$ are diagonal. This is without loss of generality since the Gaussianity of $P, Q, \widetilde{W}$ ensures that both the prior distributions and the noise matrix are rotationally invariant. Furthermore, we truncate $P, Q$ so that they are supported on $[-K, K]$ for a constant $K > 0$. The approximation error in the free energy due to truncation can be made arbitrarily small if $K$ is sufficiently large, since the free energy is pseudo-Lipschitz in the prior distribution with respect to the Wasserstein-2 metric.

The proof follows an interpolation argument [10, 52, 53] with suitable modifications to take care of the noise heteroscedasticity featured by the covariances $\Xi, \Sigma$. To start with, define the interpolating models:

$$Y_t := \sqrt{\frac{1-t}{n}} u^* v^{*\top} + \Xi^{1/2} Z \Sigma^{1/2} \in \mathbb{R}^{n \times d},$$

$$Y_t^u := \sqrt{\alpha q_1(t)} u^* + \Xi^{1/2} Z^u \in \mathbb{R}^n,$$

$$Y_t^v := \sqrt{q_2(t)}v^* + \Sigma^{1/2}Z^v \in \mathbb{R}^d,$$

where $q_1(t), q_2(t) \geq 0$ are to be determined and

$$(u^*, v^*, Z, Z^u, Z^v) \sim P^{\otimes n} \otimes Q^{\otimes d} \otimes \mathcal{N}(0_{nd}, I_{nd}) \otimes \mathcal{N}(0_n, I_n) \otimes \mathcal{N}(0_d, I_d). \qquad \text{(C.1)}$$

By definition, $Y_0 = Y$ is the model that we would like to understand, and $Y_t^u, Y_t^v$ are instances of Gaussian channels in (B.1) whose free energy we have already understood (see (B.2)). The idea is that $Y_t$ serves as a path parametrized by $t \in [0,1]$ from the original model $Y$ to the target models $(Y_1^u, Y_1^v)$. The crux of the interpolation argument lies in showing that $Y_t$ and $(Y_t^u, Y_t^v)$ are equivalent (at the level of free energy) along the path.

To study the interpolating models $(Y_t, Y_t^u, Y_t^v)$, define the Hamiltonian

$$
\begin{aligned}
\mathcal{H}_{n,t}(\widetilde{u}, \widetilde{v}; q_1, q_2) &:= \sqrt{\frac{1-t}{n}}\widetilde{u}^\top Z \widetilde{v} + \frac{1-t}{n}\widetilde{u}^\top \widetilde{u}^* \widetilde{v}^\top \widetilde{v}^* - \frac{1-t}{2n}\|\widetilde{u}\|_2^2 \|\widetilde{v}\|_2^2 \\
&\quad + \alpha q_1 \widetilde{u}^\top \widetilde{u}^* + \sqrt{\alpha q_1}\widetilde{u}^\top Z^u - \frac{\alpha q_1}{2}\|\widetilde{u}\|_2^2 \\
&\quad + q_2 \widetilde{v}^\top \widetilde{v}^* + \sqrt{q_2}\widetilde{v}^\top Z^v - \frac{q_2}{2}\|\widetilde{v}\|_2^2.
\end{aligned}
\qquad \text{(C.2)}
$$

Then the posterior distribution of $(u^*, v^*)$ given $(Y_t, Y_t^u, Y_t^v)$ is

$$dP(u, v \,|\, Y_t, Y_t^u, Y_t^v) = \frac{1}{\mathcal{Z}_{n,t}} \exp\Big(\mathcal{H}_{n,t}(\Xi^{-1/2}u, \Sigma^{-1/2}v; q_1(t), q_2(t))\Big) dP^{\otimes n}(u) \, dQ^{\otimes d}(v). \qquad \text{(C.3)}$$

Let the partition function be

$$
\begin{aligned}
\mathcal{Z}_{n,t} &:= \int_{\mathbb{R}^d} \int_{\mathbb{R}^n} \exp\Big(\mathcal{H}_{n,t}(\Xi^{-1/2}u, \Sigma^{-1/2}v; q_1(t), q_2(t))\Big) dP^{\otimes n}(u) \, dQ^{\otimes d}(v) \\
&= \int_{\mathbb{R}^d} \int_{\mathbb{R}^n} \exp(\mathcal{H}_{n,t}(\widetilde{u}, \widetilde{v}; q_1(t), q_2(t))) \, d\widetilde{P}(\widetilde{u}) \, d\widetilde{Q}(\widetilde{v}).
\end{aligned}
\qquad \text{(C.4)}
$$

Define the free energy as

$$f_n(t) := \frac{1}{n}\mathbb{E}[\log \mathcal{Z}_{n,t}]. \qquad \text{(C.5)}$$

The Gibbs bracket $\langle \cdot \rangle_{n,t}$ denotes the expectation with respect to the posterior distribution in (C.3):

$$\langle g(\widetilde{u}, \widetilde{v}) \rangle_{n,t} := \frac{1}{\mathcal{Z}_{n,t}} \int_{\mathbb{R}^d} \int_{\mathbb{R}^n} g(\widetilde{u}, \widetilde{v}) \exp(\mathcal{H}_{n,t}(\widetilde{u}, \widetilde{v}; q_1(t), q_2(t))) \, d\widetilde{P}(\widetilde{u}) \, d\widetilde{Q}(\widetilde{v}), \qquad \text{(C.6)}$$

for any $g \colon \mathbb{R}^n \times \mathbb{R}^d \to \mathbb{R}$ such that the expectation exists. That is,

$$\langle g(\widetilde{u}, \widetilde{v}) \rangle_{n,t} = \mathbb{E}[g(\widetilde{u}^*, \widetilde{v}^*) \,|\, Y_t, Y_t^u, Y_t^v],$$

where we recall the notation

$$\widetilde{u}^* := \Xi^{-1/2}u^*, \qquad \widetilde{v}^* := \Sigma^{-1/2}v^*. \qquad \text{(C.7)}$$

We will also use the notation $\langle \cdot \rangle_n$ for the Gibbs bracket with respect to the original posterior $dP(u, v \,|\, Y)$ in (4.11).

**Lemma C.1.** *Consider $f_n(t)$ defined in (C.5) with $t \in \{0, 1\}$. Assume that $q_1(0), q_2(0)$ satisfy*

$$q_1(0) \geq 0, \quad q_2(0) \geq 0, \quad \lim_{n \to \infty} q_1(0) = \lim_{n \to \infty} q_2(0) = 0.$$

*Then we have*

$$f_n(0) = \mathcal{F}_n + \mathcal{O}(q_1(0) + q_2(0)), \qquad \text{(C.8)}$$

$$\lim_{n \to \infty} f_n(1) = \psi_{\overline{\Xi}}(\alpha q_1(1)\gamma) + \alpha \psi_{\overline{\Sigma}}(q_2(1)) + o_K, \qquad \text{(C.9)}$$

*where $\lim_{K \to \infty} o_K = 0$.*

*Proof.* To show the first statement (C.8), let us control $f_n(0) - \mathcal{F}_n$. Denoting

$$\mathcal{H}'_{n,t}(\widetilde{u}, \widetilde{v}; q_1, q_2) := \alpha q_1 \widetilde{u}^\top \widetilde{u}^* + \sqrt{\alpha q_1} \widetilde{u}^\top Z^u - \frac{\alpha q_1}{2} \|\widetilde{u}\|_2^2 + q_2 \widetilde{v}^\top \widetilde{v}^* + \sqrt{q_2} \widetilde{v}^\top Z^v - \frac{q_2}{2} \|\widetilde{v}\|_2^2$$

and recalling the Gibbs bracket notation $\langle \cdot \rangle_n$, we have

$$f_n(0) - \mathcal{F}_n = \frac{1}{n} \mathbb{E}\left[ \log \frac{\mathcal{Z}_{n,0}}{\mathcal{Z}_n} \right] = \frac{1}{n} \mathbb{E}\left[ \log \langle \exp\big(\mathcal{H}'_{n,0}(\widetilde{u}, \widetilde{v}; q_1(0), q_2(0))\big) \rangle_n \right], \tag{C.10}$$

where the outer expectation is over all randomness in $u^*, v^*, Z^u, Z^v$. The second equality above follows since

$$\mathcal{H}_{n,0}(\widetilde{u}, \widetilde{v}; q_1(0), q_2(0)) = \mathcal{H}_n(\widetilde{u}, \widetilde{v}) + \mathcal{H}'_{n,0}(\widetilde{u}, \widetilde{v}; q_1(0), q_2(0)).$$

We will derive double-sided bounds on $f_n(0) - \mathcal{F}_n$.

To upper bound it, use Jensen's inequality $\mathbb{E}[\log(\cdot)] \le \log \mathbb{E}[\cdot]$ on the partial expectation over $Z^u, Z^v$ in (C.10):

$$f_n(0) - \mathcal{F}_n \le \frac{1}{n} \mathop{\mathbb{E}}_{u^*, v^*} \left[ \log \mathop{\mathbb{E}}_{Z^u, Z^v} \left[ \langle \exp\big(\mathcal{H}'_{n,0}(\widetilde{u}, \widetilde{v}; q_1(0), q_2(0))\big) \rangle_n \right] \right]$$
$$= \frac{1}{n} \mathop{\mathbb{E}}_{u^*, v^*} \left[ \log \Big\langle \mathop{\mathbb{E}}_{Z^u, Z^v} \left[ \exp\big(\mathcal{H}'_{n,0}(\widetilde{u}, \widetilde{v}; q_1(0), q_2(0))\big) \right] \Big\rangle_n \right],$$

where the equality is legit since $\langle \cdot \rangle_n$ does not depend on $Z^u, Z^v$. By the Gaussian integral formula (Proposition G.1), the inner expectation equals

$$\mathop{\mathbb{E}}_{Z^u, Z^v} \left[ \exp\big(\mathcal{H}'_{n,0}(\widetilde{u}, \widetilde{v}; q_1(0), q_2(0))\big) \right] = \exp\big(\alpha q_1(0) \widetilde{u}^\top \widetilde{u}^* + q_2(0) \widetilde{v}^\top \widetilde{v}^*\big).$$

Replacing the Gibbs bracket with $\max$, we obtain an upper bound:

$$f_n(0) - \mathcal{F}_n \le \frac{1}{n} \mathop{\mathbb{E}}_{u^*, v^*} \left[ \log \max_{(\widetilde{u}, \widetilde{v}) \in \Xi^{-1/2}[-K,K]^n \times \Sigma^{-1/2}[-K,K]^d} \exp\big(\alpha q_1(0) \widetilde{u}^\top \widetilde{u}^* + q_2(0) \widetilde{v}^\top \widetilde{v}^*\big) \right]$$
$$\le \alpha q_1(0) \|\Xi\|_2^{-1} K^2 + q_2(0) \|\Sigma\|_2^{-1} K^2 \frac{d}{n}$$
$$\le \frac{2\alpha q_1(0) K^2}{\inf \operatorname{supp}(\overline{\Xi})} + \frac{2\alpha q_2(0) K^2}{\inf \operatorname{supp}(\overline{\Sigma})}, \tag{C.11}$$

where the last inequality holds for all sufficiently large $n$ by (3.3) and $n \asymp d$.

To lower bound $f_n - \mathcal{F}_n$, we use Jensen's inequality again but this time on the Gibbs bracket in (C.10):

$$f_n(0) - \mathcal{F}_n \ge \frac{1}{n} \mathbb{E}\left[ \langle \mathcal{H}'_{n,0}(\widetilde{u}, \widetilde{v}; q_1(0), q_2(0)) \rangle_n \right] = \frac{1}{n} \mathop{\mathbb{E}}_{u^*, v^*} \left[ \Big\langle \mathop{\mathbb{E}}_{Z^u, Z^v} \left[ \mathcal{H}'_{n,0}(\widetilde{u}, \widetilde{v}; q_1(0), q_2(0)) \right] \Big\rangle_n \right]. \tag{C.12}$$

Since $(Z^u, Z^v) \sim \mathcal{N}(0_n, I_n) \otimes \mathcal{N}(0_d, I_d)$, the inner expectation equals

$$\mathop{\mathbb{E}}_{Z^u, Z^v} \left[ \mathcal{H}'_{n,0}(\widetilde{u}, \widetilde{v}; q_1(0), q_2(0)) \right] = \alpha q_1(0) \widetilde{u}^\top \widetilde{u}^* - \frac{\alpha q_1(0)}{2} \|\widetilde{u}\|_2^2 + q_2(0) \widetilde{v}^\top \widetilde{v}^* - \frac{q_2(0)}{2} \|\widetilde{v}\|_2^2.$$

So

$$\max_{(\widetilde{u}, \widetilde{v}) \in \Xi^{-1/2}[-K,K]^n \times \Sigma^{-1/2}[-K,K]^d} \left| \mathop{\mathbb{E}}_{Z^u, Z^v} \left[ \mathcal{H}'_{n,0}(\widetilde{u}, \widetilde{v}; q_1(0), q_2(0)) \right] \right|$$
$$\le \frac{3}{2} \alpha q_1(0) \|\Xi\|_2^{-1} n K^2 + \frac{3}{2} q_2(0) \|\Sigma\|_2^{-1} d K^2.$$

Using this, we obtain a lower bound on $f_n - \mathcal{F}_n$ by replacing the Gibbs bracket on the RHS of (C.12) with $-\max|\cdot|$:

$$f_n(0) - \mathcal{F}_n \ge -\frac{1}{n} \mathop{\mathbb{E}}_{u^*, v^*} \left[ \max_{(\widetilde{u}, \widetilde{v}) \in \Xi^{-1/2}[-K,K]^n \times \Sigma^{-1/2}[-K,K]^d} \left| \mathop{\mathbb{E}}_{Z^u, Z^v} \left[ \mathcal{H}'_{n,0}(\widetilde{u}, \widetilde{v}; q_1(0), q_2(0)) \right] \right| \right]$$

$$\geq -\frac{3}{2}\alpha q_1(0)\|\Xi\|_2^{-1}K^2 - \frac{3}{2}q_2(0)\|\Sigma\|_2^{-1}K^2\frac{d}{n}$$

$$\geq -\frac{3\alpha q_1(0)K^2}{\inf \mathrm{supp}(\overline{\Xi})} - \frac{3\alpha q_2(0)K^2}{\inf \mathrm{supp}(\overline{\Sigma})}, \tag{C.13}$$

where the last inequality holds for all sufficiently large $n$ by (3.3) and $n \asymp d$. Combining (C.11) and (C.13) gives the first result (C.8).

We then prove the second statement (C.9). Since $f_n$ is pseudo-Lipschitz as a function of the priors, up to a term $o_K$ that vanishes as $K \to \infty$ uniformly over $n$, it suffices to ignore the truncation at $K$ and assume $P = \mathcal{N}(0, \gamma), Q = \mathcal{N}(0, 1)$. From the definition (C.4) of $\mathcal{Z}_{n,t}$, we have

$$\mathcal{Z}_{n,1} = \int_{\mathbb{R}^d}\int_{\mathbb{R}^n} \exp\Big(\mathcal{H}'_{n,1}(\Xi^{-1/2}u, \Sigma^{-1/2}v; q_1(1), q_2(1))\Big)\,\mathrm{d}P^{\otimes n}(u)\,\mathrm{d}Q^{\otimes d}(v)$$

$$= \sqrt{\frac{\det(\Xi)}{\gamma^n \det(\Xi/\gamma + \alpha q_1(1)I_n)}}$$

$$\times \exp\left(\frac{1}{2}\Big(\alpha q_1(1)\widetilde{u}^* + \sqrt{\alpha q_1(1)}Z^u\Big)^\top (\Xi/\gamma + \alpha q_1(1)I_n)^{-1}\Big(\alpha q_1(1)\widetilde{u}^* + \sqrt{\alpha q_1(1)}Z^u\Big)\right)$$

$$\times \sqrt{\frac{\det(\Sigma)}{\det(\Sigma + q_2(1)I_d)}}\exp\left(\frac{1}{2}\Big(q_2(1)\widetilde{v}^* + \sqrt{q_2(1)}Z^v\Big)^\top (\Sigma + q_2(1)I_d)^{-1}\Big(q_2(1)\widetilde{v}^* + \sqrt{q_2(1)}Z^v\Big)\right),$$

where in the second equality, we use the Gaussian integral formula (Proposition G.1). Therefore

$$\lim_{n\to\infty} f_n(1) = \lim_{n\to\infty}\frac{1}{n}\mathbb{E}[\log \mathcal{Z}_{n,1}]$$

$$= \frac{1}{2}\alpha q_1(1)\gamma\mathbb{E}\Big[\overline{\Xi}^{-1}\Big] - \frac{1}{2}\mathbb{E}\Big[\log\Big(1 + \alpha q_1(1)\gamma\overline{\Xi}^{-1}\Big)\Big]$$

$$+ \frac{\alpha}{2}q_2(1)\mathbb{E}\Big[\overline{\Sigma}^{-1}\Big] - \frac{\alpha}{2}\mathbb{E}\Big[\log\Big(1 + q_2(1)\overline{\Sigma}^{-1}\Big)\Big],$$

verifying the identity (C.9). In the second equality, we have used Proposition G.2. $\qquad\square$

**Lemma C.2** (Free energy variation). *For all $t \in (0, 1)$,*

$$f'_n(t) = \frac{\alpha}{2}q'_1(t)q'_2(t) - \frac{1}{2}\mathbb{E}\left[\left\langle\left(\frac{\widetilde{u}^\top\widetilde{u}^*}{n} - q'_2(t)\right)\left(\frac{\widetilde{v}^\top\widetilde{v}^*}{n} - \alpha q'_1(t)\right)\right\rangle_{n,t}\right]. \tag{C.14}$$

*Proof.* From the definitions (C.4) and (C.5), we compute

$$f'_n(t) = \frac{1}{n}\mathbb{E}\left[\frac{1}{\mathcal{Z}_{n,t}}\frac{\partial}{\partial t}\mathcal{Z}_{n,t}\right]$$

$$= \frac{1}{n}\mathbb{E}\left[\frac{1}{\mathcal{Z}_{n,t}}\int_{\mathbb{R}^d}\int_{\mathbb{R}^n}\left(\frac{\partial}{\partial t}\mathcal{H}_{n,t}(\widetilde{u}, \widetilde{v}; q_1(t), q_2(t))\right)\exp(\mathcal{H}_{n,t}(\widetilde{u}, \widetilde{v}; q_1(t), q_2(t)))\,\mathrm{d}\widetilde{P}(\widetilde{u})\,\mathrm{d}\widetilde{Q}(\widetilde{v})\right]$$

$$= \frac{1}{n}\mathbb{E}\left[\left\langle\frac{\partial}{\partial t}\mathcal{H}_{n,t}(\widetilde{u}, \widetilde{v}; q_1(t), q_2(t))\right\rangle_{n,t}\right].$$

By the definition (C.2), the time derivative of the Hamiltonian is

$$\frac{\partial}{\partial t}\mathcal{H}_{n,t}(\widetilde{u}, \widetilde{v}; q_1(t), q_2(t)) = -\frac{1}{2\sqrt{(1-t)n}}\widetilde{u}^\top Z\widetilde{v} - \frac{1}{n}\widetilde{u}^\top\widetilde{u}^*\widetilde{v}^\top\widetilde{v}^* + \frac{1}{2n}\|\widetilde{u}\|_2^2\|\widetilde{v}\|_2^2$$

$$+ \alpha q'_1(t)\widetilde{u}^\top\widetilde{u}^* + \frac{\sqrt{\alpha}}{2\sqrt{q_1(t)}}q'_1(t)\widetilde{u}^\top Z^u - \frac{\alpha}{2}q'_1(t)\|\widetilde{u}\|_2^2 \tag{C.15}$$

$$+ q'_2(t)\widetilde{v}^\top\widetilde{v}^* + \frac{1}{2\sqrt{q_2(t)}}q'_2(t)\widetilde{v}^\top Z^v - \frac{1}{2}q'_2(t)\|\widetilde{v}\|_2^2.$$

The expectation of $\langle \partial_t \mathcal{H}_{n,t} \rangle_{n,t}$ can be computed using the Stein's lemma (Proposition G.7). Indeed, let us consider the term

$$
\begin{aligned}
\mathbb{E}\Big[\langle \widetilde{u}^\top Z \widetilde{v} \rangle_{n,t}\Big] &= \sum_{i=1}^{n}\sum_{j=1}^{d} \mathbb{E}\Big[\langle \widetilde{u}_i \widetilde{v}_j \rangle_{n,t} Z_{i,j}\Big] = \sum_{i=1}^{n}\sum_{j=1}^{d} \mathbb{E}\left[\frac{\partial \langle \widetilde{u}_i \widetilde{v}_j \rangle_{n,t}}{\partial Z_{i,j}}\right] \\
&= \sum_{i=1}^{n}\sum_{j=1}^{d}\sqrt{\frac{1-t}{n}}\,\mathbb{E}\Big[\langle \widetilde{u}_i^2 \widetilde{v}_j^2 \rangle_{n,t}\Big] - \sum_{i=1}^{n}\sum_{j=1}^{d}\sqrt{\frac{1-t}{n}}\,\mathbb{E}\Big[\langle \widetilde{u}_i \widetilde{v}_j \rangle_{n,t}^2\Big] \qquad \text{(C.16)} \\
&= \sqrt{\frac{1-t}{n}}\,\mathbb{E}\Big[\big\langle \|\widetilde{u}\|_2^2 \|\widetilde{v}\|_2^2 \big\rangle_{n,t}\Big] - \sqrt{\frac{1-t}{n}}\,\mathbb{E}\Big[\langle \widetilde{u}^\top \widetilde{u}^* \widetilde{v}^\top \widetilde{v}^* \rangle_{n,t}\Big],
\end{aligned}
$$

where the last step is by the Nishimori identity (Proposition G.4). So the first line of (C.15) upon taken the Gibbs bracket and the expectation becomes

$$
-\frac{1}{2n}\mathbb{E}\Big[\langle \widetilde{u}^\top \widetilde{u}^* \widetilde{v}^\top \widetilde{v}^* \rangle_{n,t}\Big].
$$

Similar cancellations happen for the second and third lines of (C.15). Putting them together, we obtain

$$
f_n'(t) = -\frac{1}{2n^2}\mathbb{E}\Big[\langle \widetilde{u}^\top \widetilde{u}^* \widetilde{v}^\top \widetilde{v}^* \rangle_{n,t}\Big] + \frac{\alpha}{2n}q_1'(t)\mathbb{E}\Big[\langle \widetilde{u}^\top \widetilde{u}^* \rangle_{n,t}\Big] + \frac{1}{2n}q_2'(t)\mathbb{E}\Big[\langle \widetilde{v}^\top \widetilde{v}^* \rangle_{n,t}\Big],
$$

which is the same as (C.14) with the parentheses opened up. $\qquad\square$

In what follows, our strategy is:

1. Show that $\langle \widetilde{u}^\top \widetilde{u}^* \rangle_{n,t}$ is concentrated around its mean $\mathbb{E}\Big[\langle \widetilde{u}^\top \widetilde{u}^* \rangle_{n,t}\Big]$;

2. Choose $q_2(t)$ to be the solution to

$$
q_2'(t) = \frac{1}{n}\mathbb{E}\Big[\langle \widetilde{u}^\top \widetilde{u}^* \rangle_{n,t}\Big].
$$

Once Items 1 and 2 are done, we then have

$$
\begin{aligned}
\mathcal{F}_n \approx f_n(0) &= f_n(1) - \int_0^1 f_n'(t)\,\mathrm{d}t \\
&\approx \psi_{\underline{\Xi}}(\alpha q_1(1)\gamma) + \alpha \psi_{\underline{\Sigma}}(q_2(1)) - \int_0^1 \frac{\alpha}{2}q_1'(t)q_2'(t)\,\mathrm{d}t \\
&\quad + \int_0^1 \frac{1}{2}\mathbb{E}\left[\left\langle \left(\frac{\widetilde{u}^\top \widetilde{u}^*}{n} - q_2'(t)\right)\left(\frac{\widetilde{v}^\top \widetilde{v}^*}{n} - \alpha q_1'(t)\right)\right\rangle_{n,t}\right]\mathrm{d}t \\
&\approx \psi_{\underline{\Xi}}(\alpha q_1(1)\gamma) + \alpha \psi_{\underline{\Sigma}}(q_2(1)) - \int_0^1 \frac{\alpha}{2}q_1'(t)q_2'(t)\,\mathrm{d}t,
\end{aligned}
$$

where the first line above uses (C.8) in Lemma C.1; the second line uses (C.9) in Lemma C.1 and Lemma C.2; the third line uses Items 1 and 2. This will almost lead to the desired characterization of the free energy $\mathcal{F}_n$ in Theorem 4.4:

$$
\sup_{q_v \geq 0}\,\inf_{q_u \geq 0}\,\psi_{\underline{\Xi}}(\alpha \gamma q_v) + \alpha \psi_{\underline{\Sigma}}(q_u) - \frac{\alpha}{2}q_u q_v.
$$

Consider the function

$$
\phi_t(q_1, q_2) = \frac{1}{n}\log \mathscr{Z}_{n,t}.
$$

Note that $\phi_t(q_1, q_2)$ also depends on $n$, $u^*$, $v^*$, $Z$, $Z^u$, $Z^v$, and $\mathbb{E}[\phi_t(q_1, q_2)] = f_n(t)$ where the expectation is over $(u^*, v^*, Z, Z^u, Z^v)$ distributed according to (C.1).

**Lemma C.3** (Free energy concentration). *Fix a constant $M > 0$. There exists a constant $C > 0$ depending only on $K, M, \alpha, \overline{\Xi}, \overline{\Sigma}$ such that for any $t \in [0, 1]$, $0 \le q_1(t), q_2(t) \le M$ and sufficiently large $n$,*

$$\mathbb{E}[|\phi_t(q_1, q_2) - \mathbb{E}[\phi_t(q_1, q_2)]|] \le \frac{C}{\sqrt{n}}.$$

*Proof.* Fix $u^*, v^*$. Consider $\phi_t(q_1, q_2)$ as a function of $(Z, Z^u, Z^v)$. Then

$$\left\| \nabla_{(Z, Z^u, Z^v)} \phi_t(q_1, q_2) \right\|_2^2$$

$$= \|\nabla_Z \phi_t(q_1, q_2)\|_2^2 + \|\nabla_{Z^u} \phi_t(q_1, q_2)\|_2^2 + \|\nabla_{Z^v} \phi_t(q_1, q_2)\|_2^2$$

$$= \sum_{i=1}^n \sum_{j=1}^d \left( \frac{1}{n} \mathbb{E}\left[ \left\langle \sqrt{\frac{1-t}{n}} \widetilde{u}_i \widetilde{v}_j \right\rangle_{n,t} \right] \right)^2 + \sum_{i=1}^n \left( \frac{1}{n} \mathbb{E}\left[ \langle \sqrt{\alpha q_1} \widetilde{u}_i \rangle_{n,t} \right] \right)^2 + \sum_{j=1}^d \left( \frac{1}{n} \mathbb{E}\left[ \langle \sqrt{q_2} \widetilde{v}_j \rangle_{n,t} \right] \right)^2$$

$$\le \frac{1}{n^3} \mathbb{E}\left[ \left\langle \|\widetilde{u}\|_2^2 \|\widetilde{v}\|_2^2 \right\rangle_{n,t} \right] + \frac{\alpha q_1}{n^2} \mathbb{E}\left[ \left\langle \|\widetilde{u}\|_2^2 \right\rangle_{n,t} \right] + \frac{q_2}{n^2} \mathbb{E}\left[ \left\langle \|\widetilde{v}\|_2^2 \right\rangle_{n,t} \right]$$

$$\le \frac{\alpha}{n} \|\Xi\|_2^{-1} \|\Sigma\|_2^{-1} K^4 + \frac{\alpha}{n} M \|\Xi\|_2^{-1} K^2 + \frac{\alpha}{n} M \|\Sigma\|_2^{-1} K^2 \le \frac{C}{n},$$

where $C > 0$ is a constant depending only on $\alpha, M, \overline{\Xi}, \overline{\Sigma}, K$. The penultimate line is by Cauchy–Schwarz and the last line holds for all sufficiently large $n$ by (3.3) and $n \asymp d$. Then by the Gaussian Poincaré inequality (Proposition G.8),

$$\mathop{\mathbb{E}}_{Z, Z^u, Z^v}\left[ \left| \phi_t(q_1, q_2) - \mathop{\mathbb{E}}_{Z, Z^u, Z^v}[\phi_t(q_1, q_2)] \right| \right] \le \sqrt{\mathop{\mathrm{Var}}_{Z, Z^u, Z^v}[\phi_t(q_1, q_2)]} \le \frac{C}{\sqrt{n}}. \tag{C.17}$$

The above result holds for any fixed $u^*, v^*$. We then verify that $\mathbb{E}_{Z, Z^u, Z^v}[\phi_t(q_1, q_2)]$ has bounded difference as a function of $u^*, v^*$. We do so by bounding the derivatives of $\mathbb{E}_{Z, Z^u, Z^v}[\phi_t(q_1, q_2)]$ with respect to $u_i^*, v_j^*$ for any $i \in [n], j \in [d]$. We have

$$\frac{\partial}{\partial u_i^*} \mathop{\mathbb{E}}_{Z, Z^u, Z^v}[\phi_t(q_1, q_2)] = \frac{1}{n} \mathop{\mathbb{E}}_{Z, Z^u, Z^v}\left[ \left\langle \frac{\partial}{\partial u_i^*} \mathcal{H}_{n,t}(\widetilde{u}, \widetilde{v}; q_1, q_2) \right\rangle_{n,t} \right], \tag{C.18}$$

and a similar expression holds for the derivative with respect to $v_j^*$. Recall the definition of $\mathcal{H}_{n,t}$ from (C.2). We have

$$\frac{\partial}{\partial u_i^*} \mathcal{H}_{n,t}(\Xi^{-1/2} u, \Sigma^{-1/2} v; q_1, q_2) = \frac{1-t}{n} v^\top \Sigma^{-1} v^* (\Xi^{-1})_{i,i} u_i + \alpha q_1 (\Xi^{-1})_{i,i} u_i,$$

where we have used the fact that $\Xi$ is diagonal. Therefore,

$$\left| \frac{\partial}{\partial u_i^*} \mathcal{H}_{n,t}(\Xi^{-1/2} u, \Sigma^{-1/2} v; q_1, q_2) \right| \le \frac{1}{n} \cdot d K^2 \|\Sigma\|_2^{-1} \cdot \|\Xi\|_2^{-1} K + \alpha q_1 \cdot \|\Xi\|_2^{-1} K \le C,$$

where $C > 0$ is a constant depending only on $\alpha, M, K, \overline{\Xi}, \overline{\Sigma}$. The last inequality holds for all sufficiently large $n$. A similar bound holds for $\frac{\partial}{\partial v_j^*} \mathcal{H}_{n,t}(\Xi^{-1/2} u, \Sigma^{-1/2} v; q_1, q_2)$. This, by (C.18), implies that $\mathbb{E}_{Z, Z^u, Z^v}[\phi_t(q_1, q_2)]$ as a function of $(u^*, v^*)$ satisfies the bounded difference property with $c_i = C/n$ (see Proposition G.9). So by Proposition G.9,

$$\mathop{\mathbb{E}}_{u^*, v^*}\left[ \left| \mathop{\mathbb{E}}_{Z, Z^u, Z^v}[\phi_t(q_1, q_2)] - \mathop{\mathbb{E}}_{u^*, v^*}\left[ \mathop{\mathbb{E}}_{Z, Z^u, Z^v}[\phi_t(q_1, q_2)] \right] \right| \right] \le \sqrt{\mathop{\mathrm{Var}}_{u^*, v^*}\left[ \mathop{\mathbb{E}}_{Z, Z^u, Z^v}[\phi_t(q_1, q_2)] \right]} \le \frac{C}{\sqrt{n}}. \tag{C.19}$$

Finally using (C.17) and (C.19) and the triangle inequality,

$$\mathbb{E}[|\phi_t(q_1, q_2) - \mathbb{E}[\phi_t(q_1, q_2)]|]$$

$$\le \mathop{\mathbb{E}}_{u^*, v^*}\left[ \mathop{\mathbb{E}}_{Z, Z^u, Z^v}\left[ \left| \phi_t(q_1, q_2) - \mathop{\mathbb{E}}_{Z, Z^u, Z^v}[\phi_t(q_1, q_2)] \right| \right] \right] + \left| \mathop{\mathbb{E}}_{Z, Z^u, Z^v}[\phi_t(q_1, q_2)] - \mathop{\mathbb{E}}_{u^*, v^*}\left[ \mathop{\mathbb{E}}_{Z, Z^u, Z^v}[\phi_t(q_1, q_2)] \right] \right|$$

$$\le \frac{C}{\sqrt{n}},$$

concluding the proof. $\qquad\qquad\square$

Suppose $a, b \geq 0$ are constants. With $q_1 = s_n a^2$, $q_2 = s_n b^2$, we can write $\mathcal{H}_{n,t}$ in (C.2) as

$$\mathcal{H}_{n,t}(\widetilde{u}, \widetilde{v}; s_n a^2, s_n b^2) = \mathcal{H}_{n,t}(\widetilde{u}, \widetilde{v}) + \mathcal{H}_{n,a}^u(\widetilde{u}) + \mathcal{H}_{n,b}^v(\widetilde{v}),$$

where

$$\mathcal{H}_{n,t}(\widetilde{u}, \widetilde{v}) := \sqrt{\frac{1-t}{n}} \widetilde{u}^\top Z \widetilde{v} + \frac{1-t}{n} \widetilde{u}^\top \widetilde{u}^* \widetilde{v}^\top \widetilde{v}^* - \frac{1-t}{2n} \|\widetilde{u}\|_2^2 \|\widetilde{vv}\|_2^2,$$

$$\mathcal{H}_{n,a}^u(\widetilde{u}) := \alpha s_n a^2 \widetilde{u}^\top \widetilde{u}^* + a\sqrt{\alpha s_n} \widetilde{u}^\top Z^u - \frac{\alpha s_n a^2}{2} \|\widetilde{u}\|_2^2,$$

$$\mathcal{H}_{n,b}^v(\widetilde{v}) := s_n b^2 \widetilde{v}^\top \widetilde{v}^* + b\sqrt{s_n} \widetilde{v}^\top Z^v - \frac{s_n b^2}{2} \|\widetilde{v}\|_2^2.$$

Fix $A \geq 2$. Define

$$\phi^u(a) = \frac{1}{ns_n} \log \int_{\mathbb{R}^n} \exp\big(\mathcal{H}_{n,a}^u(\widetilde{u})\big) \, \mathrm{d}\widetilde{P}(\widetilde{u}), \qquad \phi^v(b) = \frac{1}{ns_n} \log \int_{\mathbb{R}^d} \exp\big(\mathcal{H}_{n,b}^v(\widetilde{v})\big) \, \mathrm{d}\widetilde{Q}(\widetilde{v}),$$

$$\xi_n^u(s_n) := \sup_{1/2 \leq a \leq A+1/2} \mathbb{E}[|\phi^u(a) - \mathbb{E}[\phi^u(a)]|], \quad \xi_n^v(s_n) := \sup_{1/2 \leq b \leq A+1/2} \mathbb{E}[|\phi^v(b) - \mathbb{E}[\phi^v(b)]|],$$

$$\tag{C.20}$$

$$\Upsilon^u(\widetilde{u}) := \frac{1}{ns_n} \frac{\partial}{\partial a} \mathcal{H}_{n,a}^u(\widetilde{u}), \qquad\qquad \Upsilon^v(\widetilde{v}) := \frac{1}{ns_n} \frac{\partial}{\partial b} \mathcal{H}_{n,b}^b(\widetilde{v}).$$

Denote by $\langle g(u^*, v^*) \rangle_{n,a,b}$ the conditional expectation of $g(u^*, v^*)$ given

$$Y_t, \quad Y_t^u(a) := a\sqrt{\alpha s_n} u^* + \Xi^{1/2} Z^u, \quad Y_t^v(b) := b\sqrt{s_n} v^* + \Sigma^{1/2} Z^v, \tag{C.21}$$

where the expectation is with respect to the distribution in (C.1).

**Corollary C.4.** *Let $s_n = n^{-1/32}$ and $A \leq \sqrt{M/s_n} - 1/2$ for a constant $M > 0$ independent of $n$. Then there exists a constant $C > 0$ depending only on $K, \alpha, \overline{\Xi}$ such that for all sufficiently large $n$,*

$$\xi_n^u(s_n) \leq \frac{C}{s_n \sqrt{n}}.$$

*Proof.* Note that if $b = 0, t = 1$, it holds that $\mathcal{H}_{n,1}(\widetilde{u}, \widetilde{v}; s_n a^2, 0) = \mathcal{H}_{n,a}^u(\widetilde{u})$ and $\phi_1(s_n a^2, 0) = s_n \phi^u(a)$. The conclusion then follows immediately from Lemma C.3 since by the assumption on $A$, $q_1 = s_n a^2 \in [0, M]$ for any $1/2 \leq a \leq A + 1/2$. $\qquad\square$

**Lemma C.5.** *Let $s_n = n^{-1/32}$. For all $A \geq 2$,*

$$\frac{1}{A-1} \int_1^A \mathbb{E}\Big[\Big\langle \Big|\Upsilon^u(\widetilde{u}) - \mathbb{E}\big[\langle \Upsilon^u(\widetilde{u}) \rangle_{n,a,b}\big]\Big| \Big\rangle_{n,a,b}\Big] \, \mathrm{d}a \leq C\left(\frac{1}{\sqrt{ns_n}} + \sqrt{\xi_n^u(s_n)}\right),$$

*where $C > 0$ only depends on $\alpha, \overline{\Xi}, K$.*

*Proof.* By the triangle inequality,

$$\mathbb{E}\Big[\Big\langle \Big|\Upsilon^u(\widetilde{u}) - \mathbb{E}\big[\langle \Upsilon^u(\widetilde{u}) \rangle_{n,a,b}\big]\Big| \Big\rangle_{n,a,b}\Big] \leq \mathbb{E}\Big[\Big\langle \Big|\Upsilon^u(\widetilde{u}) - \langle \Upsilon^u(\widetilde{u}) \rangle_{n,a,b}\Big| \Big\rangle_{n,a,b}\Big]$$

$$+ \mathbb{E}\Big[\Big\langle \Big|\langle \Upsilon^u(\widetilde{u}) \rangle_{n,a,b} - \mathbb{E}\big[\langle \Upsilon^u(\widetilde{u}) \rangle_{n,a,b}\big]\Big| \Big\rangle_{n,a,b}\Big].$$

$$\tag{C.22}$$

We will bound the two terms on the RHS separately. We first bound

$$\frac{1}{A-1} \int_1^A \mathbb{E}\Big[\Big\langle \Big|\Upsilon^u(\widetilde{u}) - \langle \Upsilon^u(\widetilde{u}) \rangle_{n,a,b}\Big| \Big\rangle_{n,a,b}\Big] \, \mathrm{d}a$$

$$\leq \left(\frac{1}{A-1} \int_1^A \mathbb{E}\Big[\Big\langle \Big(\Upsilon^u(\widetilde{u}) - \langle \Upsilon^u(\widetilde{u}) \rangle_{n,a,b}\Big)^2 \Big\rangle_{n,a,b}\Big] \, \mathrm{d}a\right)^{1/2}. \tag{C.23}$$

The first two derivatives of $\phi^u$ are

$$(\phi^u)'(a) = \frac{1}{ns_n} \frac{\int_{\mathbb{R}^n} \exp\big(\mathcal{H}^u_{n,a}(\widetilde{u})\big) \frac{\partial}{\partial a} \mathcal{H}^u_{n,a}(\widetilde{u}) \, d\widetilde{P}(\widetilde{u})}{\int_{\mathbb{R}^n} \exp\big(\mathcal{H}^u_{n,a}(\widetilde{u})\big) \, d\widetilde{P}(\widetilde{u})} = \langle \Upsilon^u(\widetilde{u}) \rangle_{n,a,b}, \tag{C.24a}$$

$$(\phi^u)''(a) = \frac{1}{ns_n}\left[ \left\langle \left(\frac{\partial}{\partial a}\mathcal{H}^u_{n,a}(\widetilde{u})\right)^2 \right\rangle_{n,a,b} - \left\langle \frac{\partial}{\partial a}\mathcal{H}^u_{n,a}(\widetilde{u}) \right\rangle^2_{n,a,b} + \left\langle \frac{\partial^2}{\partial a^2}\mathcal{H}^u_{n,a}(\widetilde{u}) \right\rangle_{n,a,b} \right]$$

$$= ns_n\left[ \langle \Upsilon^u(\widetilde{u})^2 \rangle_{n,a,b} - \langle \Upsilon^u(\widetilde{u}) \rangle^2_{n,a,b} \right] + \frac{\alpha}{n}\left[ 2\langle \widetilde{u}^\top \widetilde{u}^* \rangle_{n,a,b} - \left\langle \|\widetilde{u}\|^2_2 \right\rangle_{n,a,b} \right]. \tag{C.24b}$$

Since there exists $C > 0$ depending only on $\alpha, \overline{\overline{\Xi}}, K$ such that for all sufficiently large $n$,

$$\left| \frac{\alpha}{n}\left[ 2\langle \widetilde{u}^\top \widetilde{u}^* \rangle_{n,a,b} - \left\langle \|\widetilde{u}\|^2_2 \right\rangle_{n,a,b} \right] \right| \leq \frac{4\alpha K^2}{\inf \mathrm{supp}(\overline{\Xi})} =: C, \tag{C.25}$$

the second result (C.24b) above implies that for all sufficiently large $n$,

$$\left\langle \left(\Upsilon^u(\widetilde{u}) - \langle \Upsilon^u(\widetilde{u}) \rangle_{n,a,b}\right)^2 \right\rangle_{n,a,b} \leq \frac{1}{ns_n}((\phi^u)''(a) + C).$$

Consequently,

$$\int_1^A \mathbb{E}\left[ \left\langle \left(\Upsilon^u(\widetilde{u}) - \langle \Upsilon^u(\widetilde{u}) \rangle_{n,a,b}\right)^2 \right\rangle_{n,a,b} \right] da \leq \frac{1}{ns_n}\left( \mathbb{E}[(\phi^u)'(A)] - \mathbb{E}[(\phi^u)'(1)] + C(A-1) \right). \tag{C.26}$$

To proceed, we compute for any $a$,

$$\mathbb{E}[(\phi^u)'(a)] = \mathbb{E}\left[ \langle \Upsilon^u(\widetilde{u}) \rangle_{n,a,b} \right], \tag{C.27}$$

which is by (C.24a). By definition,

$$\Upsilon^u(\widetilde{u}) = \frac{1}{n}\left( 2\alpha a \widetilde{u}^\top \widetilde{u}^* + \sqrt{\alpha/s_n}\, \widetilde{u}^\top Z^u - \alpha a \|\widetilde{u}\|^2_2 \right),$$

whose expectation is therefore given by

$$\mathbb{E}\left[ \langle \Upsilon^u(\widetilde{u}) \rangle_{n,a,b} \right] = \frac{2\alpha a}{n}\mathbb{E}\left[ \langle \widetilde{u}^\top \widetilde{u}^* \rangle_{n,a,b} \right] + \frac{\sqrt{\alpha}}{n\sqrt{s_n}}\mathbb{E}\left[ \langle \widetilde{u}^\top Z^u \rangle_{n,a,b} \right] - \frac{\alpha a}{n}\mathbb{E}\left[ \left\langle \|\widetilde{u}\|^2_2 \right\rangle_{n,a,b} \right]. \tag{C.28}$$

Using Stein's lemma (Proposition G.7), the middle term is equal to

$$\frac{\sqrt{\alpha}}{n\sqrt{s_n}}\mathbb{E}\left[ \langle \widetilde{u}^\top Z^u \rangle_{n,a,b} \right] = \frac{\sqrt{\alpha}}{n\sqrt{s_n}}\sum_{i=1}^n \mathbb{E}\left[ Z^u_i \langle \widetilde{u}_i \rangle_{n,a,b} \right] = \frac{\sqrt{\alpha}}{n\sqrt{s_n}}\sum_{i=1}^n \mathbb{E}\left[ \frac{\partial}{\partial Z^u_i} \langle \widetilde{u}_i \rangle_{n,a,b} \right]$$

$$= \frac{a\alpha}{n}\mathbb{E}\left[ \left\langle \|\widetilde{u}\|^2_2 \right\rangle_{n,a,b} \right] - \frac{a\alpha}{n}\mathbb{E}\left[ \langle \widetilde{u}^\top \widetilde{u}^* \rangle_{n,a,b} \right].$$

Therefore,

$$\mathbb{E}\left[ \langle \Upsilon^u(\widetilde{u}) \rangle_{n,a,b} \right] = \frac{\alpha a}{n}\mathbb{E}\left[ \langle \widetilde{u}^\top \widetilde{u}^* \rangle_{n,a,b} \right], \tag{C.29}$$

and

$$\left| \mathbb{E}\left[ \langle \Upsilon^u(\widetilde{u}) \rangle_{n,a,b} \right] \right| \leq aC, \tag{C.30}$$

where $C$ depends only on $\alpha, \overline{\overline{\Xi}}, K$.

Using the last inequality, we can further upper bound the RHS of (C.26) by $\frac{CA}{ns_n}$ for some $C$ depending only on $\alpha, \overline{\overline{\Xi}}, K$. Putting this back to (C.23), we get

$$\frac{1}{A-1}\int_1^A \mathbb{E}\left[ \left\langle \left| \Upsilon^u(\widetilde{u}) - \langle \Upsilon^u(\widetilde{u}) \rangle_{n,a,b} \right| \right\rangle_{n,a,b} \right] da \leq \sqrt{\frac{CA}{ns_n(A-1)}} \leq \sqrt{\frac{2C}{ns_n}}, \tag{C.31}$$

since $A \geq 2$.

Now it remains to bound the second term on the RHS of (C.22) which can be written as

$$\mathbb{E}\left[\left\langle\left|\langle\Upsilon^u(\widetilde{u})\rangle_{n,a,b} - \mathbb{E}\left[\langle\Upsilon^u(\widetilde{u})\rangle_{n,a,b}\right]\right|\right\rangle_{n,a,b}\right] = \mathbb{E}[|(\phi^u)'(a) - \mathbb{E}[(\phi^u)'(a)]|] \tag{C.32}$$

using (C.24a). To further bound the RHS, consider the following two functions

$$a \mapsto \phi^u(a) + \frac{2\alpha K^2 a^2}{\inf \operatorname{supp}(\overline{\Xi})}, \quad a \mapsto \mathbb{E}[\phi^u(a)] + \frac{2\alpha K^2 a^2}{\inf \operatorname{supp}(\overline{\Xi})}.$$

They are both differentiable and convex for all sufficiently large $n$ since their second derivatives are non-negative by (C.25) and (C.24b). Applying Proposition G.10 with the above two functions, taking the expectation and using the triangle inequality, we have that for any $1 \leq a \leq A, 0 < a' \leq 1/2$,

$$\mathbb{E}[|(\phi^u)'(a) - \mathbb{E}[(\phi^u)'(a)]|] \leq \mathbb{E}[(\phi^u)'(a + a')] - \mathbb{E}[(\phi^u)'(a - a')] + 3\xi_n^u(s_n)/a' + \frac{8\alpha K^2 a'}{\inf \operatorname{supp}(\overline{\Xi})}. \tag{C.33}$$

Then

$$\begin{aligned}
&\int_1^A \mathbb{E}[(\phi^u)'(a + a')] - \mathbb{E}[(\phi^u)'(a - a')]\, da \\
&= (\mathbb{E}[\phi^u(A + a')] - \mathbb{E}[\phi^u(1 + a')]) - (\mathbb{E}[\phi^u(A - a')] - \mathbb{E}[\phi^u(1 - a')]) \\
&= (\mathbb{E}[\phi^u(A + a')] - \mathbb{E}[\phi^u(A - a')]) - (\mathbb{E}[\phi^u(1 + a')] - \mathbb{E}[\phi^u(1 - a')]) \\
&= \int_{-a'}^{a'} \mathbb{E}[(\phi^u)'(A + a)]\, da - \int_{-a'}^{a'} \mathbb{E}[(\phi^u)'(1 + a)]\, da \\
&\leq 4a'(A + a')C \leq 8a'AC,
\end{aligned}$$

where the last step is by (C.27) and (C.30), and $C$ depends only on $\alpha, \overline{\Xi}, K$. Using this in (C.33), we obtain

$$\int_1^A \mathbb{E}[|(\phi^u)'(a) - \mathbb{E}[(\phi^u)'(a)]|]\, da \leq CA(a' + \xi_n^u(s_n)/a'),$$

and the RHS is minimized by $a' = \sqrt{\xi_n^u(s_n)}$ which lies in the interval $(0, 1/2]$ for all sufficiently large $n$ due to Corollary C.4. Using this result in (C.32) and integrating over $a$, we have

$$\frac{1}{A - 1}\int_1^A \mathbb{E}\left[\left\langle\left|\langle\Upsilon^u(\widetilde{u})\rangle_{n,a,b} - \mathbb{E}\left[\langle\Upsilon^u(\widetilde{u})\rangle_{n,a,b}\right]\right|\right\rangle_{n,a,b}\right] da \leq \frac{CA}{A - 1} \cdot 2\sqrt{\xi_n^u(s_n)} \leq 4C\sqrt{\xi_n^u(s_n)}, \tag{C.34}$$

since $A \geq 2$.

Finally, combining (C.22), (C.31) and (C.34) proves the lemma. $\qquad\square$

**Lemma C.6.** *There exists $C > 0$ depending only on $\alpha, K, \overline{\Xi}$ such that for any $A \geq 2$,*

$$\frac{1}{A - 1}\int_1^A \frac{1}{n^2}\mathbb{E}\left[\left\langle\left(\left(\widetilde{u}^{(1)}\right)^\top\widetilde{u}^{(2)} - \mathbb{E}\left[\left\langle\left(\widetilde{u}^{(1)}\right)^\top\widetilde{u}^{(2)}\right\rangle_{n,a,b}\right]\right)^2\right\rangle_{n,a,b}\right] da \leq C\left(\frac{1}{\sqrt{ns_n}} + \sqrt{\xi_n^u(s_n)}\right),$$

*where $\widetilde{u}^{(1)} = \Xi^{-1/2}u^{(1)}, \widetilde{u}^{(2)} = \Xi^{-1/2}u^{(2)}$ with $u^{(1)}, u^{(2)}$ being two i.i.d. copies from the conditional law of $u^*, v^*$ given (C.21).*

*Proof.* Since $\operatorname{supp}(P), \operatorname{supp}(Q), \|\Xi\|_2^{-1}$ are all bounded, by the triangle inequality,

$$\frac{1}{n}\left|\mathbb{E}\left[\left\langle\Upsilon^u(\widetilde{u}^{(1)})(\widetilde{u}^{(1)})^\top\widetilde{u}^{(2)}\right\rangle_{n,a,b}\right] - \mathbb{E}\left[\langle\Upsilon^u(\widetilde{u})\rangle_{n,a,b}\right]\mathbb{E}\left[\left\langle(\widetilde{u}^{(1)})^\top\widetilde{u}^{(2)}\right\rangle_{n,a,b}\right]\right| \tag{C.35}$$

$$\leq \|\Xi\|_2^{-1}K^2\mathbb{E}\left[\left\langle\left|\Upsilon^u(\widetilde{u}) - \mathbb{E}\left[\langle\Upsilon^u(\widetilde{u})\rangle_{n,a,b}\right]\right|\right\rangle_{n,a,b}\right]. \tag{C.36}$$

On the other hand, let us compute (C.35) on the LHS of the above inequality. Recall from (C.29) in the proof of Lemma C.5 that

$$\mathbb{E}\left[\left\langle \Upsilon^u(\widetilde{u}^{(1)})\right\rangle_{n,a,b}\right] = \frac{\alpha a}{n}\mathbb{E}\left[\left\langle (\widetilde{u}^{(1)})^\top \widetilde{u}^{(2)}\right\rangle_{n,a,b}\right]. \tag{C.37}$$

Similar to (C.28), we have

$$\mathbb{E}\left[\left\langle \Upsilon^u(\widetilde{u}^{(1)})(\widetilde{u}^{(1)})^\top \widetilde{u}^{(2)}\right\rangle_{n,a,b}\right] = \frac{2\alpha a}{n}\mathbb{E}\left[\left\langle (\widetilde{u}^{(1)})^\top \widetilde{u}^*(\widetilde{u}^{(1)})^\top \widetilde{u}^{(2)}\right\rangle_{n,a,b}\right]$$
$$+ \frac{\sqrt{\alpha}}{n\sqrt{s_n}}\mathbb{E}\left[\left\langle (\widetilde{u}^{(1)})^\top Z^u(\widetilde{u}^{(1)})^\top \widetilde{u}^{(2)}\right\rangle_{n,a,b}\right] - \frac{\alpha a}{n}\mathbb{E}\left[\left\langle \left\|\widetilde{u}^{(1)}\right\|_2^2(\widetilde{u}^{(1)})^\top \widetilde{u}^{(2)}\right\rangle_{n,a,b}\right].$$

Using Stein's lemma and following similar derivations leading to (C.29), the second term on the RHS equals

$$\frac{\sqrt{\alpha}}{n\sqrt{s_n}}\mathbb{E}\left[\left\langle (\widetilde{u}^{(1)})^\top Z^u(\widetilde{u}^{(1)})^\top \widetilde{u}^{(2)}\right\rangle_{n,a,b}\right] = \frac{a\alpha}{n}\mathbb{E}\left[\left\langle \left\|\widetilde{u}^{(1)}\right\|_2^2(\widetilde{u}^{(1)})^\top \widetilde{u}^{(2)}\right\rangle_{n,a,b}\right]$$
$$- \frac{2a\alpha}{n}\mathbb{E}\left[\left\langle (\widetilde{u}^{(1)})^\top \widetilde{u}^{(3)}(\widetilde{u}^{(1)})^\top \widetilde{u}^{(2)}\right\rangle_{n,a,b}\right] + \frac{a\alpha}{n}\mathbb{E}\left[\left\langle \left((\widetilde{u}^{(1)})^\top \widetilde{u}^{(2)}\right)^2\right\rangle_{n,a,b}\right].$$

Therefore,

$$\mathbb{E}\left[\left\langle \Upsilon^u(\widetilde{u}^{(1)})(\widetilde{u}^{(1)})^\top \widetilde{u}^{(2)}\right\rangle_{n,a,b}\right] = \frac{\alpha a}{n}\mathbb{E}\left[\left\langle \left((\widetilde{u}^{(1)})^\top \widetilde{u}^{(2)}\right)^2\right\rangle_{n,a,b}\right]. \tag{C.38}$$

Putting (C.37) and (C.38) together and using Nishimori identity (Proposition G.4), we have that (C.35) equals

$$\frac{\alpha a}{n^2}\left|\mathbb{E}\left[\left\langle \left((\widetilde{u}^{(1)})^\top \widetilde{u}^{(2)}\right)^2\right\rangle_{n,a,b}\right] - \mathbb{E}\left[\left\langle (\widetilde{u}^{(1)})^\top \widetilde{u}^{(2)}\right\rangle_{n,a,b}\right]^2\right|$$
$$= \frac{\alpha a}{n^2}\left|\mathbb{E}\left[\left\langle \left((\widetilde{u}^{(1)})^\top \widetilde{u}^{(2)} - \mathbb{E}\left[\left\langle (\widetilde{u}^{(1)})^\top \widetilde{u}^{(2)}\right\rangle_{n,a,b}\right]\right)^2\right\rangle_{n,a,b}\right]\right|.$$

So by the inequality (C.36), for any $a \geq 1$,

$$\frac{1}{n^2}\mathbb{E}\left[\left\langle \left((\widetilde{u}^{(1)})^\top \widetilde{u}^{(2)} - \mathbb{E}\left[\left\langle (\widetilde{u}^{(1)})^\top \widetilde{u}^{(2)}\right\rangle_{n,a,b}\right]\right)^2\right\rangle_{n,a,b}\right] \leq \frac{\|\Xi\|_2^{-1}K^2}{\alpha}\mathbb{E}\left[\left\langle \left|\Upsilon^u(\widetilde{u}) - \mathbb{E}\left[\left\langle \Upsilon^u(\widetilde{u})\right\rangle_{n,a,b}\right]\right|\right\rangle_{n,a,b}\right].$$

Integrating over $a \in [1, A]$ and invoking Lemma C.5 concludes the proof. $\qquad\square$

**Lemma C.7** (Overlap concentration). *Let $R_1, R_2 \colon [0,1] \times \mathbb{R}_{>0}^2 \to \mathbb{R}_{\geq 0}$ be two continuous bounded functions such that their partial derivatives with respect to the second and third arguments are continuous and non-negative. Let $s_n = n^{-1/32}$. For $(\varepsilon_1, \varepsilon_2) \in [1,2]^2$, slightly abusing notation, let $q_1(\cdot, \varepsilon_1, \varepsilon_2), q_2(\cdot, \varepsilon_1, \varepsilon_2)$ be the unique solution to*

$$\begin{cases} q_1(0) = s_n\varepsilon_1 \\ q_2(0) = s_n\varepsilon_2 \end{cases}, \qquad \begin{cases} q_1'(t) = R_1(t, q_1(t), q_2(t)) \\ q_2'(t) = R_2(t, q_1(t), q_2(t)) \end{cases}. \tag{C.39}$$

*Then there exists a constant $C > 0$ depending only on $K$, $\alpha$, $\|R_1\|_\infty$, $\|R_2\|_\infty$, $\overline{\overline{\Xi}}$ such that for every $t \in [0,1]$,*

$$\int_1^2 \int_1^2 \frac{1}{n^2}\mathbb{E}\left[\left\langle \left(\widetilde{u}^\top \widetilde{u}^* - \mathbb{E}\left[\left\langle \widetilde{u}^\top \widetilde{u}^*\right\rangle_{n,t}\right]\right)^2\right\rangle_{n,t}\right] d\varepsilon_1 \, d\varepsilon_2 \leq Cn^{-1/8}.$$

*Proof.* The existence and uniqueness of the solution to the Cauchy problem (C.39) is a direct consequence of the Cauchy–Lipschitz theorem [37, Theorem 3.1, Chapter V]. For any $t \in [0,1]$, the function $Q_t(\varepsilon_1, \varepsilon_2) = (q_1(t, \varepsilon_1, \varepsilon_2), q_2(t, \varepsilon_1, \varepsilon_2))$ is a $C^1$-diffeomorphism. Its Jacobian determinant is given by the Liouville formula [37, Corollary 3.1, Chapter V] and can be lower bounded as

$$J(\varepsilon_1, \varepsilon_2) := \det(\nabla Q_t(\varepsilon_1, \varepsilon_2))$$
$$= s_n^2 \exp\left( \int_0^t \partial_2 R_1(s, Q_s(\varepsilon_1, \varepsilon_2)) \, ds + \int_0^t \partial_3 R_2(s, Q_s(\varepsilon_1, \varepsilon_2)) \, ds \right) \geq s_n^2, \quad \text{(C.40)}$$

since the partial derivatives are non-negative by assumptions.

We then view the RHS below as a function of $q_1, q_2$ and denote it by

$$V(q_1, q_2) = \mathbb{E}\left[ \left\langle \left( \widetilde{u}^\top \widetilde{u}^* - \mathbb{E}\left[ \langle \widetilde{u}^\top \widetilde{u}^* \rangle_{n,t} \right] \right)^2 \right\rangle_{n,t} \right]. \quad \text{(C.41)}$$

Denote $\Omega_t = Q_t([1,2]^2)/s_n$ and $M := \max\{\|R_1\|_\infty, \|R_2\|_\infty\} + 2$. Since $R_1, R_2 \geq 0$ by assumptions, $q_1, q_2$ are non-decreasing in $t$ by (C.39). So for $i \in \{1,2\}$, and any $t \in [0,1]$ and $(\varepsilon_1, \varepsilon_2) \in [1,2]^2$,

$$q_i(t, \varepsilon_1, \varepsilon_2)/s_n \geq q_i(0, \varepsilon_1, \varepsilon_2)/s_n = \varepsilon_i \geq 1,$$

$$q_i(t, \varepsilon_1, \varepsilon_2)/s_n \leq q_i(1, \varepsilon_1, \varepsilon_2)/s_n = s_n^{-1} \int_0^1 q_i'(t, \varepsilon_1, \varepsilon_2) \, dt + s_n^{-1} q_i(0, \varepsilon_1, \varepsilon_2) \leq s_n^{-1}(\|R_i\|_\infty + 2).$$

We obtain the relation $\Omega_t \subset [1, M/s_n]^2$ for any $t \in [0,1]$.

Next, using the change of variable $(r_1, r_2) = Q_t(\varepsilon_1, \varepsilon_2)/s_n$, we have

$$\int_1^2 \int_1^2 \frac{1}{n^2} \mathbb{E}\left[ \left\langle \left( \widetilde{u}^\top \widetilde{u}^* - \mathbb{E}\left[ \langle \widetilde{u}^\top \widetilde{u}^* \rangle_{n,t} \right] \right)^2 \right\rangle_{n,t} \right] d\varepsilon_1 \, d\varepsilon_2$$
$$= \frac{1}{n^2} \int_1^2 \int_1^2 V(q_1(t, \varepsilon_1, \varepsilon_2), q_2(t, \varepsilon_1, \varepsilon_2)) \, d\varepsilon_1 \, d\varepsilon_2$$
$$= \frac{1}{n^2} \int_{\Omega_t} \frac{V(s_n r_1, s_n r_2) s_n^2}{J(Q_t^{-1}(s_n r_1, s_n r_2))} \, d(r_1, r_2)$$
$$\leq \frac{1}{n^2} \int_1^{M/s_n} \int_1^{M/s_n} V(s_n r_1, s_n r_2) \, dr_1 \, dr_2, \quad \text{(C.42)}$$

where the last step is by (C.40). Further applying the change of variable $r_1 = a^2$, we have that for all $r_2 \geq 1$,

$$\frac{1}{n^2} \int_1^{M/s_n} V(s_n r_1, s_n r_2) \, dr_1 = \frac{1}{n^2} \int_1^{\sqrt{M/s_n}} V(s_n a^2, s_n r_2) 2a \, da \leq \frac{2\sqrt{M/s_n}}{n^2} \int_1^{\sqrt{M/s_n}} V(s_n a^2, s_n r_2) \, da. \quad \text{(C.43)}$$

Recalling $V$ from (C.41), we recognize that

$$V(s_n a^2, s_n r_2) = \mathbb{E}\left[ \left\langle \left( \widetilde{u}^\top \widetilde{u}^* - \mathbb{E}\left[ \langle \widetilde{u}^\top \widetilde{u}^* \rangle_{n,a,\sqrt{r_2}} \right] \right)^2 \right\rangle_{n,a,\sqrt{r_2}} \right]$$
$$= \mathbb{E}\left[ \left\langle \left( (\widetilde{u}^{(1)})^\top \widetilde{u}^{(2)} - \mathbb{E}\left[ \langle (\widetilde{u}^{(1)})^\top \widetilde{u}^{(2)} \rangle_{n,a,\sqrt{r_2}} \right] \right)^2 \right\rangle_{n,a,\sqrt{r_2}} \right],$$

where $\langle \cdot \rangle_{n,a,b}$ is defined in (C.21) and the second equality is by Nishimori identity (Proposition G.4). Since $\sqrt{M/s_n} \geq 2$ for all sufficiently large $n$, applying Lemma C.6, we get

$$\frac{1}{\sqrt{M/s_n} - 1} \int_1^{\sqrt{M/s_n}} \frac{1}{n^2} V(s_n a^2, s_n r_2) \, da \leq C\left( \frac{1}{\sqrt{n s_n}} + \sqrt{\xi_n^u(s_n)} \right),$$

where $C > 0$ depends only on $\alpha, K, \overline{\overline{\Xi}}$ and $\xi_n^u(s_n)$ is given in (C.20) with $A = \sqrt{M/s_n}$. Using this back in (C.43) and then in (C.42), we obtain

$$\int_1^2 \int_1^2 \frac{1}{n^2} \mathbb{E}\left[\left\langle \left(\widetilde{u}^\top \widetilde{u}^* - \mathbb{E}\left[\langle \widetilde{u}^\top \widetilde{u}^* \rangle_{n,t}\right]\right)^2 \right\rangle_{n,t}\right] d\varepsilon_1 \, d\varepsilon_2 \leq 2C(M/s_n)^2 (1/\sqrt{ns_n} + \sqrt{\xi_n^u(s_n)}).$$

Corollary C.4 guarantees that $\xi_n^u(s_n) \leq \frac{C}{s_n\sqrt{n}}$ for some $C > 0$ depending only on $\alpha, K, \overline{\overline{\Xi}}$. By the choice of $s_n$, we can finally upper bound (up to a positive constant depending only on $\alpha, K, \overline{\overline{\Xi}}, M$) the RHS above by

$$\frac{1}{s_n^2}\left(\frac{1}{\sqrt{ns_n}} + \frac{1}{\sqrt{s_n\sqrt{n}}}\right) \leq \frac{1}{s_n^{2.5}} \cdot \frac{2}{n^{1/4}} = 2 \cdot n^{-11/64} \leq 2 \cdot n^{-1/8},$$

for all sufficiently large $n$, which completes the proof. $\square$

Recalling $\langle \cdot \rangle_{n,t}$ defined in (C.6), let us identify $\mathbb{E}\left[\langle \widetilde{u}^\top \widetilde{u}^* \rangle_{n,t}\right]$ as a function of $(t, q_1, q_2)$:

$$\frac{1}{n}\mathbb{E}\left[\langle \widetilde{u}^\top \widetilde{u}^* \rangle_{n,t}\right] = \Delta(t, q_1, q_2). \tag{C.44}$$

Note that $\Delta$ is continuous, non-negative (by Nishimori identity) on $[0, 1] \times \mathbb{R}_{\geq 0}^2$ and bounded by $K^2$. Its partial derivatives with respect to the second and third arguments are continuous and non-negative, since the correlation between $\widetilde{u}^*$ and $\langle \widetilde{u} \rangle_{n,t}$ is a non-decreasing function of the SNRs $q_1, q_2$.

**Lemma C.8** (Fundamental sum rule). *In the setting of Lemma C.7, for $(\varepsilon_1, \varepsilon_2) \in [1, 2]^2$, let $q_1(t, \varepsilon_1, \varepsilon_2), q_2(t, \varepsilon_1, \varepsilon_2)$ be the solution to (C.39) with $R_2 = \Delta$ defined in (C.44). Then we have*

$$\mathcal{F}_n = \int_1^2 \int_1^2 \int_0^1 \psi_{\overline{\Xi}}(\alpha\gamma q_1(1, \varepsilon_1, \varepsilon_2)) + \alpha\psi_{\overline{\Sigma}}(q_2(1, \varepsilon_1, \varepsilon_2)) - \frac{\alpha}{2}q_1'(t, \varepsilon_1, \varepsilon_2)q_2'(t, \varepsilon_1, \varepsilon_2) \, dt \, d\varepsilon_1 \, d\varepsilon_2 + o(1).$$

*Proof.* Fix $(\varepsilon_1, \varepsilon_2) \in [1, 2]^2$. By the choice $R_2 = \Delta$,

$$q_2'(t, \varepsilon_1, \varepsilon_2) = \Delta(t, q_1(t, \varepsilon_1, \varepsilon_2), q_2(t, \varepsilon_1, \varepsilon_2)) = \frac{1}{n}\mathbb{E}\left[\langle \widetilde{u}^\top \widetilde{u}^* \rangle_{n,t}\right].$$

Plugging this into (C.14), integrating over $(\varepsilon_1, \varepsilon_2) \in [1, 2]^2$ and applying Lemma C.7, we have

$$f_n'(t) = \frac{\alpha}{2}q_1'(t)q_2'(t) + o(1),$$

where $o(1) \to 0$ as $n \to \infty$, uniformly over $t$. By Lemma C.1, we conclude

$$\mathcal{F}_n = \int_1^2 \int_1^2 f_n(0) \, d\varepsilon_1 \, d\varepsilon_2 + o(1) = \int_1^2 \int_1^2 \left(f_n(1) - \int_0^1 f_n'(t) \, dt\right) d\varepsilon_1 \, d\varepsilon_2 + o(1)$$

$$= \int_1^2 \int_1^2 \int_0^1 \psi_{\overline{\Xi}}(\alpha\gamma q_1(1, \varepsilon_1, \varepsilon_2)) + \alpha\psi_{\overline{\Sigma}}(q_2(1, \varepsilon_1, \varepsilon_2)) - \frac{\alpha}{2}q_1'(t, \varepsilon_1, \varepsilon_2)q_2'(t, \varepsilon_1, \varepsilon_2) \, dt \, d\varepsilon_1 \, d\varepsilon_2 + o(1),$$

as desired. Here the first and last equalities are by (C.8) and (C.9), respectively. $\square$

Finally, we prove a pair of matching upper and lower bounds, completing the proof of Theorem 4.4.

**Lemma C.9** (Lower bound). *Let $s_n = n^{-1/32}$ and $R_2 = \Delta$. Then*

$$\liminf_{n \to \infty} \mathcal{F}_n \geq \sup_{q_v \geq 0} \inf_{q_u \geq 0} \mathcal{F}(q_u, q_v).$$

*Proof.* Fix an arbitrary $q_v \geq 0$. Let $R_1 = q_v$. Then $q_1(t, \varepsilon_1, \varepsilon_2) = s_n\varepsilon_1 + tq_v$ and Lemma C.8 gives

$$\mathcal{F}_n = \int_1^2 \int_1^2 \int_0^1 \psi_{\overline{\Xi}}(\alpha\gamma(s_n\varepsilon_1 + q_v)) + \alpha\psi_{\overline{\Sigma}}(q_2(1, \varepsilon_1, \varepsilon_2)) - \frac{\alpha}{2}q_v q_2'(t, \varepsilon_1, \varepsilon_2) \, dt \, d\varepsilon_1 \, d\varepsilon_2 + o(1)$$

$$= \int_1^2 \int_1^2 \psi_{\overline{\Xi}}(\alpha\gamma q_v) + \alpha\psi_{\overline{\Sigma}}(q_2(1, \varepsilon_1, \varepsilon_2)) - \frac{\alpha}{2}q_v q_2(1, \varepsilon_1, \varepsilon_2) \, d\varepsilon_1 \, d\varepsilon_2 + o(1)$$

$$\geq \inf_{q_2 \geq 0} \psi_{\overline{\Xi}}(\alpha\gamma q_v) + \alpha\psi_{\overline{\Sigma}}(q_2) - \frac{\alpha}{2}q_v q_2 + o(1) = \inf_{q_u \geq 0} \mathcal{F}(q_u, q_v) + o(1),$$

where the second line holds since $\psi_{\overline{\Xi}}$ is Lipschitz and $q_2(0, \varepsilon_1, \varepsilon_2) = s_n\varepsilon_2 = o(1)$. This completes the proof since the above lower bound holds for all $q_v \geq 0$. $\square$

**Lemma C.10** (Upper bound). *Let $s_n = n^{-1/32}$ and $R_2 = \Delta$. Then*

$$\limsup_{n \to \infty} \mathcal{F}_n \leq \sup_{q_v \geq 0} \inf_{q_u \geq 0} \mathcal{F}(q_u, q_v).$$

*Proof.* We apply Lemma C.8 with

$$R_1(t, q_1, q_2) = 2\alpha\psi'_{\overline{\Sigma}}(\Delta(t, q_1, q_2)). \tag{C.45}$$

Since $\psi_{\overline{\Xi}}$ is Lipschitz and convex,

$$\psi_{\overline{\Xi}}(\alpha\gamma q_1(1, \varepsilon_1, \varepsilon_2)) = \psi_{\overline{\Xi}}(\alpha\gamma(q_1(1, \varepsilon_1, \varepsilon_2) - q_1(0, \varepsilon_1, \varepsilon_2))) + o(1) = \psi_{\overline{\Xi}}\left(\alpha\gamma \int_0^1 q_1'(t, \varepsilon_1, \varepsilon_2)\, \mathrm{d}t\right) + o(1)$$

$$\leq \int_0^1 \psi_{\overline{\Xi}}(\alpha\gamma q_1'(t, \varepsilon_1, \varepsilon_2))\, \mathrm{d}t + o(1),$$

and similarly

$$\psi_{\overline{\Sigma}}(q_2(1, \varepsilon_1, \varepsilon_2)) \leq \int_0^1 \psi_{\overline{\Sigma}}(q_2'(t, \varepsilon_1, \varepsilon_2))\, \mathrm{d}t + o(1).$$

Now Lemma C.8 implies

$$\mathcal{F}_n \leq \int_1^2 \int_1^2 \int_0^1 \psi_{\overline{\Xi}}(\alpha\gamma q_1'(t, \varepsilon_1, \varepsilon_2)) + \alpha\psi_{\overline{\Sigma}}(q_2'(t, \varepsilon_1, \varepsilon_2)) - \frac{\alpha}{2} q_1'(t, \varepsilon_1, \varepsilon_2) q_2'(t, \varepsilon_1, \varepsilon_2)\, \mathrm{d}t\, \mathrm{d}\varepsilon_1\, \mathrm{d}\varepsilon_2 + o(1)$$

$$= \int_1^2 \int_1^2 \int_0^1 \mathcal{G}(q_2'(t, \varepsilon_1, \varepsilon_2), q_1'(t, \varepsilon_1, \varepsilon_2))\, \mathrm{d}t\, \mathrm{d}\varepsilon_1\, \mathrm{d}\varepsilon_2 + o(1),$$

where

$$\mathcal{G}(q_u, q_v) := \psi_{\overline{\Xi}}(\alpha\gamma q_v) + \alpha\psi_{\overline{\Sigma}}(q_u) - \frac{\alpha}{2} q_u q_v.$$

With the choice of $R_1$ in (C.45) and $R_2 = \Delta$, the ODE in Lemma C.7 gives

$$q_1'(t, \varepsilon_1, \varepsilon_2) = 2\alpha\psi'_{\overline{\Sigma}}(q_2'(t, \varepsilon_1, \varepsilon_2)),$$

which corresponds to the criticality condition of $\mathcal{G}$ with respect to $q_u$:

$$\partial_1 \mathcal{G}(q_2'(t, \varepsilon_1, \varepsilon_2), q_1'(t, \varepsilon_1, \varepsilon_2)) = 0.$$

Since $\psi_{\overline{\Sigma}}$ is convex and $-\frac{\alpha}{2} q_u q_v$ is linear in $q_2$, we have that $\mathcal{G}$ is convex in $q_u$. Therefore

$$\mathcal{G}(q_2'(t, \varepsilon_1, \varepsilon_2), q_1'(t, \varepsilon_1, \varepsilon_2)) = \inf_{q_u \geq 0} \mathcal{G}(q_u, q_1'(t, \varepsilon_1, \varepsilon_2)) = \inf_{q_u \geq 0} \mathcal{F}(q_u, q_1'(t, \varepsilon_1, \varepsilon_2)) \leq \sup_{q_v \geq 0} \inf_{q_u \geq 0} \mathcal{F}(q_u, q_v),$$

which completes the proof. $\qquad\square$

# D  Proofs of Theorem 4.2 and Corollary 4.3

## D.1  Proof of (4.7)

We compute the derivative of $\mathcal{F}_n(\gamma)$:

$$\mathcal{F}_n'(\gamma) = \frac{1}{n}\mathbb{E}\left[\frac{\mathcal{Z}_n'(\gamma)}{\mathcal{Z}_n(\gamma)}\right] = \frac{1}{n}\mathbb{E}\left[\left\langle \frac{1}{2\sqrt{\gamma n}}\widetilde{u}^\top Z\widetilde{v} + \frac{1}{n}\widetilde{u}^\top \widetilde{u}^*\widetilde{v}^\top \widetilde{v}^* - \frac{1}{2n}\widetilde{u}^\top \widetilde{u}\widetilde{v}^\top \widetilde{v}\right\rangle_n\right]$$

$$= \frac{1}{2n^2}\mathbb{E}\left[\langle \widetilde{u}^\top \widetilde{u}^*\widetilde{v}^\top \widetilde{v}^* \rangle_n\right], \tag{D.1}$$

where the last step follows similar calculations in (C.16). Since $\mathcal{F}_n(\gamma) \to \sup_{q_v} \inf_{q_u} \mathcal{F}(q_u, q_v)$ as $n \to \infty$, we have $\mathcal{F}_n'(\gamma) \to \frac{\partial}{\partial\gamma} \sup_{q_v} \inf_{q_u} \mathcal{F}(q_u, q_v)$. To compute the RHS, note that

$$\sup_{q_v \geq 0} \inf_{q_u \geq 0} \mathcal{F}(q_u, q_v) = \sup_{q_v \geq 0}\left\{\psi_{\overline{\Xi}}(\alpha\gamma q_v) + \inf_{q_u \geq 0}\left\{\alpha\psi_{\overline{\Sigma}}(\gamma q_u) - \frac{\alpha}{2}\gamma q_u q_v\right\}\right\},$$

and the value of the infimum does not depend on $\gamma$. Therefore, we have

$$\frac{\partial}{\partial \gamma} \sup_{q_v \geq 0} \inf_{q_u \geq 0} \mathcal{F}(q_u, q_v) = \psi'_{\overline{\Xi}}(\alpha \gamma q_v^*) \alpha q_v^* = \frac{\alpha q_u^* q_v^*}{2}, \tag{D.2}$$

where first equality is by the envelope theorem from [51, Corollary 4] and the last equality follows since the extremizers $q_u^*, q_v^*$ solve a pair of equations in (4.16).

On the other hand, we relate $\mathcal{F}'_n(\gamma)$ to the MMSE (4.2) as follows:

$$\mathrm{MMSE}_n(\gamma) = \frac{1}{nd} \mathbb{E}\Big[ \big\| \widetilde{u}^*(\widetilde{v}^*)^\top - \langle \widetilde{u}\widetilde{v}^\top \rangle_n \big\|_{\mathrm{F}}^2 \Big] = \frac{1}{nd} \mathbb{E}\Big[ \|\widetilde{u}^*\|_2^2 \|\widetilde{v}^*\|_2^2 + \big\| \langle \widetilde{u}\widetilde{v}^\top \rangle_n \big\|_{\mathrm{F}}^2 - 2(\widetilde{u}^*)^\top \langle \widetilde{u}\widetilde{v}^\top \rangle_n \widetilde{v}^* \Big]$$

$$= \frac{\mathrm{Tr}(\Xi^{-1})}{n} \frac{\mathrm{Tr}(\Sigma^{-1})}{d} - \frac{1}{nd} \mathbb{E}\big[ \langle \widetilde{u}^\top \widetilde{u}^* \widetilde{v}^\top \widetilde{v}^* \rangle_n \big], \tag{D.3}$$

where the last step is by Nishimori identity (Proposition G.4). Combining the above with (D.1) and (D.2), we conclude

$$\mathrm{MMSE}_n(\gamma) \to \mathbb{E}\Big[\overline{\Xi}^{-1}\Big] \mathbb{E}\Big[\overline{\Sigma}^{-1}\Big] - q_u^* q_v^*,$$

as claimed.

### D.2 Proof of (4.8)

Recall $Y$ from (4.1) and define for some $\gamma' \geq 0$,

$$Y' := \sqrt{\frac{\gamma'}{n}} u^* u^{*\top} + \Xi^{1/2} Z' \Xi^{1/2}, \tag{D.4}$$

where $Z' \in \mathbb{R}^{n \times n}$ is a symmetric random matrix independent of $u^*, v^*$ with $Z'_{i,i} \overset{\text{i.i.d.}}{\sim} \mathcal{N}(0,2)$ and $Z'_{i,j} \overset{\text{i.i.d.}}{\sim} \mathcal{N}(0,1)$ for all $1 \leq i < j \leq n$. By similar derivations as before, the free energy associated with $(Y, Y')$ is given by

$$\mathsf{F}_n(\gamma, \gamma') = \frac{1}{n} \mathbb{E}\Bigg[ \log \int_{\mathbb{R}^d} \int_{\mathbb{R}^n} \exp\bigg( \mathcal{H}_n(\Xi^{-1/2} u, \Sigma^{-1/2} v)$$

$$+ \frac{1}{2}\sqrt{\frac{\gamma'}{n}} u^\top \Xi^{-1} Y' \Xi^{-1} u - \frac{\gamma'}{4n}(u^\top \Xi^{-1} u)^2 \bigg) \, \mathrm{d}P^{\otimes n}(u) \, \mathrm{d}Q^{\otimes n}(v) \Bigg],$$

where $\mathcal{H}_n$ is given in (4.12). Denote by $\langle\!\langle \cdot \rangle\!\rangle_n$ the Gibbs bracket with respect to the corresponding Hamiltonian. Let

$$\mathsf{F}(\gamma, \gamma') := \sup_{q_u, q_v \geq 0} \frac{\gamma'}{4} q_u^2 + \frac{\alpha \gamma}{2} q_u q_v - \psi^*_{\overline{\Xi}}\Big(\frac{q_u}{2}\Big) - \alpha \psi^*_{\overline{\Sigma}}\Big(\frac{q_v}{2}\Big), \tag{D.5}$$

where $f^*$ denotes the monotone conjugate of a convex non-decreasing function $f : \mathbb{R}_{\geq 0} \to \mathbb{R}$; see Definition G.1. Basic properties of monotone conjugate can be found in [67, §12].

The following lemma, proved in Appendix D.3, characterizes the high-dimensional limit of $\mathsf{F}_n(\gamma, \gamma')$.

**Lemma D.1.** *For all $\gamma, \gamma' \geq 0$,*

$$\lim_{n \to \infty} \mathsf{F}_n(\gamma, \gamma') = \mathsf{F}(\gamma, \gamma'),$$

Let us show how (4.8) can be derived from Lemma D.1.

*Proof of (4.8).* Let

$$\mathrm{MMSE}_n^u(\gamma, \gamma') := \frac{1}{n^2} \mathbb{E}\Big[ \big\| \widetilde{u}^*(\widetilde{u}^*)^\top - \mathbb{E}\big[ \widetilde{u}^*(\widetilde{u}^*)^\top \mid Y, Y' \big] \big\|_{\mathrm{F}}^2 \Big].$$

Following similar derivations as in Appendix D.1, one can verify the following two identities:

$$\partial_2 \mathsf{F}_n(\gamma, \gamma') = \frac{1}{4n^2} \mathbb{E}\Big[ \big\langle\!\big\langle (\widetilde{u}^\top \widetilde{u}^*)^2 \big\rangle\!\big\rangle_n \Big], \quad \mathrm{MMSE}_n^u(\gamma, \gamma') = \frac{\mathrm{Tr}(\Xi^{-1})^2}{n^2} - \frac{1}{n^2} \mathbb{E}\Big[ \big\langle\!\big\langle (\widetilde{u}^\top \widetilde{u}^*)^2 \big\rangle\!\big\rangle_n \Big]. \tag{D.6}$$

Therefore, the MMSE in (4.3) can be written as

$$\mathrm{MMSE}_n^u(\gamma) = \lim_{\gamma' \downarrow 0} \mathrm{MMSE}_n^u(\gamma, \gamma') = \frac{\mathrm{Tr}(\Xi^{-1})^2}{n^2} - 4 \lim_{\gamma' \downarrow 0} \partial_2 \mathsf{F}_n(\gamma, \gamma').$$

By Lemma D.1 and Proposition G.11,

$$\limsup_{n \to \infty} \lim_{\gamma' \downarrow 0} \partial_2 \mathsf{F}_n(\gamma, \gamma') \le \lim_{\gamma' \downarrow 0} \partial_2 \mathsf{F}(\gamma, \gamma').$$

The envelope theorem from [51, Corollary 4] allows us to compute the RHS:

$$\lim_{\gamma' \downarrow 0} \partial_2 \mathsf{F}(\gamma, \gamma') = \frac{q_u^{*2}}{4},$$

where $(q_u^*, q_v^*)$ are the maximizer of

$$\sup_{q_u, q_v \ge 0} \frac{\alpha}{2} \gamma q_u q_v - \psi_{\overline{\Xi}}^* \left( \frac{q_u}{2} \right) - \alpha \psi_{\overline{\Sigma}}^* \left( \frac{q_v}{2} \right) = \sup_{(q_u, q_v) \in \mathcal{C}(\gamma, \alpha)} \psi_{\overline{\Xi}}(\alpha \gamma q_v) + \alpha \psi_{\overline{\Sigma}}(\gamma q_u) - \frac{\alpha \gamma}{2} q_u q_v,$$

where the equality is by Proposition G.12. Putting the above together, we have

$$\limsup_{n \to \infty} \frac{1}{n^2} \mathbb{E}\left[ \left\langle (\widetilde{u}^\top \widetilde{u}^*)^2 \right\rangle_n \right] = \limsup_{n \to \infty} \lim_{\gamma' \downarrow 0} \frac{1}{n^2} \mathbb{E}\left[ \left\langle\!\!\left\langle (\widetilde{u}^\top \widetilde{u}^*)^2 \right\rangle\!\!\right\rangle_n \right] \le q_u^{*2},$$
$$\liminf_{n \to \infty} \mathrm{MMSE}_n^u(\gamma) \ge \mathbb{E}\left[ \overline{\Xi}^{-1} \right]^2 - q_u^{*2}. \tag{D.7}$$

By a symmetric argument, we also have

$$\limsup_{d \to \infty} \lim_{\gamma' \downarrow 0} \frac{1}{d^2} \mathbb{E}\left[ \left\langle (\widetilde{v}^\top \widetilde{v}^*)^2 \right\rangle_n \right] \le q_v^{*2}. \tag{D.8}$$

To find an upper bound on $\mathrm{MMSE}_n^u(\gamma)$, note that by (4.7) and (D.3):

$$(q_u^* q_v^*)^2 = \lim_{n \to \infty} \frac{1}{(nd)^2} \mathbb{E}\left[ \langle \widetilde{u}^\top \widetilde{u}^* \widetilde{v}^\top \widetilde{v}^* \rangle_n \right]^2 \le \liminf_{n \to \infty} \left( \frac{1}{n^2} \mathbb{E}\left[ \left\langle (\widetilde{u}^\top \widetilde{u}^*)^2 \right\rangle_n \right] \right) \left( \frac{1}{d^2} \mathbb{E}\left[ \left\langle (\widetilde{v}^\top \widetilde{v}^*)^2 \right\rangle_n \right] \right).$$

This combined with (D.7) and (D.8) implies

$$\lim_{n \to \infty} \frac{1}{n^2} \mathbb{E}\left[ \left\langle (\widetilde{u}^\top \widetilde{u}^*)^2 \right\rangle_n \right] = q_u^{*2}, \quad \lim_{d \to \infty} \frac{1}{d^2} \mathbb{E}\left[ \left\langle (\widetilde{v}^\top \widetilde{v}^*)^2 \right\rangle_n \right] = q_v^{*2},$$

which concludes the proof in view of the relation

$$\mathrm{MMSE}_n^u(\gamma) = \frac{\mathrm{Tr}(\Xi^{-1})^2}{n^2} - \frac{1}{n^2} \mathbb{E}\left[ \left\langle (\widetilde{u}^\top \widetilde{u}^*)^2 \right\rangle_n \right]. \qquad \square$$

## D.3 Proof of Lemma D.1

We assume $\gamma = \gamma' = 1$ by formally absorbing them into $P, Q$:

$$\int_{\mathbb{R}} x^2 \, \mathrm{d}P(x) = \sqrt{\gamma'}, \quad \int_{\mathbb{R}} x^2 \, \mathrm{d}Q(x) = \frac{\gamma}{\sqrt{\gamma'}},$$

so that we can drop the dependence on $\gamma, \gamma'$ in notation such as $\mathsf{F}_n, \mathsf{F}$. We then truncate $P, Q$ at a sufficiently large constant $K > 0$ so that they have bounded supports.

Recall $Y$ from (4.1) and define for $r \ge 0$, $Y'' := \sqrt{r} u^* + \Xi^{1/2} Z''$ where $Z'' \sim \mathcal{N}(0_n, I_n)$ is independent of everything else. The free energy $\widetilde{\mathsf{F}}_n$ associated with $(Y, Y'')$ is

$$\widetilde{\mathsf{F}}_n = \frac{1}{n} \mathbb{E}\left[ \log \int_{\mathbb{R}^d} \int_{\mathbb{R}^n} \exp\left( \mathcal{H}_n(\Xi^{-1/2} u, \Sigma^{-1/2} v) + r u^\top \Xi^{-1} u^* + \sqrt{r} u^\top \Xi^{-1/2} Z'' - \frac{r}{2} u^\top \Xi^{-1} u \right) \mathrm{d}P^{\otimes n}(u) \, \mathrm{d}Q^{\otimes d}(v) \right],$$

where $\mathcal{H}_n$ is given in (4.12). A straightforward adaptation of the proof of Theorem 4.4 yields a characterization of the limit of $\widetilde{\mathsf{F}}_n$. Let

$$\widetilde{\mathsf{F}}(r) := \sup_{q_v \ge 0} \inf_{q_u \ge 0} \psi_{\overline{\Xi}}\left( \sqrt{\gamma'}(\alpha q_v + r) \right) + \alpha \psi_{\overline{\Sigma}}\left( \frac{\gamma}{\sqrt{\gamma'}} q_u \right) - \frac{\alpha}{2} q_u q_v. \tag{D.9}$$

**Lemma D.2.** *For all $r \geq 0$,*

$$\lim_{n \to \infty} \widetilde{\mathsf{F}}_n(r) = \widetilde{\mathsf{F}}(r).$$

*Proof.* To obtain the result, we execute the interpolation argument as in the proof of Theorem 4.4 on the Hamiltonian of the following interpolating models:

$$
\begin{aligned}
Y_t &= \sqrt{\frac{1-t}{n}} u^* v^{*\top} + \Xi^{1/2} Z \Sigma^{1/2}, \\
Y_t^u &= \sqrt{\alpha q_1(t)} u^* + \Xi^{1/2} Z^u, \\
Y_t^v &= \sqrt{q_2(t)} v^* + \Sigma^{1/2} Z^v, \\
Y'' &= \sqrt{r} u^* + \Xi^{1/2} Z''.
\end{aligned}
$$

All steps in the proof of Theorem 4.4 carry over. $\qquad\square$

The above lemma allows us to derive the following auxiliary characterization of $\mathsf{F}_n$.

**Lemma D.3.**

$$\lim_{n \to \infty} \mathsf{F}_n = \sup_{r \geq 0} \widetilde{\mathsf{F}}(r) - \frac{r^2}{4}. \tag{D.10}$$

*Proof.* Let $r \colon [0,1] \to \mathbb{R}_{>0}$ be a differentiable function. For $t \in [0,1]$, consider $(Y, Y_t', Y_t'')$ where $Y$ is given in (4.1) and $Y_t', Y_t''$ are defined as

$$Y_t' = \sqrt{\frac{1-t}{n}} u^* u^{*\top} + \Xi^{1/2} Z' \Xi^{1/2}, \quad Y_t'' = \sqrt{r(t)} u^* + \Xi^{1/2} Z'',$$

with $Z'' \sim \mathcal{N}(0_n, I_n)$ independent of everything else.

Similar to (C.5) and (C.6), denote by

$$
\begin{aligned}
\mathsf{f}_n(t) := \frac{1}{n} \mathbb{E}\Bigg[ \int_{\mathbb{R}^d} \int_{\mathbb{R}^n} \exp\Bigg( &\sqrt{\frac{1}{n}} \widetilde{u}^\top Z \widetilde{v} + \frac{1}{n} \widetilde{u}^\top \widetilde{u}^* \widetilde{v}^\top \widetilde{v}^* - \frac{1}{2n} \|\widetilde{u}\|_2^2 \|\widetilde{v}\|_2^2 \\
&+ \frac{1}{2} \sqrt{\frac{1-t}{n}} \widetilde{u}^\top Z' \widetilde{u} + \frac{1-t}{2n} (\widetilde{u}^\top \widetilde{u}^*)^2 - \frac{1-t}{4n} \|\widetilde{u}\|_2^4 \\
&+ r(t) \widetilde{u}^\top \widetilde{u}^* + \sqrt{r(t)} \widetilde{u}^\top Z'' - \frac{r(t)}{2} \|\widetilde{u}\|_2^2 \Bigg) \, \mathrm{d}\widetilde{P}(\widetilde{u}) \, \mathrm{d}\widetilde{Q}(\widetilde{v}) \Bigg]
\end{aligned}
$$

the free energy associated with $(Y, Y_t', Y_t'')$ and by $\langle\!\langle \cdot \rangle\!\rangle_{n,t}$ the conditional expectation with respect to the corresponding Gibbs measure. The rest of the proof follows the skeleton of Theorem 4.4 and we only highlight the differences.

Parallel to Lemma C.1, we have that if $r(0) \geq 0$ and $\lim_{n \to \infty} r(0) = 0$, then

$$\mathsf{f}_n(0) = \mathsf{F}_n + \mathcal{O}(r(0)), \quad \mathsf{f}_n(1) = \widetilde{\mathsf{F}}_n(r(1)).$$

The analogue of Lemma C.2 gives

$$\mathsf{f}_n'(t) = -\frac{1}{4} \mathbb{E}\left[ \left\langle\!\left\langle \left( \frac{\widetilde{u}^\top \widetilde{u}^*}{n} - r'(t) \right)^2 \right\rangle\!\right\rangle_{n,t} \right] + \frac{r'(t)^2}{4}. \tag{D.11}$$

We now show a pair of matching lower and upper bounds on $\mathsf{F}_n$. First comes the lower bound. Using (D.11) with $r(t) = rt$ for a constant $r \geq 0$, we have

$$\mathsf{F}_n = \mathsf{f}_n(0) = \mathsf{f}_n(1) - \int_0^1 \mathsf{f}_n'(t) \, \mathrm{d}t \geq \widetilde{\mathsf{F}}_n(r) - \frac{r^2}{4}.$$

By Lemma D.2, this implies the lower bound

$$\liminf_{n\to\infty} \mathsf{F}_n \geq \sup_{r\geq 0} \widetilde{\mathsf{F}}(r) - \frac{r^2}{4}.$$

Next we show a matching upper bound. Let $\varepsilon \in [1,2]$ and slightly abusing notation, denote by $r(t;\varepsilon)$ the solution to

$$r'(t) = \frac{1}{n}\mathbb{E}\Big[\big\langle\!\big\langle \widetilde{u}^\top \widetilde{u}^* \big\rangle\!\big\rangle_{n,t}\Big], \quad r(0) = s_n \varepsilon,$$

where $s_n = n^{-1/32}$. The analogue of Lemma C.7 gives

$$\int_1^2 \frac{1}{n^2}\mathbb{E}\bigg[\Big\langle\!\Big\langle \Big(\widetilde{u}^\top \widetilde{u}^* - \mathbb{E}\big[\langle \widetilde{u}^\top \widetilde{u}^*\rangle_{n,t}\big]\Big)^2 \Big\rangle\!\Big\rangle_{n,t}\bigg]\, \mathrm{d}\varepsilon \leq \frac{C}{n^{1/8}}, \tag{D.12}$$

for a constant $C > 0$ independent of $n$. Using (D.11) with $r(t;\varepsilon)$, we have

$$\mathsf{F}_n = \int_1^2 \mathsf{f}_n(0)\,\mathrm{d}\varepsilon + o(1) = \int_1^2 \bigg(\widetilde{\mathsf{F}}_n(r(1,\varepsilon)) - \int_0^1 \frac{r'(t,\varepsilon)^2}{4}\,\mathrm{d}t\bigg)\,\mathrm{d}\varepsilon + o(1)$$

$$\leq \int_1^2 \int_0^1 \widetilde{\mathsf{F}}_n(r'(t,\varepsilon)) - \frac{r'(t,\varepsilon)^2}{4}\,\mathrm{d}t\,\mathrm{d}\varepsilon + o(1) \leq \sup_{r\geq 0} \widetilde{\mathsf{F}}_n(r) - \frac{r^2}{4} + o(1),$$

where the second equality is by (D.12) and the penultimate inequality holds since $\widetilde{\mathsf{F}}_n(\cdot)$ is convex and non-decreasing. Passing to the limit, we obtain the upper bound

$$\limsup_{n\to\infty} \mathsf{F}_n \leq \sup_{r\geq 0} \widetilde{\mathsf{F}}(r) - \frac{r^2}{4},$$

as desired. □

To establish Lemma D.1, it remains to verify that the RHSs of (D.5) and (D.10) are equal. We need the following lemma.

**Lemma D.4.** *Let $f, g\colon \mathbb{R}_{\geq 0} \to \mathbb{R}$ be non-decreasing, lower semi-continuous, convex functions with finite $f(0)$, $g(0)$ and monotone conjugates $f^*, g^*$ (see Definition G.1). Then*

$$\sup_{r\geq 0}\sup_{q_1\geq 0}\inf_{q_2\geq 0} f(q_1 + r) + g(q_2) - q_1 q_2 - \frac{r^2}{2} = \sup_{q_1,q_2\geq 0} \frac{q_2^2}{2} + q_1 q_2 - f^*(q_2) - g^*(q_1).$$

*Proof.* Writing $f_r(x) = f(x + r)$ and using Proposition G.12, we have

$$\sup_{q_1\geq 0}\inf_{q_2\geq 0} f(q_1 + r) + g(q_2) - q_1 q_2 = \sup_{q_1,q_2\geq 0} q_1 q_2 - f_r^*(q_2) - g^*(q_1)$$

$$= \sup_{q_1\geq 0} f_r(q_1) - g^*(q_1) = \sup_{q_1,q_2\geq 0} q_2(q_1 + r) - f^*(q_2) - g^*(q_1),$$

where we have used the fact that $f^{**} = f$. Therefore,

$$\sup_{r\geq 0}\sup_{q_1\geq 0}\inf_{q_2\geq 0} f(q_1 + r) + g(q_2) - q_1 q_2 - \frac{r^2}{2} = \sup_{q_1,q_2\geq 0}\bigg\{\sup_{r\geq 0}\Big\{q_2 r - \frac{r^2}{2}\Big\} + q_1 q_2 - f^*(q_2) - g^*(q_1)\bigg\}$$

$$= \sup_{q_1,q_2\geq 0} \frac{q_2^2}{2} + q_1 q_2 - f^*(q_2) - g^*(q_1),$$

as claimed. □

Applying Lemma D.4 immediately finishes the proof of Lemma D.1.

*Proof of Lemma D.1.* By Lemma D.4 and the definition (D.9),

$$\sup_{r\geq 0} \widetilde{\mathsf{F}}(r) - \frac{r^2}{4} = \sup_{q_u,q_v\geq 0} \frac{\gamma'}{4}q_u^2 + \frac{\alpha}{2}\gamma q_u q_v - \psi_{\Xi}^*\Big(\frac{q_u}{2}\Big) - \alpha\psi_{\Sigma}^*\Big(\frac{q_v}{2}\Big),$$

where we have used the fact that for $a, b \geq 0$, the monotone conjugate of $g(x) := bf(ax)$ is $g^*(x) = bf^*(x/(ab))$. □

### D.4 Proof of Corollary 4.3

Denote $M \coloneqq \mathbb{E}\left[u^* v^{*\top} \mid Y\right]$. By the Nishimori identity (Proposition G.4),

$$\lim_{n \to \infty} \frac{1}{nd} \mathbb{E}\left[\left\|\widetilde{u}^*(\widetilde{v}^*)^\top - \mathbb{E}[\widetilde{u}^*(\widetilde{v}^*)^\top \mid Y]\right\|_{\mathrm{F}}^2\right] = \mathbb{E}\left[\overline{\Xi}^{-1}\right]\mathbb{E}\left[\overline{\Sigma}^{-1}\right] - \lim_{n \to \infty} \frac{1}{nd} \mathbb{E}\left[\left\|\Xi^{-1/2} M \Sigma^{-1/2}\right\|_{\mathrm{F}}^2\right], \tag{D.13}$$

$$\lim_{n \to \infty} \frac{1}{nd} \mathbb{E}\left[\left\|u^* v^{*\top} - \mathbb{E}\left[u^* v^{*\top} \mid Y\right]\right\|_{\mathrm{F}}^2\right] = 1 - \lim_{n \to \infty} \frac{1}{nd} \mathbb{E}\left[\|M\|_{\mathrm{F}}^2\right]. \tag{D.14}$$

Theorem 4.2 and (D.13) imply that

$$\lim_{n \to \infty} \frac{1}{nd} \mathbb{E}\left[\left\|\Xi^{-1/2} M \Sigma^{-1/2}\right\|_{\mathrm{F}}^2\right] = q_u^* q_v^*, \tag{D.15}$$

which, by Proposition 4.1, is positive if and only if (4.6) holds.

We first show (4.10) assuming (4.6). Using assumption (3.3), super-multiplicativity of $\sigma_{\min}(\cdot)$, and the fact that $\|BC\|_{\mathrm{F}} \geq \|B\|_{\mathrm{F}} \sigma_{\min}(C)$, we have

$$\lim_{n \to \infty} \frac{1}{nd} \mathbb{E}\left[\|M\|_{\mathrm{F}}^2\right] \geq \lim_{n \to \infty} \frac{1}{nd} \sigma_n(\Xi^{1/2})^2 \mathbb{E}\left[\left\|\Xi^{-1/2} M \Sigma^{-1/2}\right\|_{\mathrm{F}}^2\right] \sigma_d(\Sigma^{1/2})^2$$

$$= (\inf \operatorname{supp}(\overline{\Xi}))(\inf \operatorname{supp}(\overline{\Sigma})) \lim_{n \to \infty} \frac{1}{nd} \mathbb{E}\left[\left\|\Xi^{-1/2} M \Sigma^{-1/2}\right\|_{\mathrm{F}}^2\right],$$

which is positive by (D.15). This combined with (D.14) implies (4.10).

We then show (4.10) with $<$ replaced with $=$, assuming that (4.6) is reversed. Using assumption (3.3), sub-multiplicativity of spectral norm, and the fact that $\|BC\|_{\mathrm{F}} \leq \|B\|_{\mathrm{F}} \|C\|_2$, we have

$$\lim_{n \to \infty} \frac{1}{nd} \mathbb{E}\left[\|M\|_{\mathrm{F}}^2\right] \leq \lim_{n \to \infty} \frac{1}{nd} \left\|\Xi^{1/2}\right\|_2^2 \mathbb{E}\left[\left\|\Xi^{-1/2} M \Sigma^{-1/2}\right\|_{\mathrm{F}}^2\right] \left\|\Sigma^{1/2}\right\|_2^2$$

$$= (\sup \operatorname{supp}(\overline{\Xi}))(\sup \operatorname{supp}(\overline{\Sigma})) \lim_{n \to \infty} \frac{1}{nd} \mathbb{E}\left[\left\|\Xi^{-1/2} M \Sigma^{-1/2}\right\|_{\mathrm{F}}^2\right],$$

which is 0 by (D.15). This combined with (D.14) finishes the proof of the corollary for estimating $u^* v^{*\top}$. The proofs for estimating $u^* u^{*\top}, v^* v^{*\top}$ are similar and omitted.

## E  Analysis of the spectral estimator

### E.1  Bayes-AMP

We propose an AMP algorithm that operates on $\Xi^{-1} A \Sigma^{-1}$ and maintains a pair of iterates $u^t \in \mathbb{R}^n, v^{t+1} \in \mathbb{R}^d$ for every $t \geq 0$. Specifically, given for every $t \geq 0$ a pair of denoising functions[2] $g_t \colon \mathbb{R}^n \to \mathbb{R}^n, f_{t+1} \colon \mathbb{R}^d \to \mathbb{R}^d$, the iterates are initialized at $\widetilde{u}^{-1} = 0_n$ and some $\widetilde{v}^0 \in \mathbb{R}^d$ of user's choice, and are updated for every $t \geq 0$ according to the following rules:

$$u^t = \Xi^{-1} A \Sigma^{-1} \widetilde{v}^t - b_t \Xi^{-1} \widetilde{u}^{t-1}, \quad \widetilde{u}^t = g_t(u^t), \quad c_t = \frac{1}{n} \operatorname{Tr}((\nabla g_t(u^t))\Xi^{-1}),$$

$$v^{t+1} = \Sigma^{-1} A^\top \Xi^{-1} \widetilde{u}^t - c_t \Sigma^{-1} \widetilde{v}^t, \quad \widetilde{v}^{t+1} = f_{t+1}(v^{t+1}), \quad b_{t+1} = \frac{1}{n} \operatorname{Tr}((\nabla f_{t+1}(v^{t+1}))\Sigma^{-1}),$$

$$\tag{E.1}$$

where $\nabla g_t(u^t) \in \mathbb{R}^{n \times n}, \nabla f_{t+1}(v^{t+1}) \in \mathbb{R}^{d \times d}$ denote the Jacobians of $g_t, f_{t+1}$ at $u^t, v^{t+1}$, respectively. For any fixed $t \geq 0$, the $n, d \to \infty$ limit of the iterates $u^t, v^{t+1}$ can be described by a deterministic recursion known as the state evolution. To define the latter, we need a sequence of preliminary definitions.

---

[2]Strictly speaking, for every $t \geq 0$, we are given two sequences of functions $g_t, f_{t+1}$ indexed by $n, d$ respectively. See Definition E.1 for a formal treatment of function sequences.

First, define random vectors $(U^*, W_{U,0}, W_{U,1}, \cdots, W_{U,t}) \in (\mathbb{R}^n)^{t+2}$ and $(V^*, W_{V,1}, W_{V,2}, \cdots, W_{V,t+1}) \in (\mathbb{R}^d)^{t+2}$ with joint distributions specified below:

$$
\begin{bmatrix} U^* \\ \sigma_0 W_{U,0} \\ \sigma_1 W_{U,1} \\ \vdots \\ \sigma_t W_{U,t} \end{bmatrix} \sim P^{\otimes n} \otimes \mathcal{N}(0_{n(t+1)}, \Phi_t \otimes I_n), \qquad \begin{bmatrix} V^* \\ \tau_1 W_{V,1} \\ \tau_2 W_{V,2} \\ \vdots \\ \tau_{t+1} W_{V,t+1} \end{bmatrix} \sim Q^{\otimes d} \otimes \mathcal{N}(0_{d(t+1)}, \Psi_{t+1} \otimes I_d),
$$

(E.2)

where we recall that for $A \in \mathbb{R}^{m \times n}, B \in \mathbb{R}^{p \times q}$, their Kronecker product is

$$
A \otimes B = \begin{bmatrix} A_{1,1}B & \cdots & A_{1,n}B \\ \vdots & \ddots & \vdots \\ A_{m,1}B & \cdots & A_{m,n}B \end{bmatrix} \in \mathbb{R}^{mp \times nq}.
$$

The covariance matrices $\Phi_t = (\Phi_{r,s})_{0 \le r,s \le t}, \Psi_{t+1} = (\Psi_{r+1,s+1})_{0 \le r,s \le t} \in \mathbb{R}^{(t+1) \times (t+1)}$ are given by the $(t+1) \times (t+1)$ principal minors of two infinite-dimensional symmetric matrices $\Phi := (\Phi_{r,s})_{r,s \ge 0}, \Psi := (\Psi_{r+1,s+1})_{r,s \ge 0}$ whose elements are in turn given recursively below:

$$
\Phi_{0,0} = \operatorname*{p-lim}_{n \to \infty} \frac{1}{n} (\widetilde{v}^0)^\top \Sigma^{-1} \widetilde{v}^0,
$$

$$
\Phi_{0,s} = \lim_{n \to \infty} \frac{1}{n} \mathbb{E}\big[ f_0(V^*)^\top \Sigma^{-1} f_s(V_s) \big], \quad s \ge 1,
$$

(E.3)

$$
\Phi_{r,s} = \lim_{n \to \infty} \frac{1}{n} \mathbb{E}\big[ f_r(V_r)^\top \Sigma^{-1} f_s(V_s) \big], \quad r,s \ge 1,
$$

$$
\Psi_{r+1,s+1} = \lim_{n \to \infty} \frac{1}{n} \mathbb{E}\big[ g_r(U_r)^\top \Xi^{-1} g_s(U_s) \big], \quad r,s \ge 0.
$$

Furthermore, for $t \ge 0, \sigma_t, \tau_{t+1} > 0$ are defined as

$$
\sigma_0^2 := \Phi_{0,0} = \operatorname*{p-lim}_{n \to \infty} \frac{1}{n} (\widetilde{v}^0)^\top \Sigma^{-1} \widetilde{v}^0,
$$

$$
\sigma_t^2 := \Phi_{t,t} = \lim_{n \to \infty} \frac{1}{n} \mathbb{E}\big[ f_t(V_t)^\top \Sigma^{-1} f_t(V_t) \big], \quad t \ge 1,
$$

(E.4)

$$
\tau_{t+1}^2 := \Psi_{t+1,t+1} = \lim_{n \to \infty} \frac{1}{n} \mathbb{E}\big[ g_t(U_t)^\top \Xi^{-1} g_t(U_t) \big], \quad t \ge 0.
$$

With the above definitions, note that each $W_{U,t}, W_{V,t+1}$ is marginally distributed as $\mathcal{N}(0_n, I_n), \mathcal{N}(0_d, I_d)$, respectively.

Next, define two sequences of deterministic scalars $(\mu_t, \nu_{t+1})_{t \ge 0}$:

$$
\mu_0 = \lim_{n \to \infty} \frac{\lambda}{n} \mathbb{E}\big[ \langle \Sigma^{-1} V^*, f_0(V^*) \rangle \big],
$$

$$
\mu_t = \lim_{n \to \infty} \frac{\lambda}{n} \mathbb{E}\big[ \langle \Sigma^{-1} V^*, f_t(V_t) \rangle \big], \quad t \ge 1,
$$

(E.5)

$$
\nu_{t+1} = \lim_{n \to \infty} \frac{\lambda}{n} \mathbb{E}\big[ \langle \Xi^{-1} U^*, g_t(U_t) \rangle \big], \quad t \ge 0,
$$

where $f_0$ is determined by the initializer $\widetilde{v}^0$; see Assumption (A1) below.

With these, for $t \ge 0$, define random vectors

$$
U_t = \mu_t \Xi^{-1} U^* + \sigma_t \Xi^{-1/2} W_{U,t}, \qquad V_{t+1} = \nu_{t+1} \Sigma^{-1} V^* + \tau_{t+1} \Sigma^{-1/2} W_{V,t+1}.
$$

(E.6)

Finally, we need the notion of (uniformly) pseudo-Lipschitz functions.

**Definition E.1** (Pseudo-Lipschitz functions). A function $\phi \colon \mathbb{R}^{k \times m} \to \mathbb{R}^{\ell \times m}$ is called pseudo-Lipschitz of order $j \ge 1$ if there exists $L > 0$ such that

$$
\frac{1}{\sqrt{\ell}} \|\phi(x) - \phi(y)\|_{\mathrm{F}} \le \frac{L}{\sqrt{k}} \|x - y\|_{\mathrm{F}} \left[ 1 + \left( \frac{1}{\sqrt{k}} \|x\|_{\mathrm{F}} \right)^{j-1} + \left( \frac{1}{\sqrt{k}} \|y\|_{\mathrm{F}} \right)^{j-1} \right],
$$

(E.7)

for every $x, y \in \mathbb{R}^{k \times m}$.

We will consider sequences of functions $\phi_i \colon \mathbb{R}^{k_i \times m} \to \mathbb{R}^{\ell_i \times m}$ indexed by $i \to \infty$ though the index $i$ is often suppressed. A sequence of functions $(\phi_i \colon \mathbb{R}^{k_i \times m} \to \mathbb{R}^{\ell_i \times m})_{i \geq 1}$ (with increasing dimensions $(k_i)_{i \geq 1}, (\ell_i)_{i \geq 1}$) is called uniformly pseudo-Lipschitz of order $j$ if there exists a constant $L$ such that for every $i \geq 1$, (E.7) holds.

The assumptions below are imposed on the initializer $\widetilde{v}^0$ and denoising function $(g_t, f_{t+1})_{t \geq 0}$.

(A1) $\widetilde{v}^0$ is independent of $\widetilde{W}$ but may depend on $v^*$.[3] Assume that

$$\operatorname*{p\text{-}lim}_{d \to \infty} \frac{1}{d}\big\|\widetilde{v}^0\big\|_2^2, \qquad \operatorname*{p\text{-}lim}_{d \to \infty} \frac{1}{n}(\widetilde{v}^0)^\top \Sigma^{-1}\widetilde{v}^0$$

exist and are finite. There exists a uniformly pseudo-Lipschitz function $f_0 \colon \mathbb{R}^d \to \mathbb{R}^d$ of order 1 such that

$$\lim_{d \to \infty} \frac{1}{d}\mathbb{E}[\langle f_0(V^*), f_0(V^*)\rangle] \leq \operatorname*{p\text{-}lim}_{d \to \infty} \frac{1}{d}\big\|\widetilde{v}^0\big\|_2^2$$

and for every uniformly pseudo-Lipschitz function $\phi \colon \mathbb{R}^d \to \mathbb{R}^d$ of finite order, the following two limits exist, are finite and equal:

$$\operatorname*{p\text{-}lim}_{d \to \infty} \frac{1}{d}\big\langle \widetilde{v}^0, \phi(V^*)\big\rangle = \lim_{d \to \infty} \frac{1}{d}\mathbb{E}[\langle f_0(V^*), \phi(V^*)\rangle].$$

Let $\widetilde{\nu} \in \mathbb{R}, \widetilde{\tau} \in \mathbb{R}_{\geq 0}$. For any $s \geq 1$,

$$\lim_{d \to \infty} \frac{1}{d}\mathbb{E}\Big[f_0(V^*)^\top \Sigma^{-1} f_s(\widetilde{\nu}\Sigma^{-1}V^* + \widetilde{\tau}\Sigma^{-1/2}W_V)\Big]$$

exists and is finite, where $W_V \sim \mathcal{N}(0_d, I_d)$ is independent of $V^*$.

(A2) Let $\widetilde{\nu} \in \mathbb{R}$, and $T \in \mathbb{R}^{2 \times 2}$ be positive definite. For any $r, s \geq 1$,

$$\lim_{n \to \infty} \frac{\lambda}{n}\mathbb{E}\Big[\big\langle \Sigma^{-1}V^*, f_r(\widetilde{\nu}\Sigma^{-1}V^* + \Sigma^{-1/2}N)\big\rangle\Big],$$

$$\lim_{d \to \infty} \frac{1}{d}\mathbb{E}\Big[f_r(\widetilde{\nu}\Sigma^{-1}V^* + \Sigma^{-1/2}N)^\top \Sigma^{-1} f_s(\widetilde{\nu}\Sigma^{-1}V^* + \Sigma^{-1/2}N')\Big]$$

exist and are finite, where $(N, N') \sim \mathcal{N}(0_{2d}, T \otimes I_d)$ is independent of $V^*$.
Let $\widetilde{\mu} \in \mathbb{R}$, and $S \in \mathbb{R}^{2 \times 2}$ be positive definite. For any $r, s \geq 0$,

$$\lim_{n \to \infty} \frac{\lambda}{n}\mathbb{E}\Big[\big\langle \Xi^{-1}U^*, g_r(\widetilde{\mu}\Xi^{-1}U^* + \Xi^{-1/2}M)\big\rangle\Big],$$

$$\lim_{d \to \infty} \frac{1}{d}\mathbb{E}\Big[g_r(\widetilde{\mu}\Xi^{-1}U^* + \Xi^{-1/2}M)^\top \Xi^{-1} g_s(\widetilde{\mu}\Xi^{-1}U^* + \Xi^{-1/2}M')\Big]$$

exist and are finite, where $(M, M') \sim \mathcal{N}(0_{2n}, S \otimes I_n)$ is independent of $U^*$.

We now give the state evolution result for the AMP in (E.1), which is proved in Appendix F.

**Proposition E.1** (State evolution for AMP in (E.1)). *For every $t \geq 0$, let $(g_t \colon \mathbb{R}^n \to \mathbb{R}^n)_{n \geq 1}$ and $(f_{t+1} \colon \mathbb{R}^d \to \mathbb{R}^d)_{d \geq 1}$ be uniformly pseudo-Lipschitz of finite order subject to Assumption (A2). Consider the AMP iteration in (E.1) defined by $(g_t, f_{t+1})_{t \geq 0}$ and initialized at $\widetilde{u}^{-1} = 0_n$ and some $\widetilde{v}^0 \in \mathbb{R}^d$ subject to Assumption (A1). For any fixed $t \geq 0$, let $(\phi \colon \mathbb{R}^{(t+2)n} \to \mathbb{R})_{n \geq 1}$ and $(\psi \colon \mathbb{R}^{(t+2)d} \to \mathbb{R})_{d \geq 1}$ be uniformly pseudo-Lipschitz of finite order. Then,*

$$\operatorname*{p\text{-}lim}_{n \to \infty} \phi(u^*, u^0, u^1, \cdots, u^t) - \mathbb{E}[\phi(U^*, U_0, U_1, \cdots, U_t)] = 0, \tag{E.8a}$$

$$\operatorname*{p\text{-}lim}_{d \to \infty} \psi(v^*, v^1, v^2, \cdots, v^{t+1}) - \mathbb{E}[\psi(V^*, V_1, V_2, \cdots, V_{t+1})] = 0, \tag{E.8b}$$

*where $(U_s, V_{s+1})_{0 \leq s \leq t}$ are defined in (E.6).*

---

[3]Practically one can think of the dependence of $\widetilde{v}^0$ on $v^*$ being given by some side information. However, here AMP is used solely as a proof technique, and we can consider initializers with impractical access to $v^*$.

Given $U_t, V_{t+1}$, the Bayes-optimal (in terms of mean square error) choice of $(g_t, f_{t+1})_{t \geq 0}$ is given by the conditional expectations. Specifically, for any $t \geq 0$ and $u \in \mathbb{R}^n, v \in \mathbb{R}^d$,

$$g_t^*(u) := \mathbb{E}[U^* \mid U_t = u], \qquad f_{t+1}^*(v) := \mathbb{E}[V^* \mid V_{t+1} = v]. \tag{E.9}$$

We call (E.1) with $g_t = g_t^*, f_{t+1} = f_{t+1}^*$ the Bayes-AMP.

If $P = Q = \mathcal{N}(0,1)$, by (E.2) and (E.6), $(U^*, U_t)$ and $(V^*, V_{t+1})$ are jointly Gaussian with mean zero and covariance:

$$\begin{bmatrix} I_n & \mu_t \Xi^{-1} \\ \mu_t \Xi^{-1} & \mu_t^2 \Xi^{-2} + \sigma_t^2 \Xi^{-1} \end{bmatrix} \in \mathbb{R}^{2n \times 2n}, \qquad \begin{bmatrix} I_d & \nu_{t+1} \Sigma^{-1} \\ \nu_{t+1} \Sigma^{-1} & \nu_{t+1}^2 \Sigma^{-2} + \tau_{t+1}^2 \Sigma^{-1} \end{bmatrix} \in \mathbb{R}^{2d \times 2d},$$

respectively. Therefore using Proposition G.5, $g_t^*, f_{t+1}^*$ admit the following explicit formulas:

$$\begin{aligned} g_t^*(u) &= \mu_t \Xi^{-1} (\mu_t^2 \Xi^{-2} + \sigma_t^2 \Xi^{-1})^{-1} u = \mu_t (\mu_t^2 \Xi^{-1} + \sigma_t^2 I_n)^{-1} u, \\ f_{t+1}^*(v) &= \tau_{t+1} \Sigma^{-1} (\tau_{t+1}^2 \Sigma^{-2} + \tau_{t+1}^2 \Sigma^{-1})^{-1} v = \nu_{t+1} (\nu_{t+1}^2 \Sigma^{-1} + \tau_{t+1}^2 I_d)^{-1} v. \end{aligned} \tag{E.10}$$

Under the above choice, the state evolution recursion for $\mu_t, \sigma_t, \nu_{t+1}, \tau_{t+1}$ in (E.4) and (E.5) becomes: for all $t \geq 1$,

$$\mu_t = \lim_{n \to \infty} \frac{\lambda}{n} \mathbb{E} \left[ \langle \Sigma^{-1} V^*, \nu_t (\nu_t^2 \Sigma^{-1} + \tau_t^2 I_d)^{-1} V_t \rangle \right] = \frac{\lambda}{\delta} \mathbb{E} \left[ \frac{\nu_t^2 \overline{\Sigma}^{-2}}{\nu_t^2 \overline{\Sigma}^{-1} + \tau_t^2} \right], \tag{E.11a}$$

$$\nu_{t+1} = \lim_{n \to \infty} \frac{\lambda}{n} \mathbb{E} \left[ \langle \Xi^{-1} U^*, \mu_t (\mu_t^2 \Xi^{-1} + \sigma_t^2 I_n)^{-1} U_t \rangle \right] = \lambda \mathbb{E} \left[ \frac{\mu_t^2 \overline{\Xi}^{-2}}{\mu_t^2 \overline{\Xi}^{-1} + \sigma_t^2} \right], \tag{E.11b}$$

$$\begin{aligned} \sigma_t^2 &= \lim_{n \to \infty} \frac{1}{n} \mathbb{E} \left[ V_t^\top (\nu_t^2 \Sigma^{-1} + \tau_t^2 I_d)^{-1} \nu_t \Sigma^{-1} \nu_t (\nu_t^2 \Sigma^{-1} + \tau_t^2 I_d)^{-1} V_t \right] \\ &= \frac{1}{\delta} \mathbb{E} \left[ \frac{\nu_t^4 \overline{\Sigma}^{-3}}{(\nu_t^2 \overline{\Sigma}^{-1} + \tau_t^2)^2} \right] + \frac{1}{\delta} \mathbb{E} \left[ \frac{\nu_t^2 \tau_t^2 \overline{\Sigma}^{-2}}{(\nu_t^2 \overline{\Sigma}^{-1} + \tau_t^2)^2} \right] \\ &= \frac{1}{\delta} \mathbb{E} \left[ \frac{\nu_t^2 \overline{\Sigma}^{-2}}{\nu_t^2 \overline{\Sigma}^{-1} + \tau_t^2} \right], \end{aligned} \tag{E.11c}$$

$$\begin{aligned} \tau_{t+1}^2 &= \lim_{n \to \infty} \frac{1}{n} \mathbb{E} \left[ U_t^\top (\mu_t^2 \Xi^{-1} + \sigma_t^2 I_n)^{-1} \mu_t \Xi^{-1} \mu_t (\mu_t^2 \Xi^{-1} + \sigma_t^2 I_n)^{-1} U_t \right] \\ &= \mathbb{E} \left[ \frac{\mu_t^4 \overline{\Xi}^{-3}}{(\mu_t^2 \overline{\Xi}^{-1} + \sigma_t^2)^2} \right] + \mathbb{E} \left[ \frac{\mu_t^2 \sigma_t^2 \overline{\Xi}^{-2}}{(\mu_t^2 \overline{\Xi}^{-1} + \sigma_t^2)^2} \right] \\ &= \mathbb{E} \left[ \frac{\mu_t^2 \overline{\Xi}^{-2}}{\mu_t^2 \overline{\Xi}^{-1} + \sigma_t^2} \right], \end{aligned} \tag{E.11d}$$

where we have used the definitions (E.6) of $U_t, V_{t+1}$, the joint distribution (E.2) of $(U^*, W_{U,t}), (V^*, V_{t+1})$, the convergence of the empirical spectral distributions of $\Sigma, \Xi$, and Proposition G.2.

Inspecting the expressions, we realize that

$$\mu_t = \lambda \sigma_t^2, \qquad \nu_{t+1} = \lambda \tau_t^2. \tag{E.12}$$

This allows us to only track the recursion of $\mu_t, \nu_{t+1}$:

$$\mu_t = \frac{\lambda}{\delta} \mathbb{E} \left[ \frac{\lambda \nu_t \overline{\Sigma}^{-2}}{\lambda \nu_t \overline{\Sigma}^{-1} + 1} \right], \qquad \nu_{t+1} = \lambda \mathbb{E} \left[ \frac{\lambda \mu_t \overline{\Xi}^{-2}}{\lambda \mu_t \overline{\Xi}^{-1} + 1} \right].$$

Thus, the fixed point $(\mu^*, \nu^*)$ of the above recursion must satisfy:

$$\mu^* = \frac{\lambda}{\delta} \mathbb{E} \left[ \frac{\lambda \nu^* \overline{\Sigma}^{-2}}{\lambda \nu^* \overline{\Sigma}^{-1} + 1} \right], \qquad \nu^* = \lambda \mathbb{E} \left[ \frac{\lambda \mu^* \overline{\Xi}^{-2}}{\lambda \mu^* \overline{\Xi}^{-1} + 1} \right]. \tag{E.13}$$

Note that upon a change of variable

$$q_u := \frac{\nu^*}{\lambda}, \qquad q_v := \frac{\delta \mu^*}{\lambda}, \tag{E.14}$$

the fixed point equation (E.13) coincides with that in the characterization of the free energy; see (4.5).

## E.2 Spectral estimator from Bayes-AMP

Under Gaussian priors, the Bayes-AMP algorithm specified by (E.1) and (E.10) naturally suggests a spectral estimator with respect to a matrix which is a non-trivial transformation of $A$. In what follows, we provide a heuristic derivation of this spectral estimator. Its performance guarantee (Theorem 5.1) will be proved in Appendix E.4.

Suppose, informally, that $\mu_t, \sigma_t, \nu_{t+1}, \tau_{t+1}, u^t, v^{t+1}, c_t, b_{t+1}$ converge (under the sequential limits $n \to \infty, t \to \infty$) to $\mu^*, \sigma^*, \nu^*, \tau^*, u, v, c^*, b^*$, respectively, in the sense that, e.g.,

$$\lim_{t \to \infty} \operatorname{p-lim}_{n \to \infty} \frac{1}{\sqrt{n}} \left\| u^t - u \right\|_2 = 0.$$

Recall that $(\mu^*, \nu^*)$ solves the fixed point equation (E.13), and from (E.12), the following relation holds:

$$\mu^* = \lambda \sigma^{*2}, \qquad \nu^* = \lambda \tau^{*2}. \tag{E.15}$$

So denoting

$$G := \lambda(\lambda \mu^* \Xi^{-1} + I_n)^{-1} \in \mathbb{R}^{n \times n}, \qquad F := \lambda(\lambda \nu^* \Sigma^{-1} + I_d)^{-1} \in \mathbb{R}^{d \times d}, \tag{E.16}$$

by the design of $g_t^*, f_{t+1}^*$ in (E.10), we have that $\widetilde{u}^t, \widetilde{v}^{t+1}$ converge to

$$\widetilde{u} = \mu^* \left( \mu^{*2} \Xi^{-1} + \sigma^{*2} I_n \right)^{-1} u = Gu, \qquad \widetilde{v} = \nu^* \left( \nu^{*2} \Sigma^{-1} + \tau^{*2} I_d \right)^{-1} v = Fv,$$

respectively, and the limiting Onsager coefficients $b^*, c^*$ are given by:

$$b^* = \lim_{n \to \infty} \frac{1}{n} \operatorname{Tr}(F \Sigma^{-1}) = \frac{1}{\delta} \mathbb{E} \left[ \frac{\lambda}{\lambda \nu^* + \overline{\Sigma}} \right], \quad c^* = \lim_{n \to \infty} \frac{1}{n} \operatorname{Tr}(G \Xi^{-1}) = \mathbb{E} \left[ \frac{\lambda}{\lambda \mu^* + \overline{\overline{\Xi}}} \right]. \tag{E.17}$$

At the fixed point of (E.1), we have

$$u = \Xi^{-1} A \Sigma^{-1} \widetilde{v} - b^* \Xi^{-1} \widetilde{u} = \Xi^{-1} A \Sigma^{-1} F v - b^* \Xi^{-1} G u,$$
$$v = \Sigma^{-1} A^\top \Xi^{-1} \widetilde{u} - c^* \Sigma^{-1} \widetilde{v} = \Sigma^{-1} A^\top \Xi^{-1} G u - c^* \Sigma^{-1} F v. \tag{E.18}$$

Upon rearrangement, (E.18) is equivalent to

$$\widehat{G} u = \Xi^{-1} A \Sigma^{-1} F v, \qquad \widehat{F} v = \Sigma^{-1} A^\top \Xi^{-1} G u, \tag{E.19}$$

where

$$\widehat{G} := I_n + b^* \Xi^{-1} G \in \mathbb{R}^{n \times n}, \qquad \widehat{F} := I_d + c^* \Sigma^{-1} F \in \mathbb{R}^{d \times d}. \tag{E.20}$$

We further introduce the notation:

$$\widetilde{G} := \widehat{G} G^{-1} \in \mathbb{R}^{n \times n}, \qquad \widetilde{F} := \widehat{F} F^{-1} \in \mathbb{R}^{d \times d}, \tag{E.21}$$

so that (E.19) can be rewritten as

$$\widetilde{G} G u = \Xi^{-1} A \Sigma^{-1} F v, \qquad \widetilde{F} F v = \Sigma^{-1} A^\top \Xi^{-1} G u,$$

or

$$\widetilde{G}^{1/2} G u = \widetilde{G}^{-1/2} \Xi^{-1} A \Sigma^{-1} \widetilde{F}^{-1/2} \cdot \widetilde{F}^{1/2} F v, \qquad \widetilde{F}^{1/2} F v = \widetilde{F}^{-1/2} \Sigma^{-1} A^\top \Xi^{-1} \widetilde{G}^{-1/2} \cdot \widetilde{G}^{1/2} G u.$$

The key observation is that this is a pair of singular vector equations for the matrix

$$A^* := \widetilde{G}^{-1/2} \Xi^{-1} A \Sigma^{-1} \widetilde{F}^{-1/2} \in \mathbb{R}^{n \times d} \tag{E.22}$$

with respect to left/right singular vectors (up to rescaling)

$$\widetilde{G}^{1/2} G u \in \mathbb{R}^n, \qquad \widetilde{F}^{1/2} F v \in \mathbb{R}^d$$

and singular value 1. Using the definitions (E.16), (E.20) and (E.21), we verify that the two expressions of $A^*$ in (5.3) and (E.22) are equal.

By the state evolution result (Proposition E.1), $u, v$ behave (in the sense of (E.8)) as

$$\mu^* \Xi^{-1} u^* + \sigma^* \Xi^{-1/2} W_U, \qquad \nu^* \Sigma^{-1} v^* + \tau^* \Sigma^{-1/2} W_V,$$

for $W_U \sim \mathcal{N}(0_n, I_d), W_V \sim \mathcal{N}(0_d, I_d)$ independent of each other and of $u^*, v^*$. This suggests that

$$\Xi(\widetilde{G}^{1/2} G)^{-1} u_1(A^*), \qquad \Sigma(\widetilde{F}^{1/2} F)^{-1} v_1(A^*) \tag{E.23}$$

are effective estimates of $u^*, v^*$. Simple algebra reveals that the above vectors, upon suitable rescaling, are precisely $\widehat{u}, \widehat{v}$ in (5.4).

## E.3 Right edge of the bulk

Before proceeding with the proof of Theorem 5.1, we provide a characterization of $\sigma_2(A^*)$, i.e., the right edge of the bulk of the spectrum of $A^*$. This bulk is related to the spectrum of the non-spiked random matrix

$$W^* := \lambda(\lambda(\mu^* + b^*)I_n + \Xi)^{-1/2}\widetilde{W}(\lambda(\nu^* + c^*)I_d + \Sigma)^{-1/2}. \tag{E.24}$$

We first present a characterization of $\sigma_1(W^*)$ and then relate it to $\sigma_2(A^*)$. Define random variables:

$$\overline{\Xi}^* := \frac{\lambda}{\lambda(\mu^* + b^*) + \overline{\overline{\Xi}}}, \qquad \overline{\Sigma}^* := \frac{\lambda}{\lambda(\nu^* + c^*) + \overline{\overline{\Sigma}}}. \tag{E.25}$$

Define functions $c, s \colon (\sup \operatorname{supp}(\overline{\Xi}^*), \infty) \to (0, \infty)$ as

$$c(\alpha) = \mathbb{E}\left[\frac{\overline{\Xi}^*}{\alpha - \overline{\Xi}^*}\right], \quad s(\alpha) = \sup \operatorname{supp}(\overline{\Sigma}^*)c(\alpha).$$

Define the implicit function $\beta \colon (\sup \operatorname{supp}(\overline{\Xi}^*), \infty) \to (0, \infty)$ as, for any $\alpha \in (\sup \operatorname{supp}(\overline{\Xi}^*), \infty)$, the unique solution in $(s(\alpha), \infty)$ to

$$1 = \frac{1}{\delta}\mathbb{E}\left[\frac{\overline{\Sigma}^*}{\beta - c(\alpha)\overline{\Sigma}^*}\right].$$

(The existence and uniqueness of the solution is easy to see.) Next, define $\psi \colon (\sup \operatorname{supp}(\overline{\Xi}^*), \infty) \to (0, \infty)$ as $\psi(\alpha) = \alpha\beta(\alpha)$. It is known that $\psi$ is differentiable and the set of its critical points is a nonempty finite set [79]. Let $\alpha^\circ \in (\sup \operatorname{supp}(\overline{\Xi}^*), \infty)$ be the largest critical point, i.e.,

$$1 = \frac{1}{\delta}\mathbb{E}\left[\frac{\overline{\Sigma}^{*2}}{(\beta(\alpha) - c(\alpha)\overline{\Sigma}^*)^2}\right]\mathbb{E}\left[\frac{\overline{\Xi}^{*2}}{(\alpha - \overline{\Xi}^*)^2}\right]$$

Finally, denote

$$\sigma_2^* := \sqrt{\psi(\alpha^\circ)}. \tag{E.26}$$

The characterization of $\sigma_1(W^*)$ requires an extra technical assumption on the random variables $\overline{\overline{\Xi}}, \overline{\overline{\Sigma}}$, which is the same as in [79].

(A3)  For any $c > 0$,

$$\lim_{\beta \downarrow s}\mathbb{E}\left[\frac{\overline{\Sigma}^*}{\beta - c\overline{\Sigma}^*}\right] = \lim_{\beta \downarrow s}\mathbb{E}\left[\left(\frac{\overline{\Sigma}^*}{\beta - c\overline{\Sigma}^*}\right)^2\right] = \infty.$$

where $s := c \cdot \sup \operatorname{supp}(\overline{\Sigma}^*)$. Furthermore,

$$\lim_{\alpha \downarrow \sup \operatorname{supp}(\overline{\Xi}^*)}\mathbb{E}\left[\frac{\overline{\Xi}^*}{\alpha - \overline{\Xi}^*}\right] = \infty.$$

**Lemma E.2.** *Let Assumption* (A3) *hold. Consider* $W^*$ *defined in* (E.24). *Then, we have*

$$\operatorname*{p-lim}_{n \to \infty} \sigma_1(W^*) = \sigma_2^*.$$

*Proof.* Note that $W^*W^{*\top}$ is a separable covariance matrix. Its largest eigenvalue is characterized in [25]. The explicit formulas we need are due to [79]. To apply their results, one simply observes that the covariances (as in the context of separable covariance matrices) of $W^*$ are

$$\Xi^* := \sqrt{\lambda}(\lambda(\mu^* + b^*)I_n + \Xi)^{-1/2}, \qquad \Sigma^* := \sqrt{\lambda}(\lambda(\nu^* + c^*)I_d + \Sigma)^{-1/2},$$

whose limiting spectral distributions are given by the distributions of $\overline{\Xi}^*, \overline{\Sigma}^*$ in (E.25).  $\square$

**Lemma E.3.** *Consider* $A^*$ *defined in* (5.3). *Then*

$$\operatorname*{p-lim}_{n \to \infty} \sigma_2(A^*) = \sigma_2^*.$$

*Proof.* By Weyl's inequality, $\sigma_3(W^*) \leq \sigma_1(A^*) \leq \sigma_1(W^*)$. We have already shown in Lemma E.2 that $\sigma_1(W^*)$ converges to $\sigma_2^*$. The almost sure weak convergence of the empirical spectral distribution of $W^*$ [78, Theorem 1.2.1] implies that $\sigma_3(W^*)$ (and indeed $\sigma_k(W^*)$ for any constant $k$ relative to $n, d$) must also converge to the same limit $\sigma_2^*$.  $\square$

### E.4 Proof of Theorem 5.1

We suppose throughout the proof that the condition (5.1) holds. Then, by Proposition 4.1 and the change of variable (E.14), the fixed point equation (E.13) admits a unique non-trivial solution $(\mu^*, \nu^*) \in \mathbb{R}_{>0}^2$. Construct matrices $F, G$ as in (E.16) using such $\mu^*, \nu^*$. Define also the random variables

$$\overline{G} := \frac{\lambda}{\lambda \mu^* \overline{\Xi}^{-1} + 1}, \quad \overline{F} := \frac{\lambda}{\lambda \nu^* \overline{\Sigma}^{-1} + 1}$$

whose distributions are the limiting spectral distributions of $G, F$, respectively.

Now consider the denoising functions:

$$f_{t+1}(v^{t+1}) = F v^{t+1}, \quad g_t(u^t) = G u^t.$$

With this choice, the AMP iteration (E.1) becomes

$$u^t = \Xi^{-1} A \Sigma^{-1} F v^t - b \Xi^{-1} G u^{t-1}, \quad v^{t+1} = \Sigma^{-1} A^\top \Xi^{-1} G u^t - c \Sigma^{-1} F v^t, \quad \text{(E.27)}$$

where

$$b = \frac{1}{n} \operatorname{Tr}(F \Sigma^{-1}), \quad c = \frac{1}{n} \operatorname{Tr}(G \Xi^{-1}).$$

Note that as $n \to \infty$, $b, c$ converge to $b^*, c^*$ in (E.17).

Recall from (E.15) the definition of $\sigma^*, \tau^*$. We initialize (E.27) with

$$\widetilde{u}^{-1} = 0_n, \quad \widetilde{v}^0 = F(\nu^* \Sigma^{-1} v^* + \tau^* \Sigma^{-1/2} w),$$

where $w \sim \mathcal{N}(0_d, I_d)$ is independent of everything else. Accordingly, one can take $f_0$ in Assumption (A1) to be $f_0(v) = \nu^* F \Sigma^{-1} v$. Under the above AMP initializer, the state evolution initializers in (E.3) and (E.5) specialize to

$$\mu_0 = \lim_{n \to \infty} \frac{\lambda}{n} \mathbb{E}\Big[ V^{*\top} \Sigma^{-1} F \Sigma^{-1} V^* \Big] \nu^* = \frac{\lambda}{\delta} \mathbb{E}\Big[ \overline{F \Sigma}^{-2} \Big] \nu^* = \mu^*,$$

$$\sigma_0^2 = \operatorname*{p-lim}_{n \to \infty} \frac{\nu^{*2}}{n} v^{*\top} \Sigma^{-1} F \Sigma^{-1} F \Sigma^{-1} v^* + \frac{\tau^{*2}}{n} w^\top \Sigma^{-1/2} F \Sigma^{-1} F \Sigma^{-1/2} w$$

$$= \frac{1}{\delta} \mathbb{E}\Big[ \overline{F}^2 \overline{\Sigma}^{-3} \Big] \nu^{*2} + \frac{1}{\delta} \mathbb{E}\Big[ \overline{F}^2 \overline{\Sigma}^{-2} \Big] \tau^{*2} = \sigma^{*2},$$

where the last equalities for both chains of computation are by (E.11). Since the parameters $\mu_t, \sigma_t$ are initialized at the non-trivial fixed point $(\mu_0, \sigma_0) = (\mu^*, \sigma^*)$, the state evolution recursion (E.11) will stay at the fixed point $(\mu_t, \sigma_t, \nu_{t+1}, \tau_{t+1}) = (\mu^*, \sigma^*, \nu^*, \tau^*)$ across all $t \geq 0$.

**Lemma E.4.** *Let*

$$\widehat{u}^t := \widetilde{G}^{1/2} G u^t, \quad \widehat{v}^{t+1} := \widetilde{F}^{1/2} F v^{t+1}, \quad \text{(E.28)}$$

*where $\widetilde{F}, \widetilde{G}$ are defined in (E.21). Suppose the condition (5.1) holds. Then*

$$\lim_{t \to \infty} \operatorname*{p-lim}_{n \to \infty} \frac{\big|\langle \widehat{u}^t, u_1(A^*) \rangle\big|}{\|\widehat{u}^t\|_2} = \lim_{t \to \infty} \operatorname*{p-lim}_{n \to \infty} \frac{\big|\langle \widehat{v}^{t+1}, v_1(A^*) \rangle\big|}{\|\widehat{v}^{t+1}\|_2} = 1 \quad \text{(E.29)}$$

*and*

$$\operatorname*{p-lim}_{n \to \infty} \sigma_1(A^*) = 1, \quad \text{(E.30)}$$

*where $A^*$ is defined in (E.22).*

*Proof.* Denoting

$$e_1^t := u^t - u^{t-1}, \quad e_2^{t+1} := v^{t+1} - v^t, \quad \text{(E.31)}$$

for any $t \geq 1$ and using the notation $\widehat{F}, \widehat{G}$ in (E.20), we have from (E.27) that

$$\widehat{G} u^t = \Xi^{-1} A \Sigma^{-1} F v^t + b^* \Xi^{-1} G e_1^t + (b^* - b) \Xi^{-1} G u^{t-1},$$

$$\widehat{F}v^{t+1} = \Sigma^{-1}A^\top\Xi^{-1}Gu^t + c^*\Sigma^{-1}Fe_2^{t+1} + (c^* - c)\Sigma^{-1}Fv^t.$$

Recalling the notation $\widetilde{F}, \widetilde{G}$ from (E.21) and multiplying $\widetilde{G}^{-1/2}$ (resp. $\widetilde{F}^{-1/2}$) on both sides of the first (resp. second) equation above, we arrive at

$$\widetilde{G}^{1/2}Gu^t = A^* \cdot \widetilde{F}^{1/2}Fv^t + b^*\widetilde{G}^{-1/2}\Xi^{-1}Ge_1^t + (b^* - b)\widetilde{G}^{-1/2}\Xi^{-1}Gu^{t-1},$$

$$\widetilde{F}^{1/2}Fv^{t+1} = A^{*\top} \cdot \widetilde{G}^{1/2}Gu^t + c^*\widetilde{F}^{-1/2}\Sigma^{-1}Fe_2^{t+1} + (c^* - c)\widetilde{F}^{-1/2}\Sigma^{-1}Fv^t.$$

Using the definition of $\widehat{u}^t, \widehat{v}^{t+1}$ in (E.28), we rewrite the above as

$$\widehat{u}^t = A^*\widehat{v}^t + e_u^t, \quad \widehat{v}^{t+1} = A^{*\top}\widehat{u}^t + e_v^{t+1}, \tag{E.32}$$

where

$$\begin{aligned}
e_u^t &:= b\widetilde{G}^{-1/2}\Xi^{-1}Ge_1^t + (b^* - b)\widetilde{G}^{-1/2}\Xi^{-1}Gu^{t-1}, \\
e_v^{t+1} &:= c\widetilde{F}^{-1/2}\Sigma^{-1}Fe_2^{t+1} + (c^* - c)\widetilde{F}^{-1/2}\Sigma^{-1}Fv^t.
\end{aligned} \tag{E.33}$$

Let us focus on $\widehat{u}^t$ and only prove the first equality in (E.29). The proof of the second one is similar and will be omitted. Eliminating $\widehat{v}^t$ from (E.32) gives

$$\widehat{u}^t = A^*A^{*\top}\widehat{u}^{t-1} + A^*e_v^t + e_u^t.$$

Unrolling this recursion for $s$ steps, we get:

$$\widehat{u}^{t+s} = \left(A^*A^{*\top}\right)^s\widehat{u}^t + \widehat{e}_u^{t,s}, \tag{E.34}$$

where

$$\widehat{e}_u^{t,s} := \sum_{r=1}^{s}\left(A^*A^{*\top}\right)^{s-r}(A^*e_v^{t+r} + e_u^{t+r}). \tag{E.35}$$

Taking $\frac{1}{n}\|\cdot\|_2^2$ on both sides of (E.34) and take the sequential limits of $n \to \infty$, $t \to \infty$, $s \to \infty$, we have the left hand side:

$$\begin{aligned}
\lim_{s\to\infty}\lim_{t\to\infty}\operatorname*{p-lim}_{n\to\infty}\frac{1}{n}\left\|\widehat{u}^{t+s}\right\|_2^2 &= \lim_{t\to\infty}\operatorname*{p-lim}_{n\to\infty}\frac{1}{n}\left\|\widehat{u}^t\right\|_2^2 \\
&= \lim_{t\to\infty}\operatorname*{p-lim}_{n\to\infty}\frac{1}{n}\left\|\widetilde{G}^{1/2}Gu^t\right\|_2^2 \\
&= \lim_{t\to\infty}\operatorname*{p-lim}_{n\to\infty}\frac{1}{n}\left\|\widehat{G}^{1/2}G^{1/2}u^t\right\|_2^2 \\
&= \lim_{t\to\infty}\mu_t^2\mathbb{E}\left[\overline{\Xi}^{-2}(1 + b^*\overline{\Xi}^{-1}\overline{G})\overline{G}\right] + \sigma_t^2\mathbb{E}\left[\overline{\Xi}^{-1}(1 + b^*\overline{\Xi}^{-1}\overline{G})\overline{G}\right] \\
&= \mu^{*2}\mathbb{E}\left[\overline{\Xi}^{-2}(1 + b^*\overline{\Xi}^{-1}\overline{G})\overline{G}\right] + \frac{\mu^*}{\lambda}\mathbb{E}\left[\overline{\Xi}^{-1}(1 + b^*\overline{\Xi}^{-1}\overline{G})\overline{G}\right] \\
&= \lambda\mu^{*2}\mathbb{E}\left[\frac{\lambda(\mu^* + b^*) + \overline{\Xi}}{\overline{\Xi}\left(\lambda\mu^* + \overline{\Xi}\right)^2}\right] + \mu^*\mathbb{E}\left[\frac{\lambda(\mu^* + b^*) + \overline{\Xi}}{\left(\lambda\mu^* + \overline{\Xi}\right)^2}\right] \in (0, \infty).
\end{aligned} \tag{E.36}$$

Next, we have that

$$\lim_{t\to\infty}\operatorname*{p-lim}_{n\to\infty}\frac{1}{n}\left\|e_1^t\right\|_2^2 = \lim_{t\to\infty}\operatorname*{p-lim}_{d\to\infty}\frac{1}{d}\left\|e_2^{t+1}\right\|_2^2 = 0. \tag{E.37}$$

To prove the first statement on $e_1^t$, the strategy is to write the LHS of (E.37) in terms of the state evolution parameters and prove that the latter quantities converge. We start with

$$\begin{aligned}
\operatorname*{p-lim}_{n\to\infty}\frac{1}{n}\left\|u^t - u^{t-1}\right\|_2^2 &= \left(\mu_t^2\mathbb{E}\left[\overline{\Sigma}^{-2}\right] + \sigma_t^2\mathbb{E}\left[\overline{\Sigma}^{-1}\right]\right) + \left(\mu_{t-1}^2\mathbb{E}\left[\overline{\Sigma}^{-2}\right] + \sigma_{t-1}^2\mathbb{E}\left[\overline{\Sigma}^{-1}\right]\right) \\
&\quad - 2\left(\mu_t\mu_{t-1}\mathbb{E}\left[\overline{\Sigma}^{-2}\right] + \Phi_{t,t-1}\mathbb{E}\left[\overline{\Sigma}^{-1}\right]\right)
\end{aligned}$$

$$= 2\sigma^{*2}\mathbb{E}\!\left[\overline{\Sigma}^{-1}\right] - 2\Phi_{t,t-1}\mathbb{E}\!\left[\overline{\Sigma}^{-1}\right],$$

where the first equality is by Proposition G.3 and the joint distribution of $W_{U,t}, W_{U,t-1}$ in (E.2), and the second one holds since the state evolution parameters stay at the fixed point upon initialization. So it remains to verify that $\Phi_{t,t-1} \to \sigma^{*2}$ as $t \to \infty$. To see this, note that according to the state evolution recursion (E.3),

$$\Phi_{t,t-1} = \lim_{n\to\infty} \frac{1}{n}\mathbb{E}\!\left[V_t^\top F^\top \Sigma^{-1} F V_{t-1}\right] = \frac{\nu^{*2}}{\delta}\mathbb{E}\!\left[\overline{F}^2\overline{\Sigma}^{-3}\right] + \frac{\Psi_{t,t-1}}{\delta}\mathbb{E}\!\left[\overline{F}^2\overline{\Sigma}^{-2}\right],$$

$$\Psi_{t+1,t} = \lim_{n\to\infty} \frac{1}{n}\mathbb{E}\!\left[U_t^\top G^\top \Xi^{-1} G U_{t-1}\right] = \mu^{*2}\mathbb{E}\!\left[\overline{G}^2\overline{\Xi}^{-3}\right] + \Phi_{t,t-1}\mathbb{E}\!\left[\overline{G}^2\overline{\Xi}^{-2}\right].$$

Eliminating $\Psi_{t,t-1}$ from the first equation, we arrive at

$$\Phi_{t,t-1} = \frac{\nu^{*2}}{\delta}\mathbb{E}\!\left[\overline{F}^2\overline{\Sigma}^{-3}\right] + \frac{1}{\delta}\mathbb{E}\!\left[\overline{F}^2\overline{\Sigma}^{-2}\right]\left(\mu^{*2}\mathbb{E}\!\left[\overline{G}^2\overline{\Xi}^{-3}\right] + \Phi_{t-1,t-2}\mathbb{E}\!\left[\overline{G}^2\overline{\Xi}^{-2}\right]\right)$$

$$= \frac{1}{\delta}\mathbb{E}\!\left[\frac{\lambda^2\overline{\Sigma}^{-3}}{(\lambda\nu^*\overline{\Sigma}^{-1}+1)^2}\right]\nu^{*2} + \frac{1}{\delta}\mathbb{E}\!\left[\frac{\lambda^2\overline{\Sigma}^{-2}}{(\lambda\nu^*\overline{\Sigma}^{-1}+1)^2}\right]$$

$$\times\left(\mathbb{E}\!\left[\frac{\lambda^2\overline{\Xi}^{-3}}{(\lambda\mu^*\overline{\Xi}^{-1}+1)^2}\right]\mu^{*2} + \mathbb{E}\!\left[\frac{\lambda^2\overline{\Xi}^{-2}}{(\lambda\mu^*\overline{\Xi}^{-1}+1)^2}\right]\Phi_{t-1,t-2}\right),$$

whose only fixed point is $\sigma^{*2}$ by the relations in (E.11). This concludes the proof of the first equality in (E.37). The proof of the second equality is analogous and, hence, omitted.

By using (E.37) and the fact that $b \to b^*, c \to c^*$ as $n \to \infty$ (see (E.17)), we obtain

$$\lim_{t\to\infty} \operatorname*{p-lim}_{n\to\infty} \frac{1}{n}\left\|e_u^t\right\|_2^2 = \lim_{t\to\infty} \operatorname*{p-lim}_{d\to\infty} \frac{1}{d}\left\|e_v^{t+1}\right\|_2^2 = 0. \tag{E.38}$$

Note that the operator norm of $A^*$ is almost surely bounded uniformly in $n$ by Weyl's inequality, sub-multiplicativity of matrix norms and the Bai–Yin law [3]. This together with the triangle inequality of the $\ell_2$-norm and (E.38) implies that

$$\lim_{s\to\infty} \lim_{t\to\infty} \operatorname*{p-lim}_{n\to\infty} \frac{1}{n}\left\|\widehat{e}_u^{t,s}\right\|_2^2 = 0, \tag{E.39}$$

From this, it follows that the right hand side of (E.34) (upon taken the rescaled squared norm and the sequential limits) equals

$$\lim_{s\to\infty} \lim_{t\to\infty} \operatorname*{p-lim}_{n\to\infty} \frac{1}{n}\left\|\left(A^* A^{*\top}\right)^s \widehat{u}^t\right\|_2^2.$$

We then compute the above term by taking the SVD of $A^*$. Define two spectral projectors that are orthogonal to each other:

$$\Pi := u_1(A^*)u_1(A^*)^\top, \quad \Pi^\perp := I_n - \Pi.$$

We have

$$\frac{1}{n}\left\|\left(A^* A^{*\top}\right)^s \widehat{u}^t\right\|_2^2 = \frac{1}{n}\left\|\left(A^* A^{*\top}\right)^s \Pi\widehat{u}^t\right\|_2^2 + \frac{1}{n}\left\|\left(A^* A^{*\top}\right)^s \Pi^\perp\widehat{u}^t\right\|_2^2. \tag{E.40}$$

Using the spectral decomposition

$$\left(A^* A^{*\top}\right)^s = \sum_{i=1}^n \sigma_i(A^*)^{2s} u_i(A^*)u_i(A^*)^\top,$$

we can write the first term in (E.40) as

$$\frac{1}{n}\left\|\left(A^* A^{*\top}\right)^s \Pi\widehat{u}^t\right\|_2^2 = \frac{1}{n}\left\|\sum_{i=1}^n \sigma_i(A^*)^{2s} u_i(A^*)u_i(A^*)^\top u_1(A^*)u_1(A^*)^\top \widehat{u}^t\right\|_2^2 = \sigma_1(A^*)^{4s}\frac{\langle u_1(A^*), \widehat{u}^t\rangle^2}{n}.$$
$$\tag{E.41}$$

For the second term in (E.40), we have

$$\frac{1}{n}\left\|\left(A^*A^{*\top}\right)^s\Pi^\perp\widehat{u}^t\right\|_2^2 = \frac{1}{n}\left\|\left(A^*A^{*\top}\Pi^\perp\right)^s\widehat{u}^t\right\|_2^2$$

$$\leq \frac{\|\widehat{u}^t\|_2^2}{n}\max_{u\in\mathbb{S}^{n-1}}\left\|\left(A^*A^{*\top}\Pi^\perp\right)^su\right\|_2^2$$

$$= \frac{\|\widehat{u}^t\|_2^2}{n}\sigma_1\left(\left(A^*A^{*\top}\Pi^\perp\right)^s\right)^2$$

$$= \frac{\|\widehat{u}^t\|_2^2}{n}\sigma_1\left(A^*A^{*\top}\Pi^\perp\right)^{2s}$$

$$= \frac{\|\widehat{u}^t\|_2^2}{n}\sigma_2\left(A^*A^{*\top}\right)^{2s}$$

$$= \frac{\|\widehat{u}^t\|_2^2}{n}\sigma_2(A^*)^{4s},$$

where penultimate line follows since

$$A^*A^{*\top}\Pi^\perp = \sum_{i=2}^n \sigma_i(A^*)^2 u_i(A^*)u_i(A^*)^\top.$$

From Lemma E.3 and the assumption that $\sigma_2^* < 1$, we know

$$\operatorname*{p-lim}_{n\to\infty}\sigma_2(A^*) = \sigma_2^* < 1.$$

This implies:

$$\lim_{s\to\infty}\lim_{t\to\infty}\operatorname*{p-limsup}_{n\to\infty}\frac{1}{n}\left\|\left(A^*A^{*\top}\right)^s\Pi^\perp\widehat{u}^t\right\|_2^2 \leq \lim_{s\to\infty}\lim_{t\to\infty}\operatorname*{p-limsup}_{n\to\infty}\frac{\|\widehat{u}^t\|_2^2}{n}\sigma_2(A^*)^{4s}$$

$$\leq \left(\lim_{t\to\infty}\operatorname*{p-limsup}_{n\to\infty}\frac{\|\widehat{u}^t\|_2^2}{n}\right)\left(\lim_{s\to\infty}\operatorname*{p-limsup}_{n\to\infty}\sigma_2(A^*)^{4s}\right) = 0, \quad \text{(E.42)}$$

where the last equality holds since the limit in the first parentheses is finite by (E.36).

Now (E.40) to (E.42) jointly show

$$\lim_{s\to\infty}\lim_{t\to\infty}\operatorname*{p-lim}_{n\to\infty}\frac{1}{n}\left\|\left(A^*A^{*\top}\right)^s\widehat{u}^t\right\|_2^2 = \lim_{s\to\infty}\lim_{t\to\infty}\operatorname*{p-lim}_{n\to\infty}\sigma_1(A^*)^{4s}\frac{\langle u_1(A^*),\widehat{u}^t\rangle^2}{n}$$

$$= \left(\lim_{s\to\infty}\operatorname*{p-lim}_{n\to\infty}\sigma_1(A^*)^{4s}\right)\left(\lim_{t\to\infty}\operatorname*{p-lim}_{n\to\infty}\frac{\langle u_1(A^*),\widehat{u}^t\rangle^2}{n}\right).$$

Combining this with (E.36) brings us to the following identity:

$$1 = \left(\lim_{s\to\infty}\operatorname*{p-lim}_{n\to\infty}\sigma_1(A^*)^{4s}\right)\left(\lim_{t\to\infty}\operatorname*{p-lim}_{n\to\infty}\frac{\langle u_1(A^*),\widehat{u}^t\rangle^2}{\|\widehat{u}^t\|_2^2}\right),$$

which necessarily implies

$$\operatorname*{p-lim}_{n\to\infty}\sigma_1(A^*) = 1, \qquad \lim_{t\to\infty}\operatorname*{p-lim}_{n\to\infty}\frac{\langle u_1(A^*),\widehat{u}^t\rangle^2}{\|\widehat{u}^t\|_2^2} = 1,$$

as desired. $\qquad\square$

With Lemma E.4, we can complete the proof of Theorem 5.1.

*Proof of Theorem 5.1.* The characterization (5.7) of the top two singular values have been obtained in Lemmas E.3 and E.4. It remains to compute the overlaps which can be done using Lemma E.4 and

the state evolution (Proposition E.1). Recall the estimators $\widehat{u}, \widehat{v}$ in (5.4) and their heuristic derivation in (E.23). Then

$$\operatorname*{p-lim}_{n\to\infty} \frac{\langle \widehat{u}, u^* \rangle^2}{\|\widehat{u}\|_2^2 \|u^*\|_2^2} = \operatorname*{p-lim}_{t\to\infty} \operatorname*{p-lim}_{n\to\infty} \frac{\langle \Xi u^t, u^* \rangle^2}{\|\Xi u^t\|_2^2 \|u^*\|_2^2} = \frac{\lambda\mu^*}{\lambda\mu^* + 1} = \eta_u^2,$$

establishing the first equality in (5.8). The second equality in (5.8) and other quantities in (5.9) and (5.10) can be similarly obtained. The proof is completed. □

# F  Proof of Proposition E.1

Recall $\widetilde{u}^*, \widetilde{v}^*$ from (C.7) and let

$$\widetilde{A} := \Xi^{-1/2} A \Sigma^{-1/2} = \frac{\lambda}{n} \widetilde{u}^* (\widetilde{v}^*)^\top + \widetilde{W}. \tag{F.1}$$

## F.1  Auxiliary AMP and its state evolution

For $(\breve{g}_t \colon \mathbb{R}^n \to \mathbb{R}^n, \breve{f}_{t+1} \colon \mathbb{R}^d \to \mathbb{R}^d)_{t\geq 0}$, the iterates of the auxiliary AMP, initialized at $\mathring{u}^{-1} = 0_d$ and some $\mathring{v}^0 \in \mathbb{R}^d$, are updated according to the following rules for every $t \geq 0$:

$$\breve{u}^t = \widetilde{A}\mathring{v}^t - \breve{b}_t \mathring{u}^{t-1}, \quad \mathring{u}^t = \breve{g}_t(\breve{u}^t), \quad \breve{c}_t = \frac{1}{n}\sum_{i=1}^n \partial_i \breve{g}_t(\breve{u}^t)_i,$$

$$\breve{v}^{t+1} = \widetilde{A}^\top \mathring{u}^t - \breve{c}_t \mathring{v}^t, \quad \mathring{v}^{t+1} = \breve{f}_{t+1}(\breve{v}^{t+1}), \quad \breve{b}_{t+1} = \frac{1}{n}\sum_{i=1}^d \partial_i \breve{f}_{t+1}(\breve{v}^{t+1})_i. \tag{F.2}$$

The state evolution result associated with the above auxiliary AMP iteration asserts that the distributions of $(\breve{u}^0, \breve{u}^1, \cdots, \breve{u}^t), (\breve{v}^1, \breve{v}^2, \cdots, \breve{v}^{t+1})$ converge (in the sense of (F.16)) respectively to the laws of the random vectors $(\breve{U}_0, \breve{U}_1, \cdots, \breve{U}_t), (\breve{V}_1, \breve{V}_2, \cdots \breve{V}_{t+1})$ defined below:

$$\breve{U}_t = \breve{\mu}_t \widetilde{U}^* + \breve{\sigma}_t \breve{W}_{U,t} \in \mathbb{R}^n, \quad \breve{V}_{t+1} = \breve{\nu}_{t+1} \widetilde{V}^* + \breve{\tau}_{t+1} \breve{W}_{V,t+1} \in \mathbb{R}^d, \tag{F.3}$$

where

$$\begin{bmatrix} U^* \\ \breve{\sigma}_0 \breve{W}_{U,0} \\ \breve{\sigma}_1 \breve{W}_{U,1} \\ \vdots \\ \breve{\sigma}_t \breve{W}_{U,t} \end{bmatrix} \sim P^{\otimes n} \otimes \mathcal{N}(0_{n(t+1)}, \breve{\Phi}_t \otimes I_n), \qquad \begin{bmatrix} V^* \\ \breve{\tau}_1 \breve{W}_{V,1} \\ \breve{\tau}_2 \breve{W}_{V,2} \\ \vdots \\ \breve{\tau}_{t+1} \breve{W}_{V,t+1} \end{bmatrix} \sim Q^{\otimes d} \otimes \mathcal{N}(0_{d(t+1)}, \breve{\Psi}_{t+1} \otimes I_d),$$

(F.4)

$$\widetilde{U}^* := \Xi^{-1/2} U^* \in \mathbb{R}^n, \qquad\qquad \widetilde{V}^* := \Sigma^{-1/2} V^* \in \mathbb{R}^d. \tag{F.5}$$

The parameters $\breve{\mu}_t, \breve{\nu}_{t+1} \in \mathbb{R}, \breve{\Phi}_t = (\breve{\Phi}_{r,s})_{0\leq r,s\leq t}, \breve{\Psi}_{t+1} = (\breve{\Psi}_{r+1,s+1})_{0\leq r,s\leq t} \in \mathbb{R}^{(t+1)\times(t+1)}$ are defined recursively through the following state evolution equations:

$$\breve{\mu}_0 = \lambda \lim_{d\to\infty} \frac{1}{n} \mathbb{E}\Big[\big\langle \widetilde{V}^*, \breve{f}_0(\widetilde{V}^*) \big\rangle\Big], \tag{F.6}$$

$$\breve{\mu}_t = \lambda \lim_{d\to\infty} \frac{1}{n} \mathbb{E}\Big[\big\langle \widetilde{V}^*, \breve{f}_t(\breve{V}_t) \big\rangle\Big], \quad t \geq 1, \tag{F.7}$$

$$\breve{\nu}_{t+1} = \lambda \lim_{n\to\infty} \frac{1}{n} \mathbb{E}\Big[\big\langle \widetilde{U}^*, \breve{g}_t(\breve{U}_t) \big\rangle\Big], \quad t \geq 0, \tag{F.8}$$

$$\breve{\Phi}_{0,0} = \operatorname*{p-lim}_{n\to\infty} \frac{1}{n} \big\|\mathring{v}^0\big\|_2^2, \tag{F.9}$$

$$\breve{\Phi}_{0,s} = \lim_{n\to\infty} \frac{1}{n} \mathbb{E}\Big[\big\langle \breve{f}_0(\widetilde{V}^*), \breve{f}_s(\breve{V}_s) \big\rangle\Big], \quad 1 \leq s \leq t, \tag{F.10}$$

$$\breve{\Phi}_{r,s} = \lim_{n\to\infty} \frac{1}{n} \mathbb{E}\Big[\big\langle \breve{f}_r(\breve{V}_r), \breve{f}_s(\breve{V}_s) \big\rangle\Big], \quad 1 \leq r,s \leq t, \tag{F.11}$$

$$\breve{\Psi}_{r+1,s+1} = \lim_{n\to\infty} \frac{1}{n} \mathbb{E}\Big[\Big\langle \breve{g}_r(\breve{U}_r), \breve{g}_s(\breve{U}_s)\Big\rangle\Big], \quad 0 \le r, s \le t, \tag{F.12}$$

where $\breve{f}_0$ is determined by $\mathring{v}^0$ through Assumption (A4) below. In particular,

$$\breve{\sigma}_0^2 = \operatorname*{p-lim}_{n\to\infty} \frac{1}{n} \big\| \mathring{v}^0 \big\|_2^2, \tag{F.13}$$

$$\breve{\sigma}_t^2 = \lim_{n\to\infty} \frac{1}{n} \mathbb{E}\Big[\Big\langle \breve{f}_t(\breve{V}_t), \breve{f}_t(\breve{V}_t)\Big\rangle\Big], \quad t \ge 1, \tag{F.14}$$

$$\breve{\tau}_{t+1}^2 = \lim_{n\to\infty} \frac{1}{n} \mathbb{E}\Big[\Big\langle \breve{g}_t(\breve{U}_t), \breve{g}_t(\breve{U}_t)\Big\rangle\Big], \quad t \ge 0. \tag{F.15}$$

We require the following assumptions to guarantee the existence and finiteness of the state evolution parameters defined above.

(A4)  $\mathring{v}^0$ is independent of $\widetilde{W}$ but may depend on $\widetilde{v}^*$. Assume that

$$\operatorname*{p-lim}_{d\to\infty} \frac{1}{d} \big\| \mathring{v}^0 \big\|_2^2$$

exists and is finite. There exists a uniformly pseudo-Lipschitz function $\breve{f}_0 \colon \mathbb{R}^d \to \mathbb{R}^d$ of order 1 such that

$$\lim_{d\to\infty} \frac{1}{d} \mathbb{E}\Big[\Big\langle \breve{f}_0(\widetilde{V}^*), \breve{f}_0(\widetilde{V}^*)\Big\rangle\Big] \le \operatorname*{p-lim}_{d\to\infty} \frac{1}{d} \big\| \mathring{v}^0 \big\|_2^2$$

and for every uniformly pseudo-Lipschitz function $\phi \colon \mathbb{R}^d \to \mathbb{R}^d$ of finite order, the following two limits exist, are finite and equal:

$$\operatorname*{p-lim}_{d\to\infty} \frac{1}{d} \big\langle \mathring{v}^0, \phi(\widetilde{v}^*) \big\rangle = \lim_{d\to\infty} \frac{1}{d} \mathbb{E}\Big[\Big\langle \breve{f}_0(\widetilde{V}^*), \phi(\widetilde{V}^*)\Big\rangle\Big].$$

Let $\widetilde{\nu} \in \mathbb{R}, \widetilde{\tau} \in \mathbb{R}_{\ge 0}$. For any $s \ge 1$,

$$\lim_{d\to\infty} \frac{1}{d} \mathbb{E}\Big[\breve{f}_0(\widetilde{V}^*)^\top \breve{f}_s(\widetilde{\nu}\widetilde{V}^* + \widetilde{\tau}W_V)\Big]$$

exists and is finite, where $W_V \sim \mathcal{N}(0_d, I_d)$ is independent of $\widetilde{V}^*$.

(A5)  Let $\widetilde{\nu} \in \mathbb{R}$, and $T \in \mathbb{R}^{2\times 2}$ be positive definite. For any $r, s \ge 1$,

$$\lim_{n\to\infty} \frac{\lambda}{n} \mathbb{E}\Big[\Big\langle \widetilde{V}^*, \breve{f}_r(\widetilde{\nu}\widetilde{V}^* + N)\Big\rangle\Big], \quad \lim_{d\to\infty} \frac{1}{d} \mathbb{E}\Big[\breve{f}_r(\widetilde{\nu}\widetilde{V}^* + N)^\top \breve{f}_s(\widetilde{\nu}\widetilde{V}^* + N')\Big]$$

exist and are finite, where $(N, N') \sim \mathcal{N}(0_{2d}, T \otimes I_d)$ is independent of $\widetilde{v}^*$.
Let $\widetilde{\mu} \in \mathbb{R}$, and $S \in \mathbb{R}^{2\times 2}$ be positive definite. For any $r, s \ge 0$,

$$\lim_{n\to\infty} \frac{\lambda}{n} \mathbb{E}\Big[\Big\langle \widetilde{U}^*, \breve{g}_r(\widetilde{\mu}\widetilde{U}^* + M)\Big\rangle\Big], \quad \lim_{d\to\infty} \frac{1}{d} \mathbb{E}\Big[\breve{g}_r(\widetilde{\mu}\widetilde{U}^* + M)^\top \breve{g}_s(\widetilde{\mu}\widetilde{U}^* + M')\Big]$$

exist and are finite, where $(M, M') \sim \mathcal{N}(0_{2n}, S \otimes I_n)$ is independent of $\widetilde{U}^*$.

**Proposition F.1** (State evolution for auxiliary AMP (F.2))**.**  *For every $t \ge 0$, let $(\breve{g}_t \colon \mathbb{R}^n \to \mathbb{R}^n)_{n\ge 1}$ and $(\breve{f}_{t+1} \colon \mathbb{R}^d \to \mathbb{R}^d)_{d\ge 1}$ be uniformly pseudo-Lipschitz of finite order subject to Assumption (A5). Consider the auxiliary AMP iteration in (F.2) defined by $(\breve{g}_t, \breve{f}_{t+1})_{t\ge 0}$ and initialized at $\mathring{u}^{-1} = 0_n$ and some $\mathring{v}^0 \in \mathbb{R}^d$ subject to Assumption (A4). For any fixed $t \ge 0$, let $(\phi \colon \mathbb{R}^{(t+2)n} \to \mathbb{R})_{n\ge 1}$ and $(\psi \colon \mathbb{R}^{(t+2)d} \to \mathbb{R})_{n\ge 1}$ be uniformly pseudo-Lipschitz functions of finite order. Then,*

$$\operatorname*{p-lim}_{n\to\infty} \phi\big(\widetilde{u}^*, \breve{u}^0, \breve{u}^1, \cdots, \breve{u}^t\big) - \mathbb{E}\Big[\phi\big(\widetilde{U}^*, \breve{U}_0, \breve{U}_1, \cdots, \breve{U}_t\big)\Big] = 0, \tag{F.16a}$$

$$\operatorname*{p-lim}_{d\to\infty} \psi\big(\widetilde{v}^*, \breve{v}^1, \breve{v}^2, \cdots, \breve{v}^{t+1}\big) - \mathbb{E}\Big[\psi\big(\widetilde{V}^*, \breve{V}_1, \breve{V}_2, \cdots, \breve{V}_{t+1}\big)\Big] = 0, \tag{F.16b}$$

*where $(\breve{U}_s, \breve{V}_{s+1})_{0\le s\le t}$ are defined in (F.3).*

*Proof of Proposition F.1.* By definitions of the auxiliary AMP (F.2) and the matrix $\widetilde{A}$ in (F.1), we have that for every $t \geq 0$,

$$\breve{u}^t = \widetilde{A}\mathring{v}^t - \breve{b}_t\mathring{u}^{t-1} = \frac{\lambda}{n}\langle\widetilde{v}^*, \mathring{v}^t\rangle\widetilde{u}^* + \widetilde{W}\mathring{v}^t - \breve{b}_t\mathring{u}^{t-1},$$

$$\mathring{v}^{t+1} = \widetilde{A}^\top\mathring{u}^t - \breve{c}_t\mathring{v}^t = \frac{\lambda}{n}\langle\widetilde{u}^*, \mathring{u}^t\rangle\widetilde{v}^* + \widetilde{W}^\top\mathring{u}^t - \breve{c}_t\mathring{v}^t.$$

For every $t \geq 0$, let us consider a pair of related iterates $p^t, q^{t+1}$ with initialization

$$\widetilde{p}^{-1} = 0_n, \quad \widetilde{q}^0 = \mathring{v}^0 \tag{F.17}$$

and update rules:

$$p^t = \widetilde{W}\widetilde{q}^t - \ell_t\widetilde{p}^{t-1}, \quad \widetilde{p}^t = \breve{g}_t(p^t + \breve{\mu}_t\widetilde{u}^*), \quad m_t = \frac{1}{n}\sum_{i=1}^n \partial_i\breve{g}_t(p^t + \breve{\mu}_t\widetilde{u}^*)_i,$$

$$q^{t+1} = \widetilde{W}^\top\widetilde{p}^t - m_t\widetilde{q}^t, \quad \widetilde{q}^{t+1} = \breve{f}_{t+1}(q^{t+1} + \breve{\nu}_{t+1}\widetilde{v}^*), \quad \ell_{t+1} = \frac{1}{n}\sum_{i=1}^d \partial_i\breve{f}_{t+1}(q^{t+1} + \breve{\nu}_{t+1}\widetilde{v}^*)_i,$$

$$\tag{F.18}$$

where $\breve{\mu}_t, \breve{\nu}_{t+1}$ are as in (F.6) to (F.8).

Informally, the above iterates are related to $\breve{u}^t, \mathring{v}^{t+1}$ via

$$p^t \text{ '=' } \breve{u}^t - \breve{\mu}_t\widetilde{u}^*, \quad q^{t+1} \text{ '=' } \mathring{v}^{t+1} - \breve{\nu}_{t+1}\widetilde{v}^*, \tag{F.19}$$

where the 'equalities' hold only in the large $n$ limit. These relations will be made formal in the rest of the proof.

The algorithm (F.18) takes the form of a standard AMP iteration with non-separable denoising functions as in [13, 34] for which the following state evolution result applies. For any $t \geq 0$ and uniformly pseudo-Lipschitz functions $\phi, \psi$ of finite order, it holds that

$$\operatorname*{p\text{-}lim}_{n\to\infty}\phi(\widetilde{u}^*, p^0, \cdots, p^t) - \mathbb{E}\Big[\phi(\widetilde{U}^*, \breve{\sigma}_0\breve{W}_{U,0}, \cdots, \breve{\sigma}_t\breve{W}_{U,t})\Big] = 0,$$

$$\operatorname*{p\text{-}lim}_{n\to\infty}\psi(\widetilde{v}^*, q^1, \cdots, q^{t+1}) - \mathbb{E}\Big[\psi(\widetilde{V}^*, \breve{\tau}_1\breve{W}_{V,1}, \cdots, \breve{\tau}_{t+1}\breve{W}_{V,t+1})\Big] = 0,$$

$$\tag{F.20}$$

where $(\widetilde{U}^*, \breve{\sigma}_0\breve{W}_{U,0}, \cdots, \breve{\sigma}_t\breve{W}_{U,t})$ and $(\widetilde{V}^*, \breve{\tau}_1\breve{W}_{V,1}, \cdots, \breve{\tau}_{t+1}\breve{W}_{V,t+1})$ are defined in (F.4) and (F.5). Note that [34] allows additional randomness independent of $\widetilde{W}$ that goes into the denoising functions. So the asymptotic guarantee in (F.20) holds for the joint tuple involving $\widetilde{U}^*, \widetilde{V}^*$ as well.

(F.20) immediately implies

$$\operatorname*{p\text{-}lim}_{n\to\infty}\phi(u^*, p^0 + \breve{\mu}_0\widetilde{u}^*, \cdots, p^t + \breve{\mu}_t\widetilde{u}^*) - \mathbb{E}\Big[\phi(U^*, \breve{U}_0, \cdots, \breve{U}_t)\Big] = 0,$$

$$\operatorname*{p\text{-}lim}_{n\to\infty}\psi(v^*, q^1 + \breve{\nu}_1\widetilde{v}^*, \cdots, q^{t+1} + \breve{\nu}_{t+1}\widetilde{v}^*) - \mathbb{E}\Big[\psi(V^*, \breve{V}_1, \cdots, \breve{V}_{t+1})\Big] = 0,$$

$$\tag{F.21}$$

where we recall the definition of $\breve{U}_t, \breve{V}_{t+1}$ in (F.3). We will show that

$$\operatorname*{p\text{-}lim}_{n\to\infty}\phi(u^*, p^0 + \breve{\mu}_0\widetilde{u}^*, \cdots, p^t + \breve{\mu}_t\widetilde{u}^*) - \phi(u^*, \breve{u}^0, \cdots, \breve{u}^t) = 0,$$

$$\operatorname*{p\text{-}lim}_{n\to\infty}\psi(v^*, q^1 + \breve{\nu}_1\widetilde{v}^*, \cdots, q^{t+1} + \breve{\nu}_{t+1}\widetilde{v}^*) - \psi(v^*, \mathring{v}^1, \cdots, \mathring{v}^{t+1}) = 0,$$

$$\tag{F.22}$$

which, when combined with (F.21), concludes the proof of Proposition F.1.

To show (F.22), suppose that $\phi, \psi$ are uniformly $L$-pseudo-Lipschitz of order $k$. Then by the triangle inequality,

$$\Big|\phi(u^*, p^0 + \breve{\mu}_0\widetilde{u}^*, \cdots, p^t + \breve{\mu}_t\widetilde{u}^*) - \phi(u^*, \breve{u}^0, \cdots, \breve{u}^t)\Big|$$

$$\leq L\left(\sum_{s=0}^t \frac{1}{\sqrt{n}}\|p^s + \breve{\mu}_s\widetilde{u}^* - \breve{u}^s\|_2\right)$$

$$\times \left[ 1 + \left( \frac{1}{\sqrt{n}} \|u^*\|_2 + \sum_{s=0}^{t} \frac{1}{\sqrt{n}} \|p^s + \breve{\mu}_s \widetilde{u}^*\|_2 \right)^{k-1} + \left( \frac{1}{\sqrt{n}} \|u^*\|_2 + \sum_{s=0}^{t} \frac{1}{\sqrt{n}} \|\breve{u}^s\|_2 \right)^{k-1} \right]$$

$$\leq L' \left( \sum_{s=0}^{t} \frac{1}{\sqrt{n}} \|p^s + \breve{\mu}_s \widetilde{u}^* - \breve{u}^s\|_2 \right)$$

$$\times \left[ 1 + \left( \frac{1}{\sqrt{n}} \|u^*\|_2 \right)^{k-1} + \sum_{s=0}^{t} \left( \frac{1}{\sqrt{n}} \|p^s + \breve{\mu}_s \widetilde{u}^*\|_2 \right)^{k-1} + \sum_{s=0}^{t} \left( \frac{1}{\sqrt{n}} \|\breve{u}^s\|_2 \right)^{k-1} \right],$$

for some $L'$ depending only on $t, k, L$. Similar manipulation gives

$$\left| \psi(v^*, q^1 + \breve{\nu}_1 \widetilde{v}^*, \cdots, q^{t+1} + \breve{\nu}_{t+1} \widetilde{v}^*) - \psi(v^*, \breve{v}^1, \cdots, \breve{v}^{t+1}) \right|$$

$$\leq L' \left( \sum_{s=1}^{t+1} \frac{1}{\sqrt{d}} \|q^s + \breve{\nu}_s \widetilde{v}^* - \breve{v}^s\|_2 \right)$$

$$\times \left[ 1 + \left( \frac{1}{\sqrt{d}} \|v^*\|_2 \right)^{k-1} + \sum_{s=1}^{t+1} \left( \frac{1}{\sqrt{d}} \|q^s + \breve{\nu}_s \widetilde{v}^*\|_2 \right)^{k-1} + \sum_{s=1}^{t+1} \left( \frac{1}{\sqrt{d}} \|\breve{v}^s\|_2 \right)^{k-1} \right].$$

Clearly, (F.22) holds if for every $t \geq 0$,

$$\underset{n \to \infty}{\text{p-lim}} \frac{1}{\sqrt{n}} \|p^t + \breve{\mu}_t \widetilde{u}^*\|_2 < \infty, \tag{F.23}$$

$$\underset{n \to \infty}{\text{p-lim}} \frac{1}{\sqrt{n}} \|\breve{u}^t\|_2 < \infty, \tag{F.24}$$

$$\underset{n \to \infty}{\text{p-lim}} \frac{1}{n} \|\breve{u}^t - (p^t + \breve{\mu}_t \widetilde{u}^*)\|_2^2 = 0, \tag{F.25}$$

$$\underset{d \to \infty}{\text{p-lim}} \frac{1}{\sqrt{d}} \|q^{t+1} + \breve{\nu}_{t+1} \widetilde{v}^*\|_2 < \infty, \tag{F.26}$$

$$\underset{d \to \infty}{\text{p-lim}} \frac{1}{\sqrt{d}} \|\breve{v}^{t+1}\|_2 < \infty, \tag{F.27}$$

$$\underset{d \to \infty}{\text{p-lim}} \frac{1}{d} \|\breve{v}^{t+1} - (q^{t+1} + \breve{\nu}_{t+1} \widetilde{v}^*)\|_2^2 = 0, \tag{F.28}$$

which, together with the following statements

$$\breve{\mu}_t < \infty, \quad \breve{\sigma}_t < \infty, \tag{F.29}$$

$$\breve{\nu}_{t+1} < \infty, \quad \breve{\tau}_{t+1} < \infty, \tag{F.30}$$

will be shown in the sequel by induction on $t \geq 0$.

**Base case.** Consider $t = 0$. From (F.21),

$$\underset{n \to \infty}{\text{p-lim}} \frac{1}{n} \|p^0 + \breve{\mu}_0 \widetilde{u}^*\|_2^2 = \underset{n \to \infty}{\text{p-lim}} \frac{1}{n} \mathbb{E}\left[ \|\breve{U}_0\|_2^2 \right] = \breve{\sigma}_0^2 + \breve{\mu}_0^2 \mathbb{E}\left[ \overline{\Xi}^{-1} \right], \tag{F.31}$$

where the last equality is by (F.3). Due to (F.13) and Assumption (A4), both $\breve{\mu}_0$ and $\breve{\sigma}_0$ are finite, so (F.29) holds for $t = 0$. Consequently, (F.23) also holds for $t = 0$.

Since by (F.17),

$$(p^0 + \breve{\mu}_0 \widetilde{u}^*) - \breve{u}^0 = \breve{\mu}_0 \widetilde{u}^* - \frac{\lambda}{n} \langle \widetilde{v}^*, \breve{v}^0 \rangle \widetilde{u}^*,$$

therefore (F.25) for $t = 0$ follows from (F.6) and Assumption (A4). This in turn implies, when combined with the finiteness of (F.31), that

$$\underset{n \to \infty}{\text{p-lim}} \frac{1}{\sqrt{n}} \|\breve{u}^0\|_2 < \infty, \tag{F.32}$$

verifying (F.24) for $t = 0$. Since $\breve{g}_0$ is uniformly pseudo-Lipschitz of finite order, so is the function $\frac{1}{n}\sum_{i=1}^{n}\partial_i\breve{g}_0(\cdot)_i$. (F.23) to (F.25) (for $t = 0$) together imply

$$\operatorname*{p\text{-}lim}_{n\to\infty}|m_0| < \infty, \quad \operatorname*{p\text{-}lim}_{n\to\infty}|m_0 - \breve{c}_0| = 0. \tag{F.33}$$

Using the the pseudo-Lipschitzness of $\breve{g}_0$ again,

$$\begin{aligned}
\operatorname*{p\text{-}lim}_{n\to\infty}\frac{1}{\sqrt{n}}\left\|\widetilde{p}^0 - \mathring{u}^0\right\|_2 &= \operatorname*{p\text{-}lim}_{n\to\infty}\frac{1}{\sqrt{n}}\left\|\breve{g}_0(p^0 + \breve{\mu}_0\widetilde{u}^*) - \breve{g}_0(\mathring{u}^0)\right\|_2 \\
&\leq \operatorname*{p\text{-}lim}_{n\to\infty} L\frac{\left\|(p^0 + \breve{\mu}_0\widetilde{u}^*) - \mathring{u}^0\right\|_2}{\sqrt{n}}\left[1 + \left(\frac{\left\|p^0 + \breve{\mu}_0\widetilde{u}^*\right\|_2}{\sqrt{n}}\right)^{k-1} + \left(\frac{\left\|\mathring{u}^0\right\|_2}{\sqrt{n}}\right)^{k-1}\right] \\
&= 0. \tag{F.34}
\end{aligned}$$

The last equality holds because of (F.25) (for $t = 0$) and the finiteness of (F.31) and (F.32).

To show (F.28) for $t = 0$, we use (F.2), (F.17) and (F.18) to write

$$(q^1 + \breve{\nu}_1\widetilde{v}^*) - \breve{v}^1 = \underbrace{\widetilde{W}^\top(\widetilde{p}^0 - \mathring{u}^0)}_{T_1} + \underbrace{\left(\breve{\nu}_1 - \frac{\lambda}{n}\langle\mathring{u}^0, \widetilde{u}^*\rangle\right)\widetilde{v}^*}_{T_2} - \underbrace{(m_0 - \breve{c}_0)\,\mathring{v}^0}_{T_3}.$$

By (F.34) and the Bai–Yin law [3],

$$\operatorname*{p\text{-}lim}_{d\to\infty}\frac{1}{d}\|T_1\|_2^2 = 0. \tag{F.35}$$

Using (F.34) again,

$$\begin{aligned}
\operatorname*{p\text{-}lim}_{n\to\infty}\frac{\lambda}{n}\langle\mathring{u}^0, \widetilde{u}^*\rangle &= \operatorname*{p\text{-}lim}_{n\to\infty}\frac{\lambda}{n}\langle\widetilde{p}^0, \widetilde{u}^*\rangle = \operatorname*{p\text{-}lim}_{n\to\infty}\frac{\lambda}{n}\langle\breve{g}_0(p^0 + \breve{\mu}_0\widetilde{u}^*), \widetilde{u}^*\rangle \\
&= \lim_{n\to\infty}\frac{\lambda}{n}\mathbb{E}\left[\langle\breve{g}_0(\breve{U}_0), \widetilde{u}^*\rangle\right] = \breve{\nu}_1, \tag{F.36}
\end{aligned}$$

where in the second line, the first equality is by (F.22) and the pseudo-Lipschitzness of $\breve{g}_0$, and the second equality is by the definition (F.8). We further show the finiteness of $\breve{\nu}_1$. Note that

$$\breve{\nu}_1 \leq \lim_{n\to\infty}\lambda\mathbb{E}\left[\frac{1}{n}\left\|\breve{g}_0(\breve{U}_0)\right\|_2^2\right]^{1/2}\mathbb{E}\left[\frac{1}{n}\left\|\widetilde{U}^*\right\|_2^2\right]^{1/2}.$$

The first term can be bounded as

$$\operatorname*{p\text{-}lim}_{n\to\infty}\frac{1}{n}\mathbb{E}\left[\left\|\breve{g}_0(\breve{U}_0)\right\|_2^2\right] \leq \operatorname*{p\text{-}lim}_{n\to\infty}L'\mathbb{E}\left[\left(1 + \left(\frac{1}{\sqrt{n}}\left\|\breve{U}_0\right\|_2\right)^k\right)^2\right] \leq \operatorname*{p\text{-}lim}_{n\to\infty}2L'\left(1 + \mathbb{E}\left[\left(\frac{1}{n}\left\|\breve{U}_0\right\|_2^2\right)^k\right]\right), \tag{F.37}$$

where the last step is the elementary inequality $(a + b)^2 \leq 2(a^2 + b^2)$. The RHS above is finite since

$$\frac{1}{n}\left\|\breve{U}_0\right\|_2^2 = \frac{\breve{\mu}_0^2}{n}\left\|\widetilde{U}^*\right\|_2^2 + \frac{\breve{\sigma}_0^2}{n}\left\|\breve{W}_{V,0}\right\|_2^2 \tag{F.38}$$

whose all moments are finite by finiteness of $\breve{\mu}_0, \breve{\sigma}_0$. This shows the first bound in (F.30) for $t = 0$. Recalling (F.36), we then have

$$\operatorname*{p\text{-}lim}_{n\to\infty}|T_2| = 0. \tag{F.39}$$

By (F.33),

$$\operatorname*{p\text{-}lim}_{n\to\infty}|T_3| = 0. \tag{F.40}$$

Therefore, (F.35), (F.39) and (F.40) altogether verify (F.28) for $t = 0$.

We then show (F.26) for $t = 0$. Since $\breve{\nu}_1 < \infty$, it suffices to only consider $q^1$. According to (F.17) and (F.18),

$$q^1 = \widetilde{W}^\top \widehat{p}^0 - m_0 \mathring{v}^0.$$

Pseudo-Lipschitzness of $\breve{g}_0$, finiteness (F.23) (for $t = 0$) and finiteness (F.33) jointly imply

$$\text{p-lim}_{n\to\infty} \frac{1}{\sqrt{n}} \|q^1\|_2 < \infty,$$

from which (F.26) follows. This combined with (F.28) (for $t = 0$) also implies (F.27) (for $t = 0$).

Finally, we are left with the second inequality in (F.30). From the definition (F.15),

$$\breve{\tau}_1^2 = \lim_{n\to\infty} \frac{1}{n} \mathbb{E}\left[\left\|\breve{g}_0(\breve{U}_0)\right\|_2^2\right]$$

which has already been shown to be finite; see (F.37). So the base case is finished.

**Induction step.** Assume that (F.23) to (F.28) all hold up to the $t$-th step (for an arbitrary $t \geq 1$). We now show that they hold for $t + 1$. The idea is similar to the base case. We briefly lay down the key steps for (F.23) to (F.25) and (F.29), and omit the verification of (F.26) to (F.28) and (F.30).

Using (F.3) and (F.21),

$$\text{p-lim}_{n\to\infty} \frac{1}{n} \left\|p^{t+1} + \breve{\mu}_{t+1}\widetilde{u}^*\right\|_2^2 = \breve{\sigma}_{t+1}^2 + \breve{\mu}_{t+1}^2 \mathbb{E}\left[\overline{\Xi}^{-1}\right]. \tag{F.41}$$

Using the definition (F.14) and the pseudo-Lipschitzness of $\breve{f}_{t+1}$,

$$\breve{\sigma}_{t+1}^2 = \lim_{n\to\infty} \frac{1}{n} \mathbb{E}\left[\left\|\breve{f}_{t+1}(\breve{V}_{t+1})\right\|_2^2\right] \leq \lim_{n\to\infty} 2L'\left(1 + \mathbb{E}\left[\left(\frac{1}{n}\left\|\breve{V}_{t+1}\right\|_2^2\right)^k\right]\right),$$

for some $L'$ depending only on $k, L$. The inequality is obtained in a similar way to (F.37). By induction hypothesis (F.30) and the compactness of $\text{supp}(\overline{\Sigma})$, all moments of

$$\frac{1}{n}\left\|\breve{V}_{t+1}\right\|_2^2 = \frac{\breve{\nu}_{t+1}^2}{n}\left\|\widetilde{V}^*\right\|_2^2 + \frac{\breve{\tau}_{t+1}^2}{n}\left\|\breve{W}_{V,t+1}\right\|_2^2$$

are finite. Therefore $\breve{\sigma}_{t+1}^2 < \infty$, giving the second bound in (F.29). Similarly, using the definition (F.7) and Cauchy–Schwarz,

$$\breve{\mu}_{t+1} = \lim_{d\to\infty} \frac{\lambda}{n}\mathbb{E}\left[\left\langle\widetilde{V}^*, \breve{f}_{t+1}(\breve{V}_{t+1})\right\rangle\right] \leq \lim_{d\to\infty} \frac{L'\lambda}{\sqrt{n}}\mathbb{E}\left[\left\|\widetilde{V}^*\right\|_2\left(1 + \left(\frac{1}{\sqrt{n}}\left\|\breve{V}_{t+1}\right\|_2\right)^k\right)\right]$$

$$\leq \lim_{d\to\infty} L'\lambda\mathbb{E}\left[\frac{1}{n}\left\|\widetilde{V}^*\right\|_2^2\right]^{1/2}\left(2\mathbb{E}\left[1 + \left(\frac{1}{n}\left\|\breve{V}_{t+1}\right\|_2^2\right)^k\right]\right)^{1/2},$$

which is again finite for the same reason as $\breve{\sigma}_{t+1}$, giving the first bound in (F.29). Therefore (F.41) is also finite, verifying (F.23) for $t + 1$.

We then show (F.25) for $t + 1$. Using the recursions (F.2) and (F.18),

$$p^{t+1} + \breve{\mu}_{t+1}\widetilde{u}^* - \breve{u}^{t+1} = \underbrace{\widetilde{W}(\widetilde{q}^{t+1} - \mathring{v}^{t+1})}_{T_1'} + \underbrace{\left(\breve{\mu}_{t+1} - \frac{\lambda}{n}\langle\widetilde{v}^*, \mathring{v}^{t+1}\rangle\right)\widetilde{u}^*}_{T_2'} - \underbrace{(\ell_{t+1}\widetilde{p}^t - \breve{b}_{t+1}\mathring{u}^t)}_{T_3'}.$$

Consider $T_1'$. Since (F.26) to (F.28) are assumed to hold, by pseudo-Lipschitzness of $\breve{f}_{t+1}$,

$$\text{p-lim}_{n\to\infty} \frac{1}{\sqrt{d}}\left\|\widetilde{q}^{t+1} - \mathring{v}^{t+1}\right\|_2 = 0. \tag{F.42}$$

This with the Bai-Yin law [3] gives

$$\text{p-lim}_{n\to\infty} \frac{1}{n}\|T_1'\|_2^2 = 0, \tag{F.43}$$

Consider $T_2'$. Recall that $\breve{\mu}_{t+1} < \infty$. Using (F.7), (F.21) and (F.42) and following the argument leading to (F.36), we have

$$\text{p-lim}_{n\to\infty}|T_2'| = 0. \tag{F.44}$$

Consider $T_3'$. By the triangle inequality,

$$\left\|\ell_{t+1}\widetilde{p}^t - \breve{b}_{t+1}\mathring{u}^t\right\|_2 \le \left|\ell_{t+1} - \breve{b}_{t+1}\right|\left\|\widetilde{p}^t\right\|_2 + \left|\breve{b}_{t+1}\right|\left\|\widetilde{p}^t - \mathring{u}^t\right\|_2.$$

Since (F.26) to (F.28) are assumed to hold, by pseudo-Lipschitzness of $\frac{1}{n}\sum_{i=1}^d \partial_i \breve{f}_{t+1}(\cdot)_i$,

$$\text{p-lim}_{n\to\infty}\left|\breve{b}_{t+1}\right| < \infty, \quad \text{p-lim}_{n\to\infty}\left|\ell_{t+1} - \breve{b}_{t+1}\right| = 0. \tag{F.45}$$

Similarly, pseudo-Lipschitzness of $\breve{g}_t$ and the hypothesis (F.23) to (F.25) ensures

$$\text{p-lim}_{n\to\infty}\frac{1}{\sqrt{n}}\left\|\widetilde{p}^t\right\|_2 < \infty, \quad \text{p-lim}_{n\to\infty}\frac{1}{\sqrt{n}}\left\|\widetilde{p}^t - \mathring{u}^t\right\|_2 = 0, \tag{F.46}$$

So combining (F.45) and (F.46), we have

$$\text{p-lim}_{n\to\infty}\frac{1}{n}\|T_3'\|_2^2 = 0. \tag{F.47}$$

(F.43), (F.44) and (F.47) altogether verify (F.25) for $t+1$, and therefore also (F.24) by (F.23).

The verification of (F.26) to (F.28) and (F.30) for $t+1$ is completely analogous and we do not repeat similar arguments. The proof is finally completed. □

## F.2 Proof of Proposition E.1

We will prove Proposition E.1 by reducing the AMP iteration (E.1) (and its associated state evolution (E.2) to (E.6)) to the auxiliary AMP (F.2) (and its associated state evolution (F.3) to (F.12), (F.14) and (F.15)).

Under the following change of variables

$$u^t := \Xi^{-1/2}\breve{u}^t, \qquad\qquad v^{t+1} := \Sigma^{-1/2}\breve{v}^{t+1}, \tag{F.48}$$

$$f_{t+1}(v^{t+1}) := \Sigma^{1/2}\breve{f}_{t+1}(\Sigma^{1/2}v^{t+1}), \qquad g_t(u^t) := \Xi^{1/2}\breve{g}_t(\Xi^{1/2}u^t), \tag{F.49}$$

(F.2) becomes

$$u^t = \Xi^{-1}A\Sigma^{-1}\widetilde{v}^t - b_t\Xi^{-1}\widetilde{u}^t, \qquad\qquad \widetilde{u}^t = g_t(u^t),$$
$$v^{t+1} = \Sigma^{-1}A^\top\Xi^{-1}\widetilde{u}^t - c_b\Sigma^{-1}\widetilde{v}^t, \qquad \widetilde{v}^{t+1} = f_{t+1}(v^{t+1}),$$

where $b_{t+1}, c_t$ are equal to $\breve{b}_{t+1}, \breve{c}_t$, respectively, but are expressed using the derivatives of $f_{t+1}, g_t$. Specifically,

$$c_t = \frac{1}{n}\sum_{i=1}^n \frac{\partial}{\partial \breve{u}_i^t}\breve{g}_t(\breve{u}^t)_i = \frac{1}{n}\sum_{i=1}^n \frac{\partial}{\partial \breve{u}_i^t}\left(\Xi^{-1/2}g_t(\Xi^{-1/2}\breve{u}^t)\right)_i$$
$$= \frac{1}{n}\sum_{i=1}^n\sum_{j=1}^n\sum_{k=1}^n(\Xi^{-1/2})_{i,j}(\Xi^{-1/2})_{k,i}\frac{\partial g_t(u^t)_j}{\partial u_k^t} = \frac{1}{n}\text{Tr}((\nabla g_t)\Xi^{-1}).$$

The second equality follows since by (F.49),

$$\breve{g}_t(\breve{u}^t) = \Xi^{-1/2}g_t(\Xi^{-1/2}\breve{u}^t).$$

The third equality is by the chain rule for derivatives (Proposition G.6). A similar computation gives

$$b_{t+1} = \frac{1}{n}\text{Tr}((\nabla f_{t+1})\Sigma^{-1}).$$

We now see that under the change of variables (F.48) and (F.49), the AMP iteration (E.1) can be cast as (F.2). Therefore, applying the same change of variables to the state evolution of (F.2) will produce the state evolution of (E.1). We describe the required modifications below.

The state evolution result in Proposition F.1 for the AMP in (F.2) says that the iterates $v^*, \breve{v}^1, \breve{v}^2, \cdots, \breve{v}^{t+1} \in \mathbb{R}^d$ and $u^*, \breve{u}^0, \breve{u}^1, \cdots, \breve{u}^t \in \mathbb{R}^n$ converge (in the sense of (F.16a) and (F.16b)) respectively to $V^*, \breve{V}_1, \breve{V}_2, \cdots, \breve{V}_{t+1} \in \mathbb{R}^d$ and $U^*, \breve{U}_1, \breve{U}_2, \cdots, \breve{U}_t \in \mathbb{R}^n$. Recall that AMPs (E.1) and (F.2) operate on the following matrices respectively:

$$\widetilde{A} = \frac{\lambda}{n}(\Xi^{-1/2}u^*)(\Sigma^{-1/2}v^*)^\top + \widetilde{W}, \quad \Xi^{-1}A\Sigma^{-1} = \frac{\lambda}{n}(\Xi^{-1}u^*)(\Sigma^{-1}v^*)^\top + \Xi^{-1/2}\widetilde{W}\Sigma^{-1/2}.$$

In view of (F.48), to obtain the analogous state evolution result for the AMP in (E.1), the definition (F.3) of $\breve{U}_t, \breve{V}_{t+1}$ should be multiplied by $\Xi^{-1/2}, \Sigma^{-1/2}$ respectively. This gives the new definition of $U_t, V_{t+1}$ in (E.6). By the relation (F.49), the parameters $\breve{\mu}_t, \breve{\nu}_{t+1}$ in $\breve{U}_t, \breve{V}_{t+1}$ should be modified as follows: replace $\breve{f}_t(\breve{V}_t), \breve{g}_t(\breve{U}_t)$ in the recursive equations (F.6) to (F.8) with $\Sigma^{-1/2}f_t(V_t), \Xi^{-1/2}g_t(U_t)$. This gives the new definition of $\mu_t, \nu_t$ in (E.5). Similar operations map equations (F.9) to (F.12), (F.14) and (F.15) to equations (E.3) and (E.4). Finally, under the new definition of $U_t, V_{t+1}$, the convergence result (F.16a) and (F.16b) translates to (E.8a) and (E.8b), which completes the proof.

# G   Auxiliary lemmas

**Proposition G.1** (Gaussian integral)**.** *Let $A \in \mathbb{R}^{d \times d}$ be a positive-definite matrix and $b \in \mathbb{R}^d$. Then*

$$\int_{\mathbb{R}^d} \exp\left(-\frac{1}{2}x^\top A x + b^\top x\right) \mathrm{d}x = \sqrt{\frac{(2\pi)^d}{\det(A)}} \exp\left(\frac{1}{2}b^\top A^{-1}b\right).$$

**Proposition G.2.** *Let $V \sim Q^{\otimes d}$ where $Q$ is a fixed distribution on $\mathbb{R}$ with mean 0. Let $B \in \mathbb{R}^{d \times d}$ denote a sequence (indexed by $d$) of deterministic matrices such that the empirical spectral distribution of $\frac{1}{d}B$ converges to the law of a random variable $\overline{B}$. Then*

$$\lim_{d \to \infty} \frac{1}{d}\mathbb{E}[V^\top B V] = \mathbb{E}[\overline{V}^2]\mathbb{E}[\overline{B}]$$

*where $\overline{V} \sim Q$.*

**Proposition G.3.** *Let*

$$(W_1, W_2) \sim \mathcal{N}\left(\begin{bmatrix} 0_d \\ 0_d \end{bmatrix}, \begin{bmatrix} \sigma_1^2 & \rho \\ \rho & \sigma_2^2 \end{bmatrix} \otimes I_d\right).$$

*Let $B \in \mathbb{R}^{d \times d}$ denote a sequence (indexed by $d$) of deterministic matrices such that the empirical spectral distribution of $\frac{1}{d}B$ converges to the law of a random variable $\overline{B}$. Then*

$$\lim_{d \to \infty} \frac{1}{d}\mathbb{E}[W_1^\top B W_2] = \rho\mathbb{E}[\overline{B}].$$

**Proposition G.4** (Nishimori identity)**.** *Let $(X, Y)$ be two random variables. Let $k \geq 1$ and $X_1, \cdots, X_k$ be $k$ i.i.d. samples (given $Y$) from the distribution $\mathrm{law}(X \mid Y)$. Denote by $\langle \cdot \rangle, \mathbb{E}[\cdot]$ the expectations with respect to $\mathrm{law}(X \mid Y)$ and $\mathrm{law}(X, Y)$, respectively. Then for all continuous bounded function $f$, it holds that*

$$\mathbb{E}[\langle f(Y, X_1, \cdots, X_k)\rangle] = \mathbb{E}[\langle f(Y, X_1, \cdots, X_{k-1}, X)\rangle].$$

**Proposition G.5** (Conditional distribution of Gaussians)**.** *Let $d \geq 2$ and $1 \leq p \leq d - 1$ be integers. Let*

$$\begin{bmatrix} G_1 \\ G_2 \end{bmatrix} \sim \mathcal{N}\left(\begin{bmatrix} \mu_1 \\ \mu_2 \end{bmatrix}, \begin{bmatrix} \Sigma_{1,1} & \Sigma_{1,2} \\ \Sigma_{1,2}^\top & \Sigma_{2,2} \end{bmatrix}\right)$$

*be a $d$-dimensional Gaussian random vector, where $G_1 \in \mathbb{R}^p, G_2 \in \mathbb{R}^{d-p}, \mu_1 \in \mathbb{R}^p, \mu_2 \in \mathbb{R}^{d-p}, \Sigma_{1,1} \in \mathbb{R}^{p \times p}, \Sigma_{1,2} \in \mathbb{R}^{p \times (d-p)}, \Sigma_{2,2} \in \mathbb{R}^{(d-p) \times (d-p)}$. Then for any $g_2 \in \mathbb{R}^{d-p}$, the distribution of $G_1$ conditioned on $G_2 = g_2$ is given by $G_1 \mid \{G_2 = g_2\} \sim \mathcal{N}(\mu_1', \Sigma_1')$ where*

$$\mu_1' = \mu_1 + \Sigma_{1,2}\Sigma_{2,2}^{-1}(g_2 - \mu_2) \in \mathbb{R}^p, \quad \Sigma_1' = \Sigma_{1,1} - \Sigma_{1,2}\Sigma_{2,2}^{-1}\Sigma_{1,2}^\top \in \mathbb{R}^{p \times p}$$

*and $\Sigma_{2,2}^{-1}$ denotes the generalized inverse of $\Sigma_{2,2}$.*

**Proposition G.6** (Chain rule of derivatives). *Let $A \in \mathbb{R}^{n \times n}$ and $f \colon \mathbb{R}^n \to \mathbb{R}^n$. Let $x \in \mathbb{R}^n$ and $\widetilde{x} := Ax$. Then for any $i, j \in [n]$,*

$$\frac{\partial}{\partial x_i}(Af(Ax))_i = \sum_{j=1}^{n} \sum_{k=1}^{n} A_{i,j} A_{k,i} \frac{\partial f(\widetilde{x})_j}{\partial \widetilde{x}_k},$$

*where $f(\widetilde{x})_j \in \mathbb{R}$ denotes the $j$-th ($j \in [n]$) output of $f(\widetilde{x}) \in \mathbb{R}^n$.*

*Proof.* The proof follows from elementary applications of the chain rule for derivatives. Writing $A = [a_1 \ \cdots \ a_n]^\top$ and $f = [f_1 \ \cdots \ f_n]^\top$, we have

$$\frac{\partial}{\partial x_i} f_j(Ax) = \frac{\partial}{\partial x_i} f_j(\langle a_1, x \rangle, \cdots, \langle a_n, x \rangle) = \sum_{k=1}^{n} \partial_k f_j(Ax) \frac{\partial \langle a_k, x \rangle}{\partial x_i} = \sum_{k=1}^{n} A_{k,i} \partial_k f_j(\widetilde{x}),$$

where $\partial_k f_j$ denotes the partial derivative of $f_j \colon \mathbb{R}^n \to \mathbb{R}$ with respect to its $k$-th argument. Then,

$$\frac{\partial}{\partial x_i}(Af(Ax))_i = \sum_{j=1}^{n} A_{i,j} \frac{\partial}{\partial x_i} f_j(Ax) = \sum_{j=1}^{n} \sum_{k=1}^{n} A_{i,j} A_{k,i} \partial_k f_j(\widetilde{x}),$$

as claimed. $\qquad\square$

**Proposition G.7** (Stein's lemma [68]). *Let $W \sim \mathcal{N}(0, \sigma^2)$ and let $f \colon \mathbb{R} \to \mathbb{R}$ be such that both expectations below exist. Then $\mathbb{E}[W f(W)] = \sigma^2 \mathbb{E}[f'(W)]$.*

**Proposition G.8** (Gaussian Poincaré inequality [16, Theorem 3.20]). *Let $X \sim \mathcal{N}(0_n, I_n)$ and $f \colon \mathbb{R}^n \to \mathbb{R}$ a differentiable function. Then*

$$\mathrm{Var}[f(X)] \leq \mathbb{E}\left[\|\nabla f(X)\|_2^2\right].$$

**Proposition G.9** (Bounded difference inequality [16, Corollary 3.2]). *Let $\mathcal{U} \subset \mathbb{R}$ and $f \colon \mathcal{U}^n \to \mathbb{R}$ a function such that there exist $c = (c_1, \cdots, c_n) \in \mathbb{R}^n_{\geq 0}$ satisfying for all $i \in [n]$,*

$$\sup_{(x_1, \cdots, x_n, x_i') \in \mathcal{U}^{n+1}} |f(x_1, \cdots, x_{i-1}, x_i, x_{i+1}, \cdots, x_n) - f(x_1, \cdots, x_{i-1}, x_i', x_{i+1}, \cdots, x_n)| \leq c_i.$$

*Then if $X \in \mathcal{U}^n$ is a random vector consisting of independent elements, we have $\mathrm{Var}[f(X)] \leq \|c\|_2^2/4$.*

**Proposition G.10** ([62, Lemma 3.2]). *If $f, g$ are differentiable convex functions, then for any $a \in \mathbb{R}$ and $a' > 0$,*

$$|f'(a) - g'(a)| \leq g'(a + a') - g'(a - a') + B/a',$$

*where*

$$B := |f(a + a') - g(a + a')| + |f(a - a') - g(a - a')| + |f(a) - g(a)|.$$

**Definition G.1** (Monotone conjugate). *Let $f \colon \mathbb{R}_{\geq 0} \to \mathbb{R}$ be a non-decreasing convex function. Its monotone conjugate $f^*$ is defined as*

$$f^*(x) = \sup_{y \geq 0} xy - f(y).$$

**Proposition G.11** ([53, Proposition C.1]). *Let $I \subset \mathbb{R}$ be an interval and $(f_n \colon I \to \mathbb{R})_n$ be a sequence of convex functions converging pointwise to $f$. Then for all $t \in I$ such that the following quantities exist,*

$$\lim_{s \uparrow t} f'(s) \leq \liminf_{n \to \infty} \lim_{s \uparrow t} f_n'(s) \leq \limsup_{n \to \infty} \lim_{s \downarrow t} f_n'(s) \leq \lim_{s \downarrow t} f'(s).$$

**Proposition G.12** ([53, Proposition C.6]). *Let $f, g \colon \mathbb{R}_{\geq 0} \to \mathbb{R}$ be strictly convex differentiable functions and*

$$\mathcal{C} := \left\{ (q_1, q_2) \in \mathbb{R}^2_{\geq 0} \colon q_2 = f'(q_1), q_1 = g'(q_2) \right\}.$$

*Then*

$$\sup_{(q_1, q_2) \in \mathcal{C}} f(q_1) + g(q_2) - q_1 q_2 = \sup_{q_1, q_2 \geq 0} q_1 q_2 - f^*(q_2) - g^*(q_1) = \sup_{q_1 \geq 0} \inf_{q_2 \geq 0} f(q_1) + g(q_2) - q_1 q_2,$$

*and $\sup_{(q_1, q_2) \in \mathcal{C}}$ and $\sup_{q_1, q_2 \geq 0}$ are achieved at the same $(q_1^*, q_2^*)$.*

