# OpenReview forum: "Matrix Denoising with Doubly Heteroscedastic Noise: Fundamental Limits and Optimal Spectral Methods"
_NeurIPS.cc/2024/Conference — NeurIPS 2024 poster_

### Official Review · Reviewer_Fvn7 · 2024-06-30

**Soundness:** 4
**Presentation:** 4
**Contribution:** 3
**Rating:** 7
**Confidence:** 3

**Summary:**

This paper studies the following "matrix denoising"/low-rank estimation problem. Given a rectangular matrix $Y = uv^\top + W$, the goal is to recover the rank-one factors $u,v$, when $W$ is random, with as little $\ell_2$ error as possible.

A classic and well-studied setting takes $W$ to have iid entries; often one even assumes that the entries are Gaussian. This setting has been studied in random matrix theory, high-dimensional statistics, and more recently via techniques from statistical physics.

The iid noise assumption is quite strong, and so a more recent line of works aims to relax that assumption by allowing correlations among the entries of $W$. In the "two-sided" heteroscedastic noise model, $W$ takes the form $\Theta^{1/2} \cdot W' \cdot \Sigma^{1/2}$, where $\Theta$ and $\Sigma$ are PSD matrices and $W'$ has iid Gaussian entries. The assumption is that $\Theta$ and $\Sigma$ are known.

The main contributions of this paper are twofold:

- a new spectral estimator whose performance improves over more naive spectral methods for recovery of $u$ and $v$.
- a proof that under slightly stronger assumptions ($u$,$v$ Gaussian), the spectral estimator obtains nontrivial $\ell_2$ error whenever this is information-theoretically possible, and even obtains information-theoretically optimal $\ell_2$ error when the heteroscedasticity is one-sided.

Along the way, the paper establishes a simple formula for the information-theoretic signal-to-noise threshold governing when nontrivial recovery of $u$ and $v$ is possible in this model. The paper also performs numerical experiments on synthetic data to validate the theory.

**Strengths:**

- Well-written and easy to read first 9 pages
- Thorough mathematical investigation of the two-sided heteroscedastic spiked matrix model
- New algorithm with nice optimality guarantees

**Weaknesses:**

My main reservation is that, as with a lot of papers using stat phys techniques, one gets the feeling that the assumptions are sort of designed to make the mathematical techniques work. I think the biggest offender here is the assumption that $\Sigma$ and $\Theta$ are known. Less major are the Gaussian-ness assumptions, the assumption that $n,d$ are of comparable order, and the assumption that the empirical spectral distributions of $\Theta$ and $\Sigma$ converge -- I think these allow the use of asymptotic methods, basically.

**Questions:**

none

**Limitations:**

Yes

---

> ### Author Rebuttal · Authors · 2024-08-04
>
> We thank Reviewer Fvn7 for the insightful comments and positive evaluation. Below we address the comment concerning our various assumptions.
>
> We agree with the reviewer that the Gaussian-ness assumptions, the assumption that
> $n,d$ are of comparable order, and the assumption that the empirical spectral distributions of $\Xi, \Sigma$ converge are mild, and they basically allow the use of asymptotic methods.
>
> We now make a few comments on the assumption that $\Xi, \Sigma$ are known, and we will add this discussion to the revision.
>
> * In some settings, it is possible to estimate $\Xi, \Sigma$ (even consistently). This is the case if such matrices possess additional structures, e.g., if they are sparse [R4], if their inverses are sparse [R5] or if they are circulant or Toeplitz [R6]. We refer to the survey [R7] for detailed results on estimating structured high-dimensional covariance matrices.
>
> * Recently [R8] addresses the challenge of unknown covariances by considering a modified model where one additionally observes an independent copy of noise. The statistician can then estimate the covariance from the noise-only observation and use it as a surrogate of the true covariance for estimating the signals from the spiked model. It’s possible to derive similar results in the doubly heteroskedastic setting considered in our paper.
>
> * Finally, if the covariances are completely unknown, then our model (with Gaussian priors) is equivalent to a spiked matrix model with a certain bi-rotationally invariant noise. This problem is expected to exhibit rather different behaviors than when the covariances are known. See [R9] and [R10] for recent progress on understanding the statistical and computational limits for such models.
>
> ---
>
> [R4] T. T. Cai, H. H. Zhou, "Optimal rates of convergence for sparse covariance matrix estimation", Annals of Statistics, 2012.
>
> [R5] M. Yuan, "High dimensional inverse covariance matrix estimation via linear programming", Journal of Machine Learning Research, 2010.
>
> [R6] T. T. Cai, Z. Ren, H. H. Zhou, "Optimal rates of convergence for estimating Toeplitz covariance matrices", Probability Theory and Related Fields, 2013.
>
> [R7] T. T. Cai, Z. Ren, H. H. Zhou, "Estimating structured high-dimensional covariance and precision matrices: Optimal rates and adaptive estimation", Electronic Journal of Statistics, 2016.
>
> [R8] M. Gavish, W. Leeb, E. Romanov, "Matrix denoising with partial noise statistics: optimal singular value shrinkage of spiked F-matrices", Information and Inference: A Journal of the IMA, 2023.
>
> [R9] J. Barbier, F. Camilli, M. Mondelli, Y. Xu, "Information limits and Thouless-Anderson-Palmer equations for spiked matrix models with structured noise." arXiv preprint arXiv:2405.20993, 2024.
>
> [R10] R. Dudeja, S. Liu, J. Ma, "Optimality of Approximate Message Passing Algorithms for Spiked Matrix Models with Rotationally Invariant Noise", arXiv preprint arXiv:2405.18081, 2024.

---

> > ### Comment · Reviewer_Fvn7 · 2024-08-12
> >
> > Thanks for following up!

---

### Official Review · Reviewer_hDgq · 2024-07-12

**Soundness:** 3
**Presentation:** 3
**Contribution:** 2
**Rating:** 7
**Confidence:** 2

**Summary:**

The authors study how to recover a rank one spike corrupted by doubly heteroscedastic gaussian noise in the high dimensional regime. We are given a condition on the signal to noise ratio to indicate whether it's information-theoretically possible to have a non-trivial recovery of the spike. If this is satisfied then there is spectral estimator which can obtain non-trivial recovery. In particular cases this estimator is also Bayes optimal.

**Strengths:**

The paper is well written and does a really good job at introducing the main ideas in an intuitive way. While this kind of spectral estimators are well known in the physics literature, their application to such a noise model is an interesting generalisation.

I believe this paper thoroughly explores this denoising problem, with the only easily achievable extension being looking at a rank $r$ spike in the signal (where r is a constant in d,n), which I am sure however wouldn't alter the results significantly. Thus this is in my opinion quite a solid contribution with not much room for improvement.

**Weaknesses:**

The paper introduces no significant advances in the theoretical tools or understanding of matrix denoising as all the tools used are essentially well known. I believe one issue in the writing is its lack of clarity in stating which results are completely rigorous and which aren't. The authors are upfront in saying they are guided by physics-inspired heuristics, but reading in the appendix it seems like (at least in some sections) the derivations are quite solid. A tangible improvement to the writing would be to state explicitly which results are conjectures and which are theorems.

I think it would greatly improve the readability of figures 2 and 3 to have the error on the mean instead of the std.

I would add a sentence describing more explicitly the difference and respective advantages of (3.1) (where the spectrum is O(1)) and (4.1) (where the elements of Y are O(1)).

Small typos:

1. Line 203: $\sigma_2$ is not directly defined. You will only do so in Theorem 5.1
2. Line 188: you invoke the Nishimori identity but don't state it in the main text
3. Line 200: you say that $\eta>0$. While I also expect this to be true, I think you mean to say that they are real. The same applies to all the square roots in 5.3 and 5.4


Finally, not exactly a typo but I personally find it confusing to use $\bar \Sigma$ and $\bar \Xi$ when one wants to average over the singular values of $\Sigma$ and $\Xi$. I would prefer having an explicit integral over the singular value PDF.

**Questions:**

Would it be possible to look at the singular values of $A$ before and after the pre-processing? Can we clearly see a spike emerging if 5.1 is true by doing the pre-processing?

You state that AMP has the fundamental limitation that requires a "warm start" to be effective. While this is true, initialising the estimator at random from the prior should allow it to have non-zero overlap in O(log(d)) steps. Do you see this in your numerics?

Could you run AMP after using the spectral estimator and put the additional lines and simulation dots in figures 2, 3?

**Limitations:**

The limitations are correctly addressed in section 6. There is no negative societal impact of this work.

---

> ### Author Rebuttal · Authors · 2024-08-04
>
> We thank Reviewer hDgq for carefully reading the manuscript and for the insightful comments and suggestions. Below we address each point raised in the review separately.
>
> **I believe one issue in the writing is its lack of clarity in stating which results are completely rigorous and which aren't**
>
> While our approach is indeed inspired by statistical physics, we will clarify that *all* our results (i.e., Proposition 4.1, Theorem 4.2, Corollary 4.3, Theorem 4.4, Theorem 5.1 and Corollary 5.2) are mathematically rigorous, with the only technical condition being “(5.1) implies $\sigma_2^* < 1$” in Theorem 5.1 that we only managed to verify numerically, but not analytically.
>
> For the information-theoretic results, our proof uses the Gaussian interpolation method which is a rigorous technique originating in physics.
>
> For results on spectral estimators, the analysis is inspired by Bayes-AMP. A similar approach (albeit for different learning problems) is put forward e.g. in [R1] and [R2]. However, in sharp contrast with the two works above which only provide heuristics, our results are completely rigorous. In fact, a key element of novelty in our paper is precisely to give an exact asymptotic characterization of spectral estimators via AMP tools. Thus, the heuristics on pages 8-9 are just to offer the readers an intuition on how the spectral estimators given in (5.4) arise from Bayes-AMP. We will make this clear in the revised version.
>
> **I think it would greatly improve the readability of figures 2 and 3 to have the error on the mean instead of the std.**
>
> In Figures 2-3, we report the mean averaged over 20 trials, as well as the error bars representing 1 standard deviation from the mean. Please let us know if you have any suggestions on how to improve the readability of the figures.
>
> **I would add a sentence describing more explicitly the difference and respective advantages of (3.1) (where the spectrum is O(1)) and (4.1) (where the elements of Y are O(1)).**
>
> The different scalings in (3.1) and (4.1) are purely for mathematical convenience. For the analysis of spectral estimators, it’s more convenient to have matrices whose operator norms are of constant order. For the information-theoretic analysis, it is customary in the literature to have Hamiltonians on the order of $n$ and then normalize the free energy (i.e., the expected log partition function) by $1/n$. In principle, all results and proofs can be written under a single scaling. We will clarify this in the revised version.
>
> **Small typos**
>
> Thank you for spotting the typos, we will revise accordingly.
>
> **Usage of $\overline{\Xi}, \overline{\Sigma}$**
>
> We will make it clear that all expectations involving these random variables are computed as integrals against the limiting spectral distributions of $\Xi, \Sigma$, as is common in the random matrix theory literature.
>
> **Would it be possible to look at the singular values of $A$ before and after the pre-processing? Can we clearly see a spike emerging if 5.1 is true by doing the pre-processing?**
>
> Yes, we clearly see a spike emerging if (5.1) holds by doing the pre-processing. Thank you for this excellent suggestion. We have added a plot in the PDF attached to the global response that shows the presence of spectral outliers in $A^*$, as well as the absence of such outliers in $A$. We will add this plot to the revision.
>
>
> **Initialising the estimator at random from the prior should allow it to have non-zero overlap in O(log(d)) steps**
>
> For some problems, AMP with random initialization may be able to attain performance similar to AMP with warm start (such as spectral initialization). Such a phenomenon has been empirically observed for various PCA and regression problems, including ours. However, proving such a behavior largely remains open. To our knowledge, the only progress is the recent work of Li, Fan and Wei [R3] for $\mathbb{Z}_2$ synchronization, i.e., rank-1 matrix estimation with Rademacher prior and GOE noise (a setting much simpler than the one considered here).
>
> **Could you run AMP after using the spectral estimator and put the additional lines and simulation dots in figures 2, 3?**
>
> All experiments in Figures 2,3 are for Gaussian priors and, in this case, our spectral estimators attain the same asymptotic performance as Bayes-AMP, as the heuristics on page 8-9 suggest. There is therefore no advantage of running AMP with spectral initialization, as opposed to the spectral method alone. Furthermore, in the special case of Figure 2a, our spectral estimators alone are information-theoretically optimal.
>
> ---
>
> [R1] A. Maillard, F. Krzakala, Y. M. Lu, L. Zdeborová, "Construction of optimal spectral methods in phase retrieval", Mathematical and Scientific Machine Learning, 2022.
>
> [R2] E. Troiani, Y. Dandi, L. Defilippis, L. Zdeborová, B. Loureiro, F. Krzakala, "Fundamental limits of weak learnability in high-dimensional multi-index models", arXiv preprint arXiv:2405.15480, 2024.
>
> [R3] G. Li, W. Fan, Y. Wei, "Approximate message passing from random initialization with applications to Z 2 synchronization", Proceedings of the National Academy of Sciences, 2023.

---

> > ### Comment · Reviewer_hDgq · 2024-08-13
> >
> > Thanks for your detailed rebuttal, I will raise my score accordingly.
> >
> > For the figure, I am slightly surprised the fluctuations are so large, I guess trying larger sizes might be doable and make the presentation clearer.

---

### Official Review · Reviewer_S4fP · 2024-07-13

**Soundness:** 4
**Presentation:** 4
**Contribution:** 4
**Rating:** 8
**Confidence:** 1

**Summary:**

This paper considers the problem of matrix denoising. Given an observation X = A + W where W is noise, our goal is to estimate A, which is typically low-rank. Unlike previous works, this paper treats the case where W is doubly heteroscedastic. The authors identify a condition for non-trivial estimation of the signal vectors, along with an accompanying spectral algorithm that succeeds whenever that condition holds, under a technical condition.

**Strengths:**

The paper addresses an important problem using a novel approach using statistical physics/ AMP concepts. The results, both theoretical and empirical, are strong and add considerably to the literature.

**Weaknesses:**

Strictly speaking, you do not show that whitening fails, only that the whitened matrix does not match the proposed AMP approach (lines 262-265).

**Questions:**

Initially you say that you believe (5.1) implies \sigma_2^{\star} < 1, and later you say that you believe these conditions are equivalent. Could you please clarify which one it is?

**Limitations:**

Yes

---

> ### Author Rebuttal · Authors · 2024-08-04
>
> We thank Reviewer S4fP for the positive evaluation. Below we address the comments and questions.
>
> **Strictly speaking, you do not show that whitening fails, only that the whitened matrix does not match the proposed AMP approach (lines 262-265).**
>
> We will make the following 2 clarifications regarding lines 262-265.
>
> 1. One can repeat the analysis of an AMP that operates on the whitened matrix $\Xi^{-1/2} A \Sigma^{-1/2}$. The fixed point equations of the resulting state evolution do not match the information-theoretically optimal one in (4.5). In particular, the weak recovery threshold coming out of this approach is strictly larger than the optimal one in (4.6), as long as at least one of $\Xi, \Sigma$ is not a multiple of the identity. Since these derivations led to suboptimal results, the details were left out from the paper.
> 1. Our optimal spectral estimator is motivated by Bayes-AMP. If one instead considers spectral estimators associated with the whitened matrix $\Xi^{-1/2} A \Sigma^{-1/2}$, then the resulting weak recovery threshold and estimation error (overlap and matrix MSE) are all suboptimal. The exact asymptotic values of these quantities can be retrieved from Leeb–Romanov [40] and Leeb [41]. Numerical results and the corresponding theoretical predictions for whitened spectral estimators are shown in Figure 2; note its suboptimality compared to our spectral estimator and the information-theoretic limit.
>
> To summarize, the whitening approach is provably suboptimal for both AMP and spectral estimators.
>
> **Initially you say that you believe (5.1) implies \sigma_2^{\star} < 1, and later you say that you believe these conditions are equivalent. Could you please clarify which one it is?**
>
> We apologize for the confusion, and we will clarify this point in the revision. We believe that these two conditions are actually equivalent. However, the proof of Theorem 5.1 only requires one direction: (5.1) implies $\sigma_2^* < 1$. Therefore, we formally only make the minimal conjecture of (5.1) implying $\sigma_2^* < 1$.

---

> > ### Comment · Reviewer_S4fP · 2024-08-12
> > **Reply**
> >
> > Thank you for your rebuttal!

---

### Author Rebuttal · Authors · 2024-08-04

We thank the reviewers for their reviews and we have responded to their comments separately below. In this common response, we attach a plot that shows the presence of spectral outliers in $A^*$, as well as their absence in $A$. We will add this plot to the revision.

---

### Decision · Program_Chairs · 2024-09-25

**Decision:**

Accept (poster)

**Comment:**

There is an agreement among the Area Chair and the reviewers that this is a well-written and valuable contribution to the line of work on spiked matrix models with structured noise. I therefore recommend acceptance as a poster. I do not recommend highlighting this paper as a spotlight or oral since its contribution is along a well-established line of work. The paper's novelty is that it deals with yet another model of noise that, while interesting, it does not seem more relevant in practice than others. On the technical side, while the paper is a very solid contribution, again, it does not make a significantly novel technical contribution that would likely impact a large number of follow-up works.